# Data analysis and modeling pipelines for controlled networked social science experiments

**Vanessa Cedeno-Mieles**[1,2]*, **Zhihao Hu**[3], **Yihui Ren**[7], **Xinwei Deng**[3], **Noshir Contractor**[9], **Saliya Ekanayake**[10], **Joshua M. Epstein**[11], **Brian J. Goode**[5], **Gizem Korkmaz**[4], **Chris J. Kuhlman**[4], **Dustin Machi**[4], **Michael Macy**[12], **Madhav V. Marathe**[4,6], **Naren Ramakrishnan**[1,8], **Parang Saraf**[8], **Nathan Self**[8]

**1** Department of Computer Science, Virginia Tech, Blacksburg, VA, United States of America, **2** Escuela Superior Politécnica del Litoral, ESPOL, Guayaquil, Ecuador, **3** Department of Statistics, Virginia Tech, Blacksburg, VA, United States of America, **4** Biocomplexity Institute & Initiative, University of Virginia, Charlottesville, VA, United States of America, **5** Biocomplexity Institute, Virginia Tech, Blacksburg, VA, United States of America, **6** Department of Computer Science, University of Virginia, Charlottesville, VA, United States of America, **7** Computational Science Initiative, Brookhaven National Laboratory, Upton, NY, United States of America, **8** Discovery Analytics Center, Virginia Tech, Blacksburg, VA, United States of America, **9** Department of Industrial Engineering and Management Sciences, Northwestern University, Evanston, IL, United States of America, **10** Lawrence Berkeley National Laboratory, Berkeley, CA, United States of America, **11** Department of Epidemiology, New York University, New York, NY, United States of America, **12** Department of Sociology, Cornell University, Ithaca, NY, United States of America

* vcedeno@espol.edu.ec

**Data Availability Statement:** All relevant data are within the manuscript and its Supporting information files.

## Abstract

There is large interest in networked social science experiments for understanding human behavior at-scale. Significant effort is required to perform data analytics on experimental outputs and for computational modeling of custom experiments. Moreover, experiments and modeling are often performed in a cycle, enabling iterative experimental refinement and data modeling to uncover interesting insights and to generate/refute hypotheses about social behaviors. The current practice for social analysts is to develop tailor-made computer programs and analytical scripts for experiments and modeling. This often leads to inefficiencies and duplication of effort. In this work, we propose a pipeline framework to take a significant step towards overcoming these challenges. Our contribution is to describe the design and implementation of a software system to automate many of the steps involved in analyzing social science experimental data, building models to capture the behavior of human subjects, and providing data to test hypotheses. The proposed pipeline framework consists of formal models, formal algorithms, and theoretical models as the basis for the design and implementation. We propose a formal data model, such that if an experiment can be described in terms of this model, then our pipeline software can be used to analyze data efficiently. The merits of the proposed pipeline framework is elaborated by several case studies of networked social science experiments.

**Funding:** This work has been partially supported by DARPA Cooperative Agreement D17AC00003 (NGS2), DTRA CNIMS (Contract HDTRA1-11-D-0016- 0001), NSF DIBBS Grant ACI-1443054, NSF BIG DATA Grant IIS-1633028, NSF CRISP 2.0 Grant 1916670, NSF Grants DGE-1545362 and IIS-1633363, and ARL Grant W911NF-17-1-0021. The U.S. Government is authorized to reproduce and distribute reprints for Governmental purposes notwithstanding any copyright annotation thereon. The views and conclusions contained herein are those of the authors and should not be interpreted as necessarily representing the official policies or endorsements, either expressed or implied, of DARPA, DTRA, NSF, ARL, or the U.S. Government. The funders had no role in study design, data collection and analysis, decision to publish, or preparation of the manuscript.

**Competing interests:** The authors have declared that no competing interests exist.

# 1 Introduction

## 1.1 Background and motivation

Online controlled **ne**tworked temporal **s**ocial **s**cience experiments (henceforth referred to as NESS experiments or experimental loop) are widely used to study social behaviors [1–6] and group phenomena such as collective identity [6, 7], coordination [8], and diffusion and contagion [3, 6, 9]. There are several distinguishing features of NESS experiments. First, experiments and analyses are performed in a loop. Second, experiment subjects or participants interact through prescribed communication channels, where the players and interactions can be represented as nodes and edges, respectively, of networks. Third, experiments are carried out until a specified condition is met or for a particular amount of time (as opposed to one shot games). (Sometimes the term *game* is used in this work as a substitute for *experiment* because some experiments can be viewed as games, in the sense that human subjects are working to achieve some goal. However, we are *not* addressing gaming in this work.)

Besides carrying out NESS experiments, data analytics on experimental data and computational modeling of experiments are also very important. Analytics are required to interpret experimental results and modeling is useful in reasoning about and extending results from experiments [10, 11]. Combining experiments with modeling, in a repeated, iterative process, enables each to inform and guide the other [12–14]. This approach has been undertaken in several studies without automation [15–17] or purely conceptually [18]. Reference [18] takes a combined experiment/modeling approach by defining a framework for *conceptual* modeling for simulation-based serious gaming. Often, there is emphasis on one or the other (experiments or modeling) with no experiment-and-modeling iterations. That is, experiments are emphasized and there are no iterations [9], or modeling is emphasized and there are no iterations [19–21].

The simple idea of iterative experiments and modeling can be operationalized in various ways, including deductive and abductive analyses. In **deduction**, models are first developed, and predictions from them are then compared to subsequently-generated experimental data, in order to validate the models. In **abductive looping**, experiments are performed first, patterns are searched for in the experimental data, and this information is used to construct and modify models. Detailed abductive looping examples for the study of collective identity in the social sciences are provided in [7, 22]. Fig 1 provides one representation of the steps in abductive looping. Experiments are conducted; raw data are transformed into a common format (e.g., cleaned) for processing. Then experimental data are analyzed in different ways to understand player actions, identify patterns, and evaluate hypotheses. Models are developed based on these data, and model properties are inferred from the data. Models are executed and validated, and modeling results are compared against experimental data. Predictions may be made to explore counterfactuals. These latter results and the existing experimental data are used to determine conditions for the next experiments, if any, and the loop may repeat. See [7, 23, 24] for further discussion of abduction. We note that the steps in deduction are essentially the same, but the sequencing of experiments and modeling is reversed.

In this work, our focus is automating many steps in the NESS experiments. Automating these steps can lead not only to improved productivity, but also to improved scalability and reproducibility. (This has been the case in our research group.) It is seen that NESS experiments require several classes of operations: (1) experimental design, (2) experiment execution and data collection, (3) data fusion and integration, (4) experimental data analysis, (5) modeling, design, construction, and verification, (6) model parameters inference, (7) exercising models (e.g., simulations for agent-based modeling approaches), (8) comparisons of

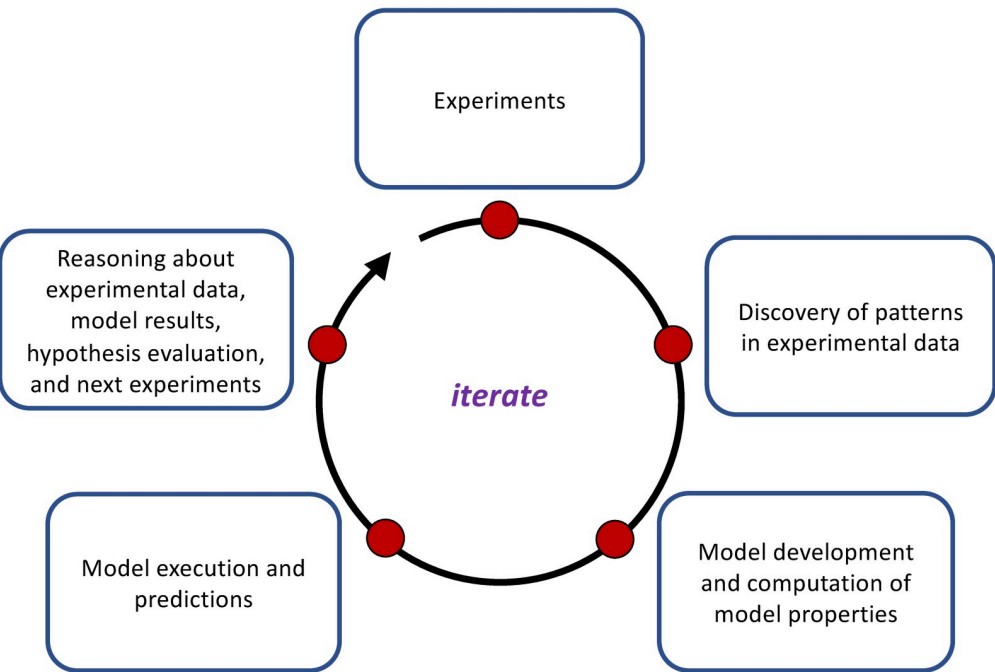

**Fig 1. A representation of the steps in iterative abductive analysis.** The process begins with conducting experiments and flows clockwise through reasoning about data and what experiments to perform next, whereupon the process repeats. Deductive analyses include these steps, but modeling occurs before experiments, so that the steps are rearranged. Parts of many of these steps (e.g., computing model properties) can be automated, and this automation is the focus of this paper. Other steps are not automated, such as the process of developing a model, because this requires a significant element of human reasoning. Thus, our software system requires human-in-the-loop execution. The process can be used in a purely experimental approach (i.e., no modeling). See the text for a description of this graphic.

experimental data against model output, (9) model executions beyond the ranges of experimental data (e.g., to explore counterfactuals), and (10) iteration on these steps.

However, current practice often entails producing custom programs and analytical scripts that pertain to the experiments and modeling. Our lab has found that this often leads to inefficiencies and duplication of effort. We propose a pipeline framework that automates many of the steps involved in analyzing social science experimental data, building models to capture the behavior of human subjects, and providing data to test hypotheses. The proposed pipeline framework is based on formal models, formal algorithms, and theoretical models. We also provide a data model such that if an experiment can be formally described in terms of this data model, then data from the experiment can be analyzed with our system. While there are software systems that address some of these operations [25, 26], they do not take the semantics of social experiments into account and largely focus on providing a generic data schema. It is important to note that our software system, presented in this work, is agnostic to deductive or abductive methodologies because our pipelines (described below) are composable. This composability also enables abduction using an experiment-only approach by removing the modeling activities in Fig 1.

## 1.2 Technical challenges of building software systems to analyze social science experiments

To realize an automated and extensible software system for NESS experiments, there are two major groups of technical challenges: those pertaining to pipelines in general, and those about

social sciences. Addressing the first group, abstractions that capture data analytics and computation are important [27]. High-level abstractions render a system more understandable and reusable [28]. General challenges include identifying appropriate levels of abstractions for tasks, pipelines, and systems. The problems of abstraction are important for automation, traceability, reproducibility, interoperability, composability, extensibility, and scalability [29]. Formal models help solve these abstraction problems [30].

In the case of the NESS system, there are three unique challenges to address. The first is specific to the features of NESS experiments. NESS experiments are often multi-phased, multi-subject, and multi-action, and hence are sophisticated. Each subject can take repeated actions from a set of action types, at any time and in any order. Interactions among subjects change the environment of a subject because they share resources. This is a far more complicated setup compared to many types of social science experiments such as one-shot games, experiments with a single type of action, and individualized experiments. Such experiments require more sophisticated software. Second, a greater range in modeling functionality is required, even for one class of problems. This is because a "model" in social sciences is often a qualitative textual description that is open to different interpretations due to lack of detail and due to uncertainty (e.g., in human behavior). Consequently, multiple interpretations of a textual description can result in different algorithmic models to build and evaluate. Third, experiments in the social sciences can vary widely, depending on the phenomena being studied [31]. Hence, data analytics for these varying experiments, including data exploration, requires custom analyses. These custom analyses can be addressed at the task level (i.e., new individual tasks within a pipeline), or at the pipeline level (i.e., the addition of new pipelines).

### 1.3 Solution approach and roadmap of work

To better present our work, Fig 2 provides a roadmap of this manuscript and the relationships among sections. Section 2 provides an overview of our solution approach, and specific contributions of the work. The data model (Section 3) is a formal specification of the features of experiments whose data can be analyzed with our system. If an experiment can be represented by this data model, then the experimental data can be analyzed with our pipelines. Graph dynamical systems (GDS) (Section 4) is a theoretical framework that we use for generating models of human behavior from experimental data. Both the data model and GDS are integral to the pipeline system software design and implementation (Section 7): the data model identifies the features of experiments and data that must be analyzed in the system, and GDS provides a formalism for model building. The pipeline system conceptual overview (Section 5) identifies the different components of the pipeline system. From this, the mathematical model for the pipeline system (framework and $h$-functions) in Section 6 is provided. This theoretical representation of the system is then used to specify the design of the system. That is, we have three theoretical models (in Sections 3, 4, and 6) that are the basis for software system design. This design, and implementation, of the pipelines are the subjects of Section 7. The implementation, along with the data model, are used in the case studies of Section 8.

## 2 Solution overview and contributions

### 2.1 Software pipelines

Our work is to provide an automated and extensible software system for evaluating social phenomena via iterative experiments and modeling. Fig 3 elaborates our solution: use of software pipelines to largely automate the process of analyzing social science experimental data, which are the classes of operations (3) through (10) given in Section 1.1. Table 1 supports the figure with overviews of the pipelines.

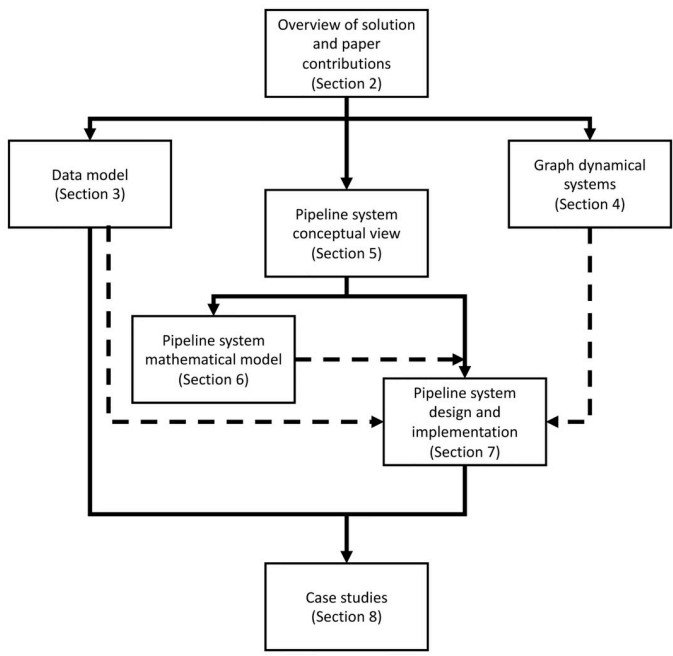

**Fig 2. Roadmap of, and relationships among, sections in this manuscript.** Arrows indicate dependencies among sections, and dashed arrows identify the theoretical models that impact the design and implement of the software pipeline system. The Introduction, Related Work, and Conclusions are not shown. See text for details.

A **pipeline** is a composition of tasks, where each **task** takes a set of inputs and produces a set of outputs. Our use of pipeline is motivated by the Pipes and Filters architecture pattern [32, 33]. A pipeline combines tasks in analyst-specified ways. We distinguish our work from *workflows* because, while there is much overlap between the capabilities of workflows and pipelines, here we do not address provenance of digital objects. Although the analysis loop in terms of experiments and modeling are presented in Fig 3, these analyses and abductive and

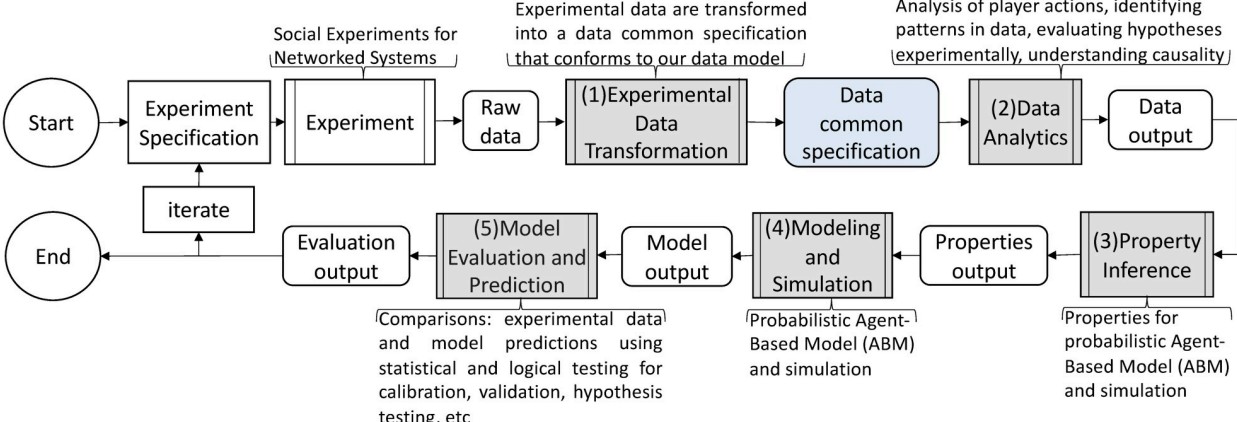

**Fig 3. Five software pipelines (in gray) for NESS experiments.** The five pipelines are itemized and described in Table 1. In this human-in-the-analysis loop, experiments (upper left in figure) are performed. Any experiment whose data can be cast in terms of the data model specification can be analyzed with this system. These pipelines are the focus of this work. The pipeline composition shown here, for abductive looping, is one of several possibilities. See Table 1 for descriptions of the pipelines in this figure. The first, second, and fifth pipelines can be used with a purely experimental approach (omitting modeling). An earlier version of the pipeline system is provided in [34], Fig 1.

**Table 1. Description of the five pipelines for NESS experiments.**

| No. | Acronym | Name | Description |
|---|---|---|---|
| (1) | EDTP | Experimental Data Transformation Pipeline | Experimental data are transformed, by the EDTP, into a data common specification that conforms to our data model (see Section 3). |
| (2) | DAP | Data Analytics Pipeline | The DAP analyzes data and generates and prepares data for property inference. |
| (3) | PIP | Property Inference Pipeline | The PIP determines properties for probabilistic agent-based modeling (ABM) and simulation (ABMS). |
| (4) | MASP | Modeling and Simulation Pipeline | Simulations are performed in the MASP. |
| (5) | MEAPP | Model Evaluation and Prediction Pipeline | The MEAPP generates comparisons between experimental data and model predictions using statistical and logical testing. This is part of model validation. We can then specify test conditions for next experiments (experiment specification). |

One composition of the pipelines is provided in Fig 3; it is one of several possibilities.

deductive looping can be executed within a study that exclusively uses experiments (i.e., no modeling). The importance of experiments, even with modeling, is observed in Fig 3 because experimental data plays a major role in pipelines 1, 2, 3, and 5. Experiments are critical, for example, in establishing causality, by comparing results from control experiments with those using treatments.

Our experimental data analysis and modeling software pipelines are complementary to current efforts to build configurable software platforms to perform social science experiments. See [35–38]. Usually, these systems only focus on the design and running of online lab experiments. Just as these experiment platforms provide the infrastructure for users to instantiate a particular *experiment* in software, we provide a pipeline framework that can be used to build pipelines for performing various types of *analyses* on the experimental data.

The focus of this paper is on formal theoretical models, and the architecture, design, implementation, and use of the pipelines that instantiate these models in software. The goal of the software system is to automate many of the steps in analyzing social science experimental data, and building and exercising models. We presume that in the great majority of cases, no one person is going to identify a social science problem or question; specify experiment requirements and design; build experimental platforms and execute experiments; specify analyses; build software to analyze experiments and perform data analyses; specify, design, build, and validate models of experiments; and evaluate hypotheses. Rather, we view these social science researches as "team science," and as such, this system is not focused on all members of such a team. So while all team members can have a general appreciation of the need for and value of such a system, the paper is focused on the team members who design and build software to automate many analysis steps.

The terms *experiment* to mean human subjects interacting in a controlled setting with their actions recorded. *Modeling* refers to building mathematical representations of experiments. *Simulation* is execution of software implementations of models, e.g., ABMs. We avoid ambiguous terms such as *computational experiment*. This paper is a full treatment of, and a significant extension of, a preliminary version (a conference paper) that appears as [34].

## 2.2 Novelty of work

There are three novel aspects of our proposed pipeline framework. First, we devise an **abstract data model** that is a representation of experiments and simulation models. One can rigorously determine whether experimental data and model outputs can be analyzed with our pipelines. Furthermore, we incorporate a second model called **graph dynamical systems** (GDS) [39]. GDS and the abstract data model provide foundations to ensure proper mappings, from

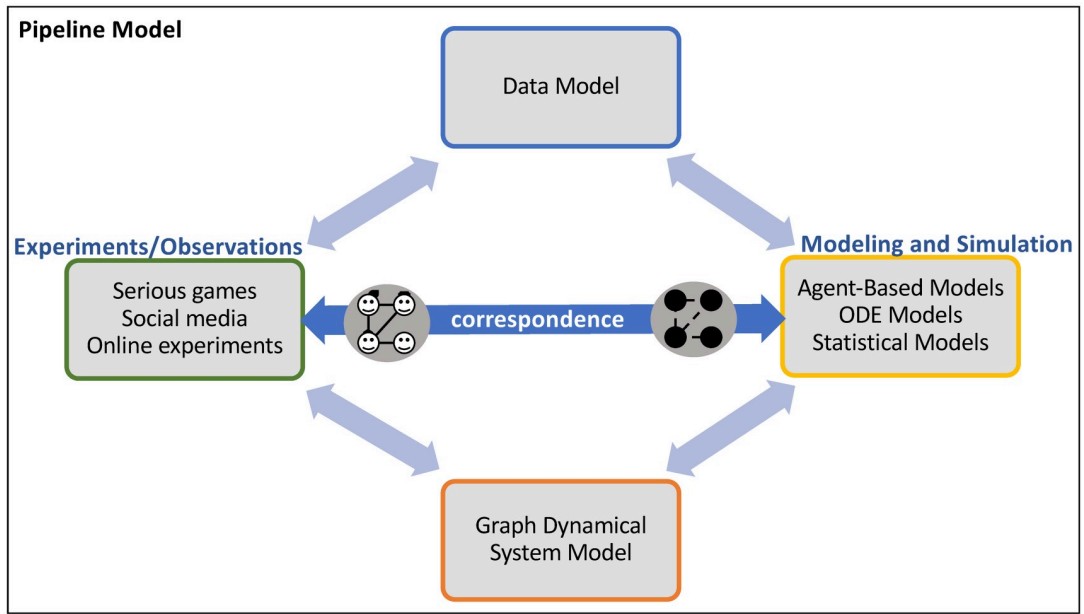

**Fig 4. The three types of models described in this work: (Abstract) data model, graph dynamical system model, and pipeline model.** The data model enables rigorous reasoning about both (*i*) experiments and experimental data specifications (requirements) and (*ii*) modeling and simulation (MAS) specifications. It, along with the graph dynamical system (GDS) model, help to ensure consistency and correspondence between experiments and MAS. We use GDS to model the dynamics of particular applications systems. Specific data sources and modeling approaches are shown. These are used within our pipeline model. Figure adapted from [34].

experimental conditions to computational model structure, and from model structure to experiments. See Fig 4, where we have an experimental platform and a modeling and simulation (MAS) platform, and we need these two to interoperate through our data and GDS models. It shows specific, illustrative types of data sources and modeling approaches.

Second, our pipeline framework is based on formal theoretical models; the three models that inform the pipelines are denoted by the dashed arrows in Fig 2. These models are crucial in providing a *principled* approach to software design and implementation. This is also useful for reasoning about abstractions. Third, our pipelines use a *microservices* conceptual approach [40–42] wherein the components (i.e., tasks) of a pipeline—which we call **functions**, **h-functions**, or **tasks**—have well-defined minimal scopes. (Functions are described below, but basically represent the software codes that provide the functionality that pipelines orchestrate.) This way, reuse is fostered because new functions can be added surgically for experiments, analyses, and models without introducing redundant capabilities. The pipeline framework can accommodate the insertion of new *h*-functions at arbitrary points in the pipeline.

In comparing our software system with others in the social science realm, we note that according to [28]: "the current focus of many social science systems is social network analysis." See other works in Section 9. As illustrated in Figs 1 and 3 our work goes far beyond structural analyses of static networks: our work centers on experiments of human behavior, where interactions among players are specified as edges in a network whose nodes are the players. Our system is used for quantifying the behavior of humans in experiments: (*i*) analyzing experimental data, (*ii*) developing models and their properties for the behavior of human subjects in these experiments, and (*iii*) conducting agent-based simulations to model these experiments, and conditions beyond those tested. Furthermore, the system is applicable to a wide range of

experiments, as long as they conform to the data model in Section 3. To our best knowledge, there are no other pipeline software systems for these types of studies.

## 2.3 Contributions

We itemize our contributions below.

**1. Development of formal models, formal algorithms, and software implementations for each of a data model and a pipeline model**. For each of data and pipeline representations (down left-hand column) of Table 2, we provide formal models, formal algorithms, and implementations. This approach demonstrates the power of modeling (including theory) to inform software system implementations. (Elements of Table 2 in blue and bold are our contributions; elements taken from other works are normal type-faced.) Thus, taking the data, GDS, and pipeline systems each in turn, this contribution is specifically that we provide a consistent (and unified) view of, and approach to, pipeline systems building for social experiments and for modeling them. Specific contributions within this context follow.

**2. Formal data model specification for NESS experiments and modeling**. We develop a **formal abstract data model** for NESS experiments. The primary use of our data model is this: any experiment that can be formally described in terms of this data model can be analyzed within our pipeline system. The model provides a single specification for both experiments and modeling, thus ensuring a correspondence between experiments and the modeling and simulation (MAS) tasks that represent the experiments. The abstract data model provides an abstraction level per Section 1.2. Characteristics of our data model are: (*i*) an experiment may contain one or more phases (i.e., sub-experiments); (*ii*) the finite duration of each phase may be different; (*iii*) the interaction structures among players (represented as networks) may be different for different phases; (*iv*) the set of player actions and the set of multi-player interactions may be different for different phases; and (*v*) players may repeat these actions and interactions any number of times, in any order, within a phase (i.e., temporal freedom of actions and interactions). A significant class of experiments is represented by these five characteristics. Illustrative works whose experiments are in this class are [1–6, 8, 9]. The data model, with our dynamical systems computational model (Section 4), provide a formal specification for experiments and models. The data common specification in Fig 3 is based on the data model.

**3. Formal pipeline framework**. We provide a conceptual view of pipelines used to construct a formal theoretical model of our pipeline framework. The **pipeline framework** is the infrastructure that executes common operations that are *invariant* across pipelines that have different functionality. (It is the same among all five pipelines that we introduce in this paper to study social science experiments and to model them.) These common fundamental

**Table 2. This work involves three major topics (left column of table): Data representation, modeling representation, and software pipelines.**

| Representation | (Theoretical) Models | Algoritms | Implementations |
|---|---|---|---|
| Data | Formal data model for networked experiments. | Entity-relationship diagram. | Use in multiple case studies. |
| Modeling | Use of existing Graph Dynamical Systems model. | Models tailored for particular applications. | Use in data analysis and modeling within pipelines. |
| Software pipelines | Formal, general model of pipeline framework. | Algorithm for execution of pipeline framework. | Five pipelines for data analysis and modeling. |

The first two enable developers to reason about construction of analysis pipelines; they also enable formal specification of experiments and of simulation systems. For each of these topics, there are models, algorithms, and implementations (labels across the top of the table). Our work covers all of these areas. The seven blue bold entries in the body of the table are our contributions. The other two non-bold entries are results taken from other works, but their use here is novel. Except for these two entries, all other elements in this table are contributions of our work. These contributions cover theory, implementation, and practice.

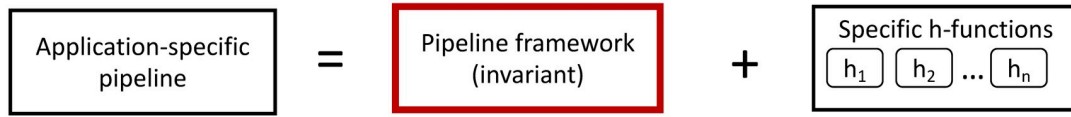

**Fig 5. An application-specific pipeline is composed of an invariant framework that performs general operations (see text) and application-specific *h*-functions.**

operations are: (*i*) read and parse the pipeline configuration file which specifies the pipeline tasks to complete; (*ii*) control accessing of input files, JSON schema files, transformation codes, functions, etc.; (*iii*) check files against their JSON schema and terminate gracefully if a verification fails; (*iv*) invoke the proper transformation functions (if applicable); (*v*) invoke the proper *h*-functions (see Contribution 4 below for *h*-functions) in their proper order (and any other operations); and (*vi*) error handling. See Fig 5. From the model, we present an algorithm that covers these operations, and then design and construct a pipeline framework to execute these operations for any pipeline. The framework is extensible to additional pipelines: we have demonstrated in our work that it is extensible because our particular pipelines have been constructed over time using the same framework.

**4. Pipeline *h*-functions (also called functions and tasks)**. We use a microservices conceptual approach [40–42] for our pipelines, wherein the tasks or components in a pipeline—which we call *functions* or *h*-functions—have minimalist scopes. The *h*-functions are software components that give a pipeline its application domain functionality. For example, one *h*-function will perform a particular data analytics operation, such as compute time histories, or compute a particular property for a particular model from data. We provide 29 implemented *h*-functions within the five pipelines (see Appendix D). All *h*-functions are serial codes written in C++, Python, and R. New functions can be introduced for new experiments, analyses, and models in a targeted fashion (as we have done), fostering reuse without redundancy. Note that a pipeline is comprised of the pipeline framework and a sequence of *h*-functions (Fig 5). We put these parts together to form particular pipelines in the next contribution.

**5. Five extensible pipelines for modeling and simulation, and analysis, of controlled networked experiments**. We design and construct pipelines for (1) transforming experimental data, (2) analysis of data, (3) inferring model properties, (4) MAS, and (5) comparing model results with experiments results, and predicting results in the absence of data (i.e., counterfactuals). Each pipeline consists of an extensible collection of *functions* that can be composed to accomplish particular objectives. Moreover, there are several ways to order these pipelines (Fig 3 is one way), and some pipelines may be omitted or implemented as multiple instances. An example is the use of experiments only for devising and testing hypotheses (i.e., studying a phenomenon with experiments, without modeling). Across multiple iterations of Fig 3, the experiment may change, necessitating different Data Analytics Pipelines for different experiments. Execution of pipelines and tasks are robust because of syntactic data validation of inputs and outputs at the task (function) level. These pipelines execute operations (3) through (10) in Section 1.1 (note: we do not automate the process of generating software verification cases, and model design is a human task). The Fig 3 caption explains why we emphasize *controlled* experiments; however, this is not a requirement for the pipelines (e.g., they can be used with social media or other types of observational data). The automated steps in Fig 3 are executed with a human-in-the-loop to inspect results. The pipelines also help ensure extensibility, scalability, and other "ilities" of Section 1.2.

**6. Case studies**. Use of the NESS system is demonstrated with three case studies. Case study 1 combines experiments and modeling. Case study 2 addresses experiments only. Case study 3

focuses on modeling only. In case study 1, we describe social experiments to generate collective identity (CI) within a collection of individuals [7]. Collective identity (CI) is an individual's cognitive, moral, and emotional connection with a broader community, category, practice, or institution [43]. Experiments and all five pipelines in Fig 3 are used. Two additional case studies use published works from other teams, appearing as [3, 44]. The point of these case studies is to demonstrate that our pipelines are useful for other types of experiments, and can be used in other settings.

**Empirical context for our pipelines**. The works of [7, 22, 45] demonstrate the usefulness of our pipeline system, where collective identity was studied via online experiments and modeling of them. That is, these provide empirical context where our software tools are important. Analogous works that also provide context are experiments in [1–6]. Returning to [7, 22, 45], these works demonstrated that CI could be formed among players in a group anagram game, where multiple players interact with their assigned neighbors to form words from collections of letters. Devised and implemented in the software, games were played online, through players' web browsers. Game data were analyzed to understand game dynamics, to develop a model of player behaviors in the game, and to compute properties for the model. The work [45] produced additional models for the individual actions of players (word formation, letter requests of game neighbors, and replies of letters to neighbors' letter requests) in the anagram game. Although all three of the works [7, 22, 45] used the software pipelines of this work, there is no mention nor description of the software pipelines in them. The purpose of our work is to describe the software pipeline system for general NESS experiments. That is, our pipeline software system is far more general than its use in those works. Nonetheless, those works demonstrate the value of our pipeline system.

## 2.4 Significant work beyond the conference paper

A preliminary 12-page version of this paper was published as [34]. Significant extensions of that work, presented herein, are summarized as follows. (1) In Section 3, we demonstrate how our abstract data model can be transformed into data models used in software development, such as an entity-relationship diagram in unified modeling language (UML) format. This enables reasoning about and representing the data model as a software artifact. (2) In Section 4, the graph dynamical systems (GDS) framework is presented in more detail and an example is given that uses the model. This makes more precise the GDS framework and its correspondence with the data model. (3) In Section 6, we provide a formal mathematical model of the pipeline system; we provide an algorithm of its functionality; and we describe how the model maps onto software. This is important because the formal model is the basis for the architecture and design of the pipeline system. (4) In Section 7 and Appendices A through D, we provide a greatly expanded description of the software design and implementation. This also demonstrates how the model of Section 6 is used to design and implement the software pipelines.

## 3 Abstract data model for NESS experiments and for modeling and simulation

We present a formal abstract data model. The utility of this model is to determine whether an experiment can be analyzed with our pipeline system. If an experiment can be represented by the characteristics of our data model, then data from the experiment can be analyzed with our pipelines. We provide a short example of its use, and then we demonstrate how the data model can be transformed into an entity-relationship diagram that is a more typical representation for reasoning about software, for implementation purposes.

## 3.1 Formal data model

A general adaptive abstract data model is presented. This data model for networked social experiments follows the five characteristics of Section 2.3, Contribution 2. The purpose of the data model, provided in Table 3, together with the computational model of Section 4 and the pipeline model in section 6, is to provide formal representations for experiments and MAS, and their iterative interactions, per Fig 4. We focus only on the data model, and for compactness, we describe the data model in terms of experiments, but the description is equally valid for modeling. Given a description of an experiment or model, one can determine whether our system of five pipelines can be applied. Also, given a phenomenon to study, the data model can be used to formulate experiments and models for simulating experiments. The data model produces the "data common specification" in Fig 3 (blue). We note that even for different types of experiments that do not conform to our data model, a pipeline system of collections

**Table 3. Definition of our abstract data model.**

| # | Parameters | Symbols | Description |
|---|---|---|---|
| **Experiment Schema** | | | |
| 1 | Experiment id | $exp\_id$ | Unique ID (identifier) for an experiment. |
| 2 | Number of phases | $n_p$ | Number of phases in the experiment. |
| 3 | Number of players | $n$ | The number of unique players over all phases in the experiment. |
| 4 | Begin time | $t\_begin$ | Timestamp of experiment start time. |
| 5 | End time | $t\_end$ | Timestamp of experiment end time. |
| 6 | Set of player IDs | $V$ | $V = \{v_1, \ldots, v_n\}$. Set of players over all phases; $v_i \in V$ is a unique ID for player. |
| 7 | Player attributes | $\Omega$ | $\Omega = \cup_{j=1}^n \Omega_j$. $\Omega_j = (\omega_{j1}, \omega_{j2}, \ldots, \omega_{j,n_{sa}})$ is the sequence of $n_{sa}$ attributes for $v_j \in V$. |
| **Phase Schema *Structure*** | | | |
| 1 | Phase schema id | $ph\_sch\_id$ | Unique id for phase schema. |
| 2 | Sequence | $i_{n_p}$ | $1 \le i_{n_p} \le n_p$. Element of the sequence of phases of the experiment. |
| 3 | Phase begin | $t\_ph\_begin$ | Timestamp of beginning of a phase. |
| 4 | Phase duration | $t_p$ | Number of time increments in a phase. |
| 5 | Unit of time | $u_p$ | Time unit of one time increment (e.g., seconds, days). |
| 6 | Network definition | $G(V', E')$ | Node set $V' = \{v_1, \ldots, v_\eta\}$ and edge set $E' = \{e_1, \ldots, e_m\}$, where $V' \subseteq V$ may not be all nodes (players) in the system, and edge $e_i = \{v_j, v_\ell\}$ with $v_j, v_\ell \in V'$. Note that $E'$ may be empty. |
| 7 | Meaning of an edge. | $\Lambda$ | Set $\Lambda$ of string representations $\lambda \in \Lambda$ stating the meaning(s) of an edge (e.g., $\lambda$ = "communication channel" or "influence"). |
| 8 | Node attributes for a phase. | $\Gamma$ | $\Gamma = \cup_{t=0}^{t_p}(\cup_{j=1}^\eta \Gamma_j(t))$. $\Gamma_j(t) = (\gamma_{j1}(t), \gamma_{j2}(t), \ldots, \gamma_{j,\eta_v}(t))$ is the sequence of $\eta_v$ attributes for $v_j \in V'$ in the phase $i_{n_p}$ at time $t$. $\Gamma$ is a triple nested sequence in attributes, player ID, and time. |
| 9 | Edge attributes for a phase. | $\Psi$ | $\Psi = \cup_{t=0}^{t_p}(\cup_{j=1}^m \Psi_j(t))$. $\Psi_j(t) = (\psi_{j1}(t), \psi_{j2}(t), \ldots, \psi_{j,\eta_e}(t))$ is the sequence of $\eta_e$ attributes for $e_j \in E'$ in the phase $i_{n_p}$ at time $t$. $\Psi$ is a triple nested sequence in attributes, edge ID, and time. |
| 10 | Initial conditions for nodes | $B^v$ | Nodes: $B^v = \cup_{j=1}^\eta B_j^v$. $B_j^v = (b_{j1}, b_{j2}, \ldots, b_{j,\mu_v})$ is the sequence of $\mu_v$ initial conditions for the phase, for $v_j \in V'$; $\mu_v \ge 0$. |
| 11 | Initial conditions for edges | $B^e$ | Edges: $B^e = \cup_{j=1}^m B_j^e$. $B_j^e = (\beta_{j1}, \beta_{j2}, \ldots, \beta_{j,\mu_e})$ is the sequence of $\mu_e$ initial conditions for the phase, for $e_j \in E'$; $\mu_e \ge 0$. |
| 12 | Action set | $A$ | $A = \{a_1, a_2, \ldots, a_{n_a}\}$. Set of $n_a$ actions that each player can execute, over time, any number of times, during a phase, where $n_a \ge 0$. |
| 13 | Action sequence | $T$ | $T = \cup_{t=0}^{t_p}(\cup_{k=1}^\eta T_k)$. $T_k = (\sigma_i, a_j, v_k, v_\ell, t_o, py_q)$ is the schema for an *action tuple*. $\sigma_i$ is a string that is a unique identifier for an action sequence. Action $a_j \in A$ is initiated by node $v_k \in V'$, and $v_\ell$ is the target node of the action, with edge $e = \{v_k, v_\ell\} \in E'$. $t_o \in \mathbb{R}$ is the time of the action ($0 \le t_o \le t_p$); $py_q$ is the payload represented as a JSON schema. |

The experiment schema describes experiment parameters. The phase schema structure describes parameter types for an experimental phase; an experiment can have any number $n_p$ of phases. Particular instance variables within the phase schema structure can vary across phases. We use *experiment* throughout in the table and text for ease of exposition, but the data model is also used for *(simulation) models*.

of operations can still be built, but would have different *h*-functions than those we have constructed. We now describe the two major sections of Table 3.

**Experiment schema**. Per Table 3, an experiment has the following parameters: a unique ID *exp_id*, the number $n_p$ of experiment phases, the number $n$ of players (i.e., human subjects) over all phases of the experiment, a *t_begin* timestamp for the start of the experiment, and a *t_end* timestamp for the end of the experiment. Each player has a (universally) unique ID $v_i$ for identification. A set *V* of players in an experiment is defined by $V = \{v_1, \ldots, v_n\}$. An experiment has $n_{sa}$ attributes defined for each player. Player attributes $\Omega$ are invariant across phases (e.g., age, gender, education level, and income that might be obtained through a questionaire).

**Phase schema**. An experiment is composed of one or more phases. All phases have a common schema, per Table 3, but particular phases may have different variable values for parameters in the schema.

Each phase schema has the following parameters: a unique ID *ph_sch_id*, the number $i_{n_p} (1 \leq i_{n_p} \leq n_p)$ of the phase in the sequence of phases, a *t_ph_begin* timestamp for the start time of the phase, number of time increments in the phase $t_p$, and the unit $u_p$ of time of one time increment. The interaction channels of pairwise interactions among players is defined by a network $G(V', E')$, with meanings of edges $\Lambda$, for each phase. Edge attributes $\Psi$ and node attributes $\Gamma$ over all edges and nodes capture time-varying attribute changes for phase $i_{n_p}$. Players (i.e., nodes) and edges may have initial conditions $B^v$ and $B^e$, respectively, whose elements may be the same as $\Gamma$ and $\Psi$. The permissible player actions during a phase is denoted as the set *A*. An action tuple $T_i$, which captures pair-wise interactions between players, may be intimately tied to the attribute sequences $\Gamma$ and $\Psi$ of a phase because action tuples, for example, may cause or be caused by changes in node and edge attributes. In essence, $\Gamma$ and $\Psi$ can be viewed as sequences of node and edge states. Items 8 through 11 and 13 of the phase schema in Table 3 follow the same basic pattern, to capture features by node or edge, and by time. There is a sequence of values for a particular node $v_j$ or edge $e_j$ (e.g., $\Gamma_j$, $\Psi_j$, $B_j^v$, $B_j^e$, and $T_j$). Each entry in a sequence can be a scalar, array, set, map, or other structure. Then, these entries are sequenced over time through the union of entries over time, from time 0 through $t_p$, as shown in rows 8, 9, and 13 of Table 3. The exceptions are the initial conditions $B_j^v$ and $B_j^e$ (rows 10 and 11), because by definition, they are specified only at time 0.

## 3.2 Illustrative instances of data model parameters

We provide a few illustrative examples of data model elements. A 3-phase game is described in Section 8, Case Study 1. Phase 2 is a group anagram (word construction) game. In phase 2, a network $G(V', E')$ is imposed on the players, where the meaning $\lambda$ of a edge is a communication channel to request letters and reply to requests. A node initial condition $b_{j1}$ for a game is the number of alphabet letters a player receives at the beginning of the phase to use in forming words, and $b_{j2}$ is the set of letters. Each player can execute any action from the action set *A*, such as request a letter from a neighbor.

We now provide an example of an action tuple of an action sequence. If player $v_i$ requests letter "z" (a request is action $a_\ell \in A$) from player $v_j$ at time $t_o$, which initiates a sequence of actions (because there may be a subsequent letter reply from $v_j$) then the action tuple is $T_i = (\sigma_i, a_\ell, v_i, v_j, t_o, \text{"z"})$. Here, $\sigma_i = v_i + \text{"−"} + counter$ (e.g., a string) is a concatenation of the initiator's ($v_i$'s) ID with a player-specific counter to form a unique ID for the sequence of actions that is initiated with the letter request. If $v_j$ responds with "z," then this (second) action tuple will use the same $\sigma_i$ as the first element of the tuple, consistent with $T_i$. This is how action tuples are defined and identified in data processing, in forming action sequences *T* for a phase.

### 3.3 From abstract data model to software specification

Ours is an abstract mathematical data model. There are several reasons for our choice of model representation. First, a mathematical representation is more abstract (which means, among other things, more versatile and flexible) in its use. Second, it corresponds much more closely to the information required for pipeline capabilities, and enables compact representations of simulation models. Third, it is naturally amenable to translation into other data model representations that are more common in software. We elaborate on each of these.

**1. Abstract representation**. An element of a sequence can *abstractly* represent any type of data, including scalars, vectors, sets, tensors, and complicated data structures (that may be implemented via a JSON schema). For example, consider $\gamma_{j2}$ of $\Gamma_j$ of $\Gamma$ in Table 3, which is an attribute for node or player $v_j \in V'$. This variable might represent a 2-D matrix or a set. Furthermore, if the representation needs to be changed, it is much easier to do so with an abstract representation.

**2. Compactness**. Consider a capability for a simulation model, as part of a pipeline: multiplying two matrices, $M_1$ and $M_2$. A mathematical representation is simply $M_1 \cdot M_2$ or $M_1 M_2$. A pseudo code representation for this functionality would require some five lines of code including three FOR loops. Clearly, $M_1 \cdot M_2$ is far more compact.

**3. Principled transitions (progression) among software artifacts**. The steps in progressing from a mathematical data model to a software model are shown in Fig 6. Experiment and phase schemas in Table 3 contain data structures. Instances of our abstract data model (generated from the execution of an experiment) can be represented as entity-relationship diagrams, which are *conceptual* or *logical* data models. Examples are relational models [46], object-oriented models like Object Definition Language (ODL) [47] or Unified Modeling Language (UML) [48], or data structure diagrams [49], among others. A UML representation of an entity-relationship diagram for our abstract data model is presented in Fig 7. UML is the

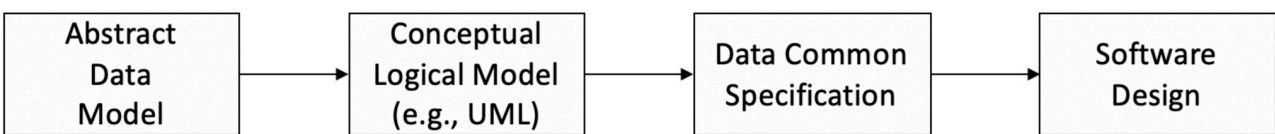

**Fig 6. Sequence of data models for reasoning about experiments and modeling and simulation.** We advocate for pre-pending the abstract data model to the front end of the model process, as shown here. Table 3 shows our abstract data model and Fig 7 shows this data model translated into a entity-relationship diagram in unified modeling language (UML) form. The table and figures in A (which support Section 7) show the Data Common Specification for our software design.

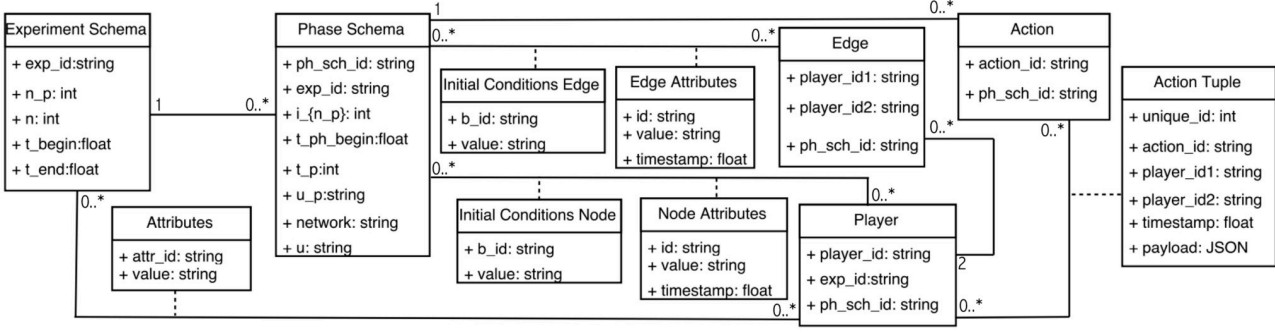

**Fig 7. Data model of Table 3 translated into a entity-relationship diagram in unified modeling language (UML) form.** This illustrates that the abstract data model can be translated to customary forms of data models (e.g., UML) that are more amenable for software development.

industry-standard language for specifying, visualizing, constructing, and documenting the artifacts of software systems [48]. All of the structures from the abstract data model of Table 3 are translated into a entity-relationship diagram in unified modeling language (UML) form, demonstrating that the abstract data model can be translated into standard forms of data models more amenable for software development.

*Data common specification.* Every JSON input file in the pipelines needs a corresponding JSON schema for the verification of formats. For our Data Common Specification there are five classes of input every experiment needs to define. The formal data model in Section 3.1 specifies that an experiment can have any number $n_p$ of phases and a different set of players with an action set for each phase. Table 8 in Appendix A shows a description of the elements of the Data Common Specification. Figures in Appendix A define through JSON schemas the formats and compositions of the elements of the Data Common Specification. These are implementation aspects of our pipelines. These are also the types of files we use in the case studies in Section 8.

## 4 Graph dynamical system model

In this section, we present a formal framework for NESS experiments and Agent-Based Models. We use a computational model known as a discrete **graph dynamical system** (GDS) [39], to specify, build, and execute experiments and simulators of experiments (and of other conditions). GDS is also correspondent with the data model of Section 3 and is a general model of computation [50, 51], and hence can ensure that experiments and models are synchronized, per Fig 4. A number of other formal models could have been used; we find GDS to be a natural model for specifying NESS experiments. Table 4 shows a description of all the symbols used in our equations.

**Table 4. Symbols used to describe our computational model known as a discrete Graph Dynamical System (GDS).**

| # | Parameters | Symbols | Description |
|---|---|---|---|
| **Experiment Schema** | | | |
| 1 | GDS | $S$ | A synchronous Graph Dynamical System (GDS). |
| 2 | Node set | $V$ | $V = \{v_1, -, v_n\}$; $v_i \in V$ is a unique ID for a node. |
| 3 | Edge set | $E$ | $E = \{e_1, -, e_m\}$; $e_i \in E$ is a unique ID for an edge. Each undirected edge $\{v_i, v_j\} \in E$ with $v_i, v_j \in V$ can be represented by two directed edges: edge from $v_i$ to $v_j$, denoted $e_{ij} = (v_i, v_j)$ and $e_{ji} = (v_j, v_i)$. |
| 4 | Network definition | $G(V, E)$ | Node set $V$ and edge set $E$. |
| 5 | Undirected graph | $G$ | $G \equiv G(V, E)$ is an undirected graph with $n = |V|$, and represents the underlying graph of the GDS, with node set $V$ and edge set $E$. |
| 6 | State space | $W$ | The union of the state space $W^v$ for nodes and the state space $W^e$ for edges; $W = W^v \cup W^e$. We assume here that only nodes have states; there are no edge states. |
| 7 | Function | $F$ | Collection of functions in the system, $F = (f_1, f_2, \ldots, f_n)$. Function $f_i$, represents the local function associated with node $v_i$, $1 \leq i \leq n$, that describes how $v_i$ updates its state. We use the synchronous update scheme where all $f_i$ execute in parallel. |
| 8 | Method | $U$ | Describes how the local functions are ordered at each discrete time. |
| 9 | Sequence of vertices | $N(v_i)$ | The sequence of vertices adjacent to $v_i$ in $G$, including $v_i$ itself, so that $1 \leq |N(v_i)| \leq n$ for each $v_i \in V$. |
| 10 | Degree | $d(v_i)$ | The degree of $v_i$ in $G$. |
| 11 | Sequence of vertex states | $s(v_i)$ | The sequence of vertex states of the vertices in $N(v_i)$ so that $1 \leq |s(v_i)| \leq n$ for each $v_i \in V$. |
| 12 | System state or configuration | $C$ | The system state or configuration $C$ of a GDS is the vector of length $n$, $C = (s_1, s_2, \ldots, s_n)$. |

## 4.1 GDS formal model

A synchronous **Graph Dynamical System** (GDS) [52] $S$ is specified as $S = (G, W, F, U)$, where we define each in the following. (*a*) $G \equiv G(V, E)$ is an undirected graph with $n = |V|$, and represents the underlying graph of the GDS, with node set $V$ and edge set $E$. Nodes represent agents in a system or test subjects in our experiments, and edges denote pair-wise interactions between agents. (*b*) $W$ is the state space, which is the union of the state space $W^v$ for nodes and the state space $W^e$ for edges; i.e., $W = W^v \cup W^e$. These are the states that nodes and edges can take during the dynamics. Each undirected edge $\{v_i, v_j\} \in E$, with $v_i, v_j \in V$, can be represented by two directed edges: edge from $v_i$ to $v_j$, denoted $e_{ij} = (v_i, v_j)$, and $e_{ji} = (v_j, v_i)$. (*c*) $F = (f_1, f_2, \ldots, f_n)$ is a collection of functions in the system. Function $f_i$ represents the **local function** associated with node $v_i$, $1 \leq i \leq n$, that describes how $v_i$ updates its state. (*d*) $U$ is the method which describes how the local functions are ordered at each discrete time. Here, we use the synchronous update scheme where all $f_i$ execute in parallel.

Each node $v_i \in V$ of $G$ has a state value from $W^v$ at each time $t$. Each edge $e_{ij} \in E$ of $G$ has a state value from $W^e$ at each $t$. Each function $f_i$ specifies the local interaction between node $v_i$ and its neighbors in $G$. The inputs to function $f_i$ are the state of $v_i$, the states of the neighbors of $v_i$, and the states of the edges outgoing from $v_i$ in $G$. Function $f_i$ maps each combination of inputs to $s_i' \in W^v$ for $v_i$, and to $s_{ij}' \in W^e$ for each directed edge $e_{ij}$. $s_i'$ is the next state of node $v_i$, and $s_{ij}'$ is the next state of edge $e_{ij}$. These functions are executed in parallel at each time step $t$.

We provide details of the dynamics of a GDS, based on the overview above. We assume here that only nodes have states; there are no edge states. Let $G(V, E)$ be a graph with node set $V$ and edge set $E$, and where $n = |V|$. Each node $v_i$ has a state $s_i$ Let $N(v_i)$ be the sequence of vertices adjacent to $v_i$ in $G$, including $v_i$ itself, so that $1 \leq |N(v_i)| \leq n$ for each $v_i \in V$. That is,

$$N(v_i) = \left( v_{v_i,1}, v_{v_i,2}, \ldots, v_{v_i,d(v_i)+1} \right), \tag{1}$$

where $d(v_i)$ is the degree of $v_i$ in $G$. Let $s(v_i)$ be the sequence of vertex states of the vertices in $N(v_i)$, so that $1 \leq |s(v_i)| \leq n$ for each $v_i \in V$, i.e., and $d(v_i) = |N(v_i)| - 1$.

$$s(v_i) = \left( s_{v_i,1}, s_{v_i,2}, \ldots, s_{v_i,d(v_i)+1} \right). \tag{2}$$

We call $s(v_i)$ the *restricted state* of $v_i$. The **system state** or **configuration** $C$ of a GDS is the vector of length $n$, $C = (s_1, s_2, \ldots, s_n)$.

A local function $f_i : (W^v)^{d(v_i)+1} \rightarrow W^v$ quantifies the dynamics of node $v_i$ by computing $v_i$'s next state $s_i'$ using the states of nodes in its closed 1-neighborhood as

$$s_i' = f_i(s(v_i)). \tag{3}$$

Updating the entire set of nodes in $G$ at some time $t$ is accomplished with the **GDS mapping**

$$\mathbf{F} : (W^v)^n \rightarrow (W^v)^n. \tag{4}$$

For the *synchronous* update scheme, where all $f_i$, $i \in \{1, 2, \ldots, n\}$, execute in parallel, the GDS mapping is defined by

$$\mathbf{F}(s_1, s_2, \ldots, s_n) = (f_1(s(v_1)), f_2(s(v_2)), \ldots, f_n(s(v_n))). \tag{5}$$

In a simulation, we compute successive system states using this last equation, as $C(t + 1) = \mathbf{F}(C(t))$, where $C(t)$ is the system state or configuration at time $t$, and $C(t + 1)$ is the next system state.

To make this explicit, we now cast the preceding formalism into a pseudo-algorithm in computing the dynamics of a GDS. Let us assume for simplicity that only nodes possess state, and edges do not. At any time $t$, the **configuration** $C(t)$ of a GDS is $C(t) = (s_1^t, s_2^t, \ldots, s_n^t)$, where $s_i^t \in W^v$ is the state of node $v_i$ at time $t$ ($1 \leq i \leq n$). In a synchronous GDS, all nodes compute and update their next state *synchronously*, i.e., in parallel. A GDS transition from one configuration $C(t)$ to a next configuration $C(t + 1)$ in parallel at each time $t$ can be expressed as follows,

**for** each node $v_i \in V$ **do in parallel**

 (i). Compute the value of $f_i$ (Eq (3)) using states in $C(t)$ and assign it to $s_i'$.

 (ii). Assign $s_i'$ as the next state of $v_i$ in $C(t + 1)$.

**end for**

Note that if the $f_i$ are stochastic, $C(t + 1)$ may not be unique. The extension to the update of edge states $s_{ij}'$ is natural.

**Associations between the data model and GDS**. The data model in Section 3 is consistent with a GDS. The graph $G(V', E')$, per phase, in Table 3 corresponds to the graph $G(V, E)$ of the GDS. Node $W^v$ and edge $W^e$ state spaces in the model represent subsets of the node ($\Gamma$) and edge attributes ($\Psi$) in the data model, respectively. Attributes may have additional parameters that are not part of the node or edge state, such as gender and age. Action tuples may be part of the state. The sequencing of action tuples is related to the update scheme $U$, e.g., whether each node takes turns performing some action in series, or whether players can act simultaneously.

## 4.2 Example GDS and resulting dynamics: Threshold systems

We provide an example of a GDS and the dynamics that it generates. We use a threshold contagion system, motivated by the work [3, 13, 53] in the social sciences. Also, we use this model in the second case study of Section 8. A progressive threshold system works as follows. The network $G(V, E)$ is provided at the left in Fig 8. The valid state set $W$ for a node is $W = W^v = \{0, 1\}$, where state 0 means that a node does not possess a contagion and state 1 means that a node possesses the contagion and will assist in transmitting it. The threshold local function works as follows. Each node $v_i$ is assigned a threshold $0 \leq \theta_i \leq d_i + 1$, where $d_i$ is the degree of $v_i$ in $G$. If the state $s_i$ of node $v_i$ at time $t$ is 1 (i.e., $s_i^t = 1$), then the output of $f_i$ is 1 (that is, a node in state 1 at $t$ remains in state 1 at $(t + 1)$). If $s_i^t = 0$, then $s_i^{t+1} = f_i = 1$ if at least $\theta_i$ of $v_i$'s neighbors are in state 1 at $t$; otherwise, $s_i^{t+1} = f_i = 0$. That is,

$$s_i^{t+1} = f_i(s^t(v_i)) = \begin{cases} 1 & \text{if } s_i^t = 1, \\ 1 & \text{if } s_i^t = 0 \text{ and } n_1 \geq \theta_i, \text{ or} \\ 0 & \text{otherwise} \end{cases} \tag{6}$$

where $s^t(v_i)$ is the sequence of states in the closed neighborhood of $v_i$ at time $t$, and $n_1$ is the number of nodes in state 1 in $s^t(v_i)$. This is a deterministic GDS.

The dynamics evolve as follows; see Fig 8. We specify as initial conditions that $v_1$ has the contagion at $t = 0$, i.e., $s_1^0 = 1$; all other nodes do not have it. See $C(0)$ in Fig 8, where only $s_1^0 = 1$. At $t = 1$, $s_2^1 = f_2(s_1^0, s_2^0, s_3^0) = 1$ because $\theta_2 = 1$ and $s_1^0 = 1$, and $s_2^0 = s_3^0 = 0$. So, the threshold for $v_2$ is just met by $v_1$. For the same reason, $s_5^1 = f_5(s_1^0, s_4^0, s_5^0, s_6^0) = 1$ (because $s_1^0 = 1$; $v_5$ and all other neighbors of $v_5$ are in state 0). No other node will change state at $t = 1$ and therefore $C(1)$ has three nodes in state 1 at $t = 1$. At $t = 2$, $v_4$ will change state, even though its

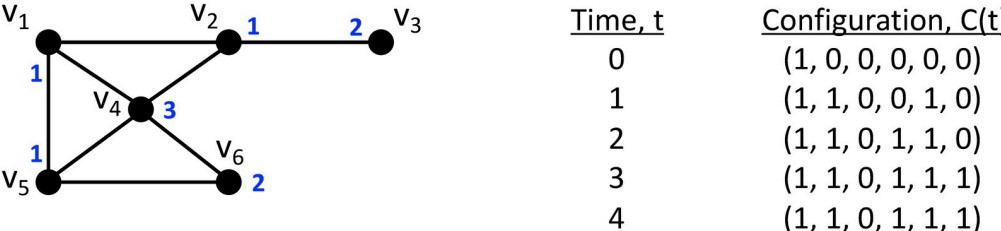

| Time, t | Configuration, C(t) |
|---------|---------------------|
| 0 | (1, 0, 0, 0, 0, 0) |
| 1 | (1, 1, 0, 0, 1, 0) |
| 2 | (1, 1, 0, 1, 1, 0) |
| 3 | (1, 1, 0, 1, 1, 1) |
| 4 | (1, 1, 0, 1, 1, 1) |

**Fig 8. Network $G(V, E)$ for a GDS example, with $V = \{v_1, v_2, v_3, v_4, v_5, v_6\}$.** Thresholds $\theta_i$ are provided for nodes $v_i$, in blue, by the respective nodes. The local functions $f_i$ are threshold functions for $v_i \in V$, $1 \leq i \leq 6$; see text for details. The discrete system dynamics are given by the configurations at successive times from 0 to 4, at the right in the figure. Each configuration is given by $C(t) = (s_1^t, s_2^t, s_3^t, s_4^t, s_5^t, s_6^t)$. The system reaches a fixed point at time $t = 3$, as evidenced by no change in the configuration in going from $t = 3$ to $t = 4$.

threshold is large ($\theta_4 = 3$) because three of $v_4$'s neighbors ($v_1$, $v_2$, and $v_5$) are now in state 1. This is the only node that changes state at $t = 2$ and so $C(2)$ is as shown in Fig 8. The same reasoning applies to the transitions of other node states. Note that $v_3$ will never transition because its threshold (2) is greater than the number of its neighbors (1). Also note that the system reaches a *fixed point* at $t = 3$ because no further state changes are possible.

## 5 Conceptual view of pipelines

The purpose of this section is to provide a high-level overview of the pipeline system. Pipeline composition, the pipeline framework, particular pipelines, and operations (*h*-functions) within pipelines are covered. This is useful for setting up formal theoretical model of Section 6 and the software implementation in Section 7.

### 5.1 Pipeline system

**Pipeline compositions**. Our system of five pipelines is shown in Fig 3. We separate the experimental platform from the pipelines so that the system can be used with different experimental software platforms, as long as an experiment conforms to the Data Common Specification, which is the data model of Section 3. An iteration of the loop may use any number of the five pipelines, and any number of functions within them, for flexible composability, consistent with data dependencies [54].

 **Pipeline framework**. Fig 9 provides a high level conceptual view of a pipeline. Specifics of *h*-functions are addressed in the next subsection. Here, our point is to emphasize the bounding

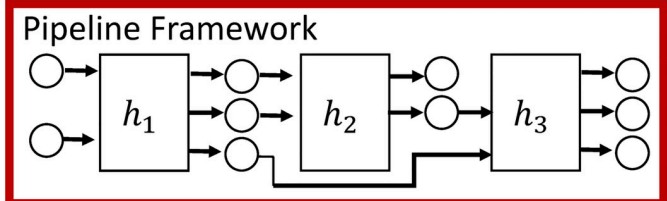

**Fig 9. Conceptual view of a pipeline that is composed of the pipeline framework (represented by the bounding box) and the *h*-functions that provide the application-based functionality of a particular pipeline.** Functions, or *h*-function, $h_i$, $1 \leq i \leq 3$ are implemented as software within a pipeline. The pipeline framework (red box) controls the execution order of functions and the inputs and outputs for each function, through a pipeline job specification. Circles in the figure denote input and output digital objects, such as ASCII files or database tables. This figure is a more detailed representation of Fig 4. Adapted from [34] Fig 3.

box around the pipeline in this figure, which represents the pipeline framework, i.e., the invariant part of pipelines that is used across all pipeline instances. The operations executed by the pipeline framework are listed in Section 2.3. It is the *h*-functions that tailor a pipeline for a given domain-based purpose.

**Pipelines**. The five pipelines of Fig 3 now described. (1) The Experimental Data Transformation Pipeline cleans the experimental data and transforms them into a data common specification. (2) The Data Analytics Pipeline analyzes temporal interactions among players to identify patterns in the data in order to understand human behavior and to assist in model development. Computational models are developed offline, as this is human reasoning-based effort. Thereafter, direct and derived data are used as input to the (3) the Property Inference Pipeline. This pipeline generates property values for parameters of simulation models, often by combining data from multiple experiments. Simulation models (e.g., ABMs) are built off-line and software implementations of these models are part of (4) the Modeling and Simulation Pipeline. This pipeline invokes the code to run simulations, using the generated property values, as well as network descriptions, initial conditions, and other inputs. Simulations may model completed or future experiments, or other scenarios beyond the scope of experiments. (5) The Model Evaluation and Prediction Pipeline compares multiple sets of data. As one case, experimental data and model predictions may be compared. As another case, results from two models may be compared. One objective may be to predict beyond experiment data (counterfactuals) and propose further investigations suggested by analysis findings.

Each pipeline is currently a sequential composition of functions. This composition is specified by an analyst through a job definition. Similarly, compositions of the pipelines of Fig 3 are specified by an analyst. The pipeline process takes care of file dependencies between functions. Also, it validates the input and output data of functions, described below. The structure of a pipeline is shown in Fig 9, where function $h_1$ takes two inputs and generates three outputs (two are inputs to function $h_2$ and one is an input to function $h_3$); function $h_2$ generates two outputs, one of which is an input to $h_3$. Note that the pipelines control execution of functionality. Execution control consists of a pipeline invoking functions sequentially, as illustrated in Fig 9. Additional details are in Sections 6 and 7. Other control structures are being added.

## 5.2 Functions within pipelines

Functions are designed as microservices—modular software with limited scope—within pipelines. They provide a range of capabilities from straight-forward plotting routines to data cleaning and organizing, storing and accessing data sets, inferring properties, and running simulations. Users may add other functions and continue community-based development. This concept is illustrated in Fig 9. Currently, inputs and outputs are files, but may include other digital objects, such as database table entries. Fig 10 drills down to show details for a function.

Fig 10 shows execution details associated with each function *h*. Input data $(\hat{q}_i)$ (e.g., in the form of an ASCII data file that may be raw data or output from a preceding function) may need to be transformed into formats required by *h*. This transformation is performed by transformation code $\tau_j$, which generates the input $(\hat{k})$ in the required format. These input objects $\hat{k}_1$ and $\hat{k}_2$ conform to JSON specifications to ensure compliance for inputs to *h*. The outputs of *h* are $\hat{\ell}_1$, $\hat{\ell}_2$, and $\hat{\ell}_3$.

**Microservices**. Our functions map directly to microservices. Appendix E addresses characteristics, benefits, and comparisons of microservices. We provide details of microservices because they are the fundamental execution units within our pipelines.

Pipeline

**Fig 10. One arbitrary software *h*-function within a pipeline.** Data instances $\hat{q}_1$, $\hat{q}_2$, and $\hat{q}_3$ are transformed by transformation code $\tau_1$ to conform to required input $\hat{k}_1$ for $h$. Similary, $\hat{q}_4$ and $\hat{q}_5$ are used by $\tau_2$ to produce input $\hat{k}_2$. Outputs from the $h$-function are $\hat{\ell}_1$, $\hat{\ell}_2$, and $\hat{\ell}_3$. Inputs and outputs are subjected to verification through comparisons with specified schema (not shown here). The pipeline framework is represented by the red box that controls execution of the h-functions and transformation codes. This is a more detailed representation of Figs 5 and 9.

## 6 Formal pipeline framework model

With the conceptual view in Section 5, we now provide a formal mathematical model for the pipeline *framework*, the invariant part of a pipeline, and *h*-functions, which are particular operations to perform on data (e.g., from experiments). First, we provide the theoretical model. Then, we provide an algorithm for its execution, which moves the system closer to the software and facilitates system design. In Section 7, we combine the pipeline framework with *h*-functions to produce particular pipelines; the emphasis there is on software design and implementation.

### 6.1 Pipeline framework model

Let $\mathcal{P}$ be a **collection of pipelines**, with **pipeline** $P \in \mathcal{P}$ represented as $P(Q, \hat{Q}, S_{ID}, S, T_{ID}, T, H)$. Here, $Q$ is a set of datatypes $q \in Q$; $\hat{Q}$ is a set of all data instances $\hat{q} \in \hat{Q}$; $S_{ID}$ is a set of mappings $s_{ID} \in S_{ID}$ from datatypes to schema evaluators; $S$ is a set of schema evaluators $s \in S$; $T_{ID}$ is a set of mappings $\tau_{ID} \in T_{ID}$ from *h*-functions and datatypes to transformations; $T$ is a set of data transformations $\tau \in T$; and $H$ is a sequence of *h*-functions $h \in H$. We detail each of these in turn.

First we address the types of data that are inputs and outputs to *h*-functions. Let $q \in Q$ be a **datatype** of the set $Q$ of all datatypes. Let $k \in K$ be an **input datatype** of the set $K$ of all input datatypes. Let $\ell \in L$ be an **output datatype** of the set $L$ of all output datatypes. Datatypes can be primitive datatypes found in most programming languages (e.g, integer, float, real, char), and data structure types (e.g., records) that are combinations of primitive types and data structures such as maps and arrays. An element $q \in Q$ may be either or both an input data element $k$ and an output data element $\ell$; we have $Q = K \cup L$. Moreover, the intersection of $K$ and $L$ will almost always be non-empty, i.e., $K \cap L \neq \emptyset$, because in a pipeline, an output element of an *h*-function may be an input to a subsequent *h*-function. We use $k$ to denote an input datatype; we use $\ell$ to denote an output datatype; and we use $q$ to denote an input datatype, an output datatype, or both.

We have the **instance** analogs of the datatypes above. That is, instances have numerical values and character (strings) assigned for each datatype. **Data instances** $\hat{q} \in \hat{Q}$, **input data instances** $\hat{k} \in \hat{K}$, and **output data instances** $\hat{\ell} \in \hat{L}$, must conform to the datatypes of $Q$, $K$, and $L$, respectively. Note that there will be an implicit relationship between an instance $\hat{q}$ and a

datatype $q$ because these are based on the semantics of a problem. In general, the relationship between one $q$ and $\hat{q}$ is 1-to-many: there are many possible instances for a single datatype. Each data instance has as a parameter the datatype to which it must conform.

We now address data schema and data format verification. Let $S_{ID}$ be the set of **schema ID mappings** $s_{ID} \in S_{ID}$, where $s_{ID}:Q \rightarrow S$ is defined by a mapping from each datatype $q$ to a unique schema evaluator $s \in S$. That is,

$$s = s_{ID}(q) . \tag{7}$$

If we have a universal schema identifier, then $|S_{ID}| = 1$, i.e., a single $s_{ID}$ is used across all $q \in Q$. To verify that an instance $\hat{q}$ of a datatype $q$ has a valid format, we use a **schema evaluator** $s: \hat{Q} \rightarrow \{0, 1\}$. A schema evaluator takes as input a data instance $\hat{q}$ and outputs a 1 when $\hat{q}$ conforms to the datatype $q$ (i.e., $\hat{q}$ is successfully verified against $q$ using $s$), and outputs a 0 otherwise. That is, $s(\hat{q})$ returns a 0 or 1.

The next phase of the model addresses data transformations. Let $T_{ID}$ be the set of transformation ID mappings $\tau_{ID} \in T_{ID}$. A **transformation ID mapping** $\tau_{ID}: H \times K \rightarrow T$ is a mapping from a *target h-function* $h \in H$ and *target input* datatype $k \in K$ for the $h$-function, to a transformation function $\tau$. That is,

$$\tau = \tau_{ID}(h, k) . \tag{8}$$

Hence, there is one transformation function $\tau$ for each input datatype $k$ and instance $\hat{k}$, respectively) to an $h$. Without loss of generality, we have can have a universal transformation ID mapping $\tau_{ID}$ across the entire set of tuples $H \times K$, so that $|T_{ID}| = 1$.

The role of a data transformation function is to operate on inputs and outputs from one or more $h$-functions (defined below) and produce a new data instance that is in the required format for input to another $h$-function. A set $T$ of **data transformation functions** $\tau \in T$ transforms data instances $\hat{q} \in \hat{Q}$ into data instances $\hat{k} \in \hat{K}$, of types $q \in Q$ and $k \in K$, respectively, that are suitable for input into an $h$. Formally, a **data transformation function** $\tau: \hat{Q}^{n_\tau} \rightarrow \hat{K}$ is defined as

$$\hat{k} = \tau(\hat{q}_1, \hat{q}_2, \ldots, \hat{q}_{n_\tau}) \tag{9}$$

where $\hat{k} \in \hat{K}$ and $\hat{q}_j \in \hat{Q}$, $1 \leq j \leq n_\tau$. Here, $n_\tau$ is the number of input arguments to $\tau$.

An **h-function** (or **function**) $h \in H$ represents a microservice that performs some unit of work in a pipeline. An $h$-function takes as input a sequence of $n_i$ input data instances and computes a sequence of $n_o$ output data instances. Each input data element $\hat{k}_j \in \hat{K}$, $1 \leq j \leq n_i$, has been verified through an $s \in S$, identified from $s_{ID} \in S_{ID}$, so that the inputs to $h$ are valid (i.e., so that the appropriate $s \in S$ outputs a 1 for each instance $\hat{k}_j$). Also, each of these input data instances may have been generated by transforming data into the required format, using one data transformation function $\tau \in T$. Each $h$ outputs a sequence of instances of $\hat{\ell}_j \in \hat{L}$, ($1 \leq j \leq n_0$) which are also verified through $s_{ID} \in S_{ID}$ and elements $s \in S$, so that the sequence of outputs from $h$ are valid (i.e., so that the appropriate $s \in S$ outputs a 1 for each instance of $\hat{\ell}_j$). Thus, we have the following. An $h$-function is $h: \hat{K}^{n_i} \rightarrow \hat{L}^{n_o}$ is defined by

$$(\hat{\ell}_1, \hat{\ell}_2, \ldots, \hat{\ell}_{n_o}) = h(\hat{k}_1, \hat{k}_2, \ldots, \hat{k}_{n_i}) , \tag{10}$$

where $\hat{k}_j \in \hat{K}$, $1 \leq j \leq n_i$, and $\hat{\ell}_j \in \hat{L}$, $1 \leq j \leq n_o$.

It is useful to define the composition of all *h*-functions within a pipeline, because this composition identifies the order in which *h*-functions execute. It naturally identifies the (input) data files that must exist before the pipeline starts and which output files are generated. Some input files for some *h*-functions are not specified initially because they are generated by other [*preceding*] *h*-functions. As the preceding model description indicates, one data transformation function may need to be executed on each input before each *h*-function is invoked, to put each input data instance *k* into the required format for *h*. If there are $n_i$ inputs to *h*, then the number of data transformation functions is $n_i$ (one or more transformation functions may be the identity function). Hence, executing one *h*-function $h_j$ can be thought of as a composition of functions $(\tau_j^*, h_j) = (h_j \circ \tau_j^*)$, where $\tau_j^*$ represents the $n_{i_j}$ transformation functions that are required to put all inputs for $h_j$ into the proper formats to execute $h_j$. A **composition of $n_f$ *h*-functions** $\mathcal{H}: \hat{K}^{n_{p,i}} \to \hat{L}^{n_{p,o}}$ is defined by

$$\mathcal{H} = (h_{n_f} \circ \tau_{n_f}^*) \circ (h_{n_f-1} \circ \tau_{n_f-1}^*) \circ \cdots \circ (h_2 \circ \tau_2^*) \circ (h_1 \circ \tau_1^*) , \qquad (11)$$

where $(\hat{\ell}_1, \hat{\ell}_2, \ldots, \hat{\ell}_{n_{p,o}}) = \mathcal{H}(\hat{k}_1, \hat{k}_2, \ldots, \hat{k}_{n_{p,i}})$.

We define $\hat{K}^* = \hat{K}^{n_{p,i}}$ and $\hat{L}^* = \hat{L}^{n_{p,o}}$ as short-hand. Thus, the $n_{p,i}$ input files that must exist before the pipeline is invoked are represented by $\hat{K}^*$. The $n_{p,o}$ pipeline outputs are represented by $\hat{L}^*$. It is often convenient to represent $\mathcal{H}$ as the (ordered) sequence $((\tau_1^*, h_1), (\tau_2^*, h_2), \ldots, (\tau_{n_f-1}^*, h_{n_f-1}), (\tau_{n_f}^*, h_{n_f}))$, where the ordering gives the order of execution.

## 6.2 Algorithm of the execution of the pipeline framework

With the formalism of Section 6.1, the execution of the pipeline framework is now presented. Algorithm 1 contains the algorithm. The algorithm steps through each $h_i \in \mathcal{H}$ and for each input $\hat{k}_i$ of $h_i$, determines whether it needs to be created by transforming one or more data instances. If so, the inputs $\hat{q}_i'$ to the transformation function $\tau$—for computing $\hat{k}_i$—are obtained. They are verified using schema verification functions *s*. The transformation function is executed and the output data instance $\hat{k}_i$ is verified. At this point the required input data for $h_i$ exist, and $h_i$ is invoked and the output files are generated. These outputs are stored. Note that at various points, data file formats are verified by using schema verification functions. The output files $\hat{L}^*$ are returned.

**Algorithm 1** Steps of the Algorithm PIPELINE EXECUTION.

```
Algorithm 1: PIPELINE EXECUTION.
Input: Pipeline configuration filename. h-functions of H to execute
for the pipeline P. Data transformation functions T to execute. The
set K̂* of input files for the pipeline. Identification of the nᵢ inputs
K̂ⁿⁱ = (k̂₁, k̂₂,...,k̂ₙᵢ) and nₒ outputs L̂ⁿᵒ = (ℓ̂₁, ℓ̂₂,...,ℓ̂ₙₒ) for each h-function.
Inputs q̂₁, q̂₂,...,q̂ₙₜ ∈ Q̂ and output q̂' ∈ Q̂ for each data transformation
function τ ∈ T, for each hᵢ ∈ H. The set S of schema s ∈ S for verifica-
tion of data elements q̂. The set S_ID of schema ID elements s_ID ∈ S_ID for
the mapping of datatypes q to schema s.
Output: The output files L̂* generated by the pipeline P, represented by
set H of h-functions.
Steps:
```

1. Read pipeline configuration file, which contains the *h*-functions $h_i$ to execute, along with pipeline inputs and file verification formats.
2. **for each** $h_i \in \mathcal{H}$ **do**

(a) Obtain the $n_i$ input instances $\hat{K}^{n_i} = (\hat{k}_1, \hat{k}_2, \ldots, \hat{k}_{n_i})$ for $h_i$ from the definition of $h_i$.

(b) **for each** $\hat{k}_i \in \hat{K}^{n_i}$ **do**

   i. **if** $\hat{k}_i$ requires generation from existing data files (i.e., using a data transformation) prior to input to $h_i$ **then**

      A. Get the datatype $k_i$ from instance $\hat{k}_i$.

      B. Identify the transformation function $\tau$ using $\tau = \tau_{ID}(h, k_i)$.

      C. Let $\hat{Q}' = \{\hat{q}'_1, \hat{q}'_2, \ldots, \hat{q}'_{n_\tau}\}$ be the set of $n_\tau$ existing input instances to the transformation function $\tau$, obtained from the definition of $\tau$, such that $\hat{k}_i = \tau(\hat{q}'_1, \hat{q}'_2, \ldots, \hat{q}'_{n_\tau})$.

      D. **for each** $\hat{q}'_j \in \hat{Q}'$ **do**

         1. Obtain the datatype $q'_j$ from instance $\hat{q}'_j$.

         2. Obtain the schema $s \in S$ as $s = s_{ID}(q'_j)$.

         3. Verify $\hat{q}'_j$ by computing $s(\hat{q}'_j)$. If $s(\hat{q}'_j) = 1$, then $\hat{q}'_j$ is verified. If $s(\hat{q}_j) = 0$, then $\hat{q}_j$ is not verified; an error is found, and this routine terminates.

      E. Use the data transformation function $\tau$ to compute the input $\hat{k}_i$ for $h_i$, in the proper format, according to $\hat{k}_i = \tau(\hat{q}'_1, \hat{q}'_2, \ldots, \hat{q}'_{n_\tau})$.

   ii. Obtain the schema $s = s_{ID}(k_i)$.

   iii. To verify $\hat{k}_i$, compute $s(\hat{k}_i)$. If $s(\hat{k}_i) = 1$, then $\hat{k}_i$ is verified. If $s(\hat{k}_i) = 0$, then $\hat{k}_i$ is not verified; an error is found, and the pipeline terminates.

(c) Invoke function $h_i$ and compute $(\hat{\ell}_1, \hat{\ell}_2, \ldots, \hat{\ell}_{n_o}) = h_i(\hat{k}_1, \hat{k}_2, \ldots, \hat{k}_{n_i})$.

(d) Verify the format of each output $\hat{\ell}_j (1 \leq j \leq n_0)$ by obtaining the corresponding datatype $\ell_j$ and schema $s = s_{ID}(\ell_j)$, and invoking $s(\hat{\ell}_j)$. If $s(\hat{\ell}_j) = 1$, then the output file format is verified. Else $\hat{\ell}_j$ is not verified, which is an error, and the pipeline gracefully terminates.

(e) Store the outputs $(\hat{\ell}_1, \hat{\ell}_2, \ldots, \hat{\ell}_{n_o})$ in $\hat{Q}$, which may be used as inputs for subsequent $h_j \in \mathcal{H}$, $(j \neq i)$.

(f) Store the outputs $(\hat{\ell}_1, \hat{\ell}_2, \ldots, \hat{\ell}_{n_o})$ in $\hat{L}^*$, which is the set of outputs from the pipeline.

3. Return $L^*$.

The description thus far in this section is focused on a single pipeline. However, the model is equally valid across pipelines. In fact, grouping sets of $h$-functions into multiple pipelines, as we do herein, is largely a matter of practicality, and aids in software system organization and in reasoning about such systems. However, from Section 6.1 and this Section 6.2, it should be clear that all data transformation functions and $h$-functions could be put into a single $\tau^*$ large pipeline.

## 6.3 Mapping of model onto the software system

One reason for the particular development in Section 6.1 above is that it parses the model into components that are the responsibility of the pipeline framework, software that users put into a pipeline, and user-supplied information regarding data. For example, input datatypes $K$ and instance $\hat{K}$ for a pipeline or a collection of pipelines must be supplied by an analysts, or come from some previous analysis.

The schema ID mapping and schema themselves are provided by the analyst to ensure that input and computed results conform to specified formats and contain the proper types of information. The execution of schema to verify data representation instances is the responsibility of the pipeline (not the functions). Data transformation functions and *h*-functions are executable software, and may be stand-alone executables that constitute processes. They are provided by an analyst or software developer. It is the pipeline's responsibility to invoke the correct functions and in the correct order, and to access the proper input files and to store the resulting output files, all of which are specified in a human-generated **pipeline configuration** file (addressed below). Functions are responsible for generating correct outputs.

## 7 Pipeline design and implementation

With the conceptual view of pipelines in Section 5 and the mathematical model and algorithm in Section 6, we now present the pipeline design and implementation. We address several topics in this section and in the referenced appendices. These include the composability of pipelines, pipeline configuration files, descriptions of the five pipelines, *h*-functions and their configuration files, examples of pipeline configuration files, detailed representations of two of the pipelines, and a compilation of all implemented *h*-functions.

### 7.1 Pipelines

Two pipelines are depicted with black boxes in Fig 11. The major elements of a pipeline are the configuration file, data files and schema, pipeline framework, *h*-functions, and transformation functions. Table 5 provides additional overview of several of these elements.

All pipelines in the system have been developed on this project and for the work described herein. We have added pipelines and functions over the course of a year, demonstrating the extensibility of the system, without modifying the pipeline framework code discussed in Section 7.1.2.

**7.1.1 Pipeline configuration file.**   To run a pipeline (called a job), a configuration input file specifies functions and their order of execution. Table 6 overviews the entire pipeline configuration file with a definition for each parameter. JSON schema files exist for each component in the data common specification from Section 3.3. The functions component defines the available *h*-functions to run in the pipeline and the input files for each function. Appendix B contains a detailed example of a configuration file.

Fig 29 shows the schema for a configuration file that specifies how to compose and execute one or more functions of a pipeline. In Fig 29, there are up to five functions available and the required parameters for each function are defined; the enumeration is the list of valid candidate values that can be specified for functions in a specific pipeline.

**7.1.2 Pipeline framework and data file schema.**   The pipeline framework software of Fig 11 (written in Python) performs these operations: (*i*) reads and parses the configuration file; (*ii*) controls accessing input files, JSON schema files, transformation codes, and h-functions; (*iii*) checks files against their JSON schema and terminates gracefully if a verification fails; (*iv*) invokes the proper transformation functions (if applicable), (*v*) invokes the proper *h*-functions in their proper order (and any other operations), and (*vi*) handles errors.

JSON schema are used in various ways: (*i*) to verify the configuration file, (*ii*) to verify inputs to transformation functions, (*iii*) to verify the outputs of transformation functions (which are inputs to the *h*-functions), and (*iv*) to verify the outputs from the *h*-functions. The pipeline operations above and the use of schema are both reflected in the algorithm of Algorithm 1.

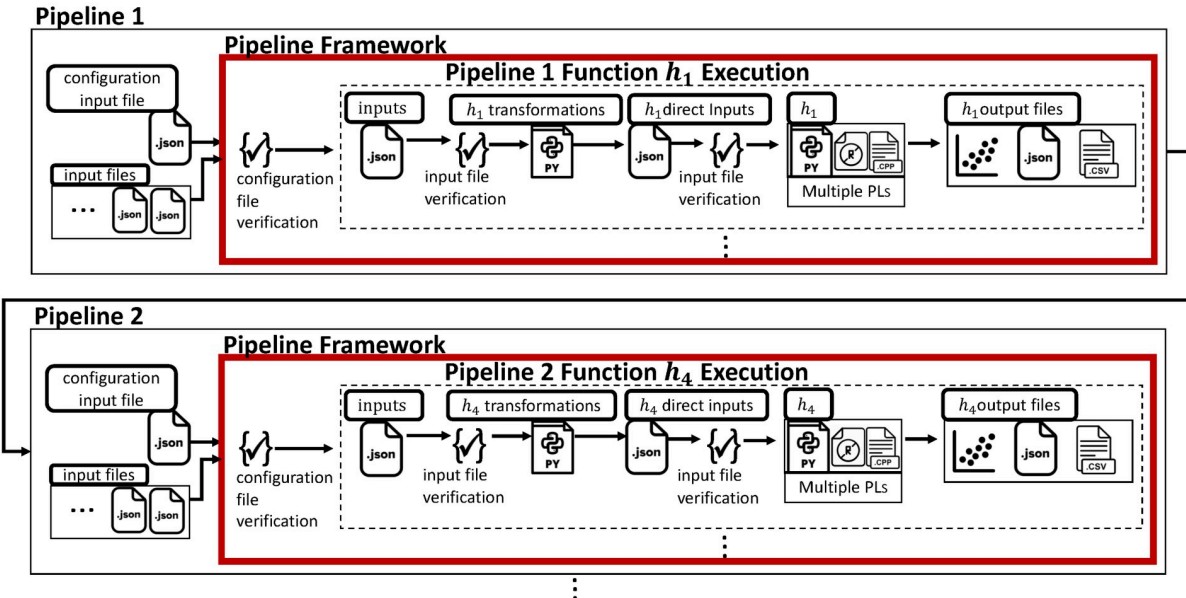

**Fig 11. Two pipelines are shown to illustrate similarities and differences between them.** To run a pipeline (called a job), a pipeline-specific configuration input file is verified and is read by the pipeline framework. The file specifies $h$-functions and their order of execution, as well as required input files to the pipeline. Here we show how function $h_1$ is executed in a pipeline 1 and how $h_4$ is executed in pipeline 2. The pipeline framework invokes the corresponding functions. If specified in the configuration file, the pipeline framework invokes a transformation function that transforms the contents of one or more files into an input file of correct format for the $h$-function. There may be one transformation function for each direct input to an $h$-function. At appropriate points in a pipeline, data files are verified against their corresponding JSON schema (input file verification). The $h$-function is executed and output files are generated (these digital object outputs may be, e.g., plot files, ASCII data files, and binary data files). There may be additional $h$-functions within pipeline 1, indicated by the ellipsis below pipeline 1 function $h_1$ execution. In this example, outputs from the generic pipeline 1 are inputs for the generic pipeline 2. Function $h_4$ in pipeline 2 is executed in a similar fashion to function $h_1$ in pipeline 1. See the text for descriptions of these various components. Note: the pipeline framework (in brown) is the same code for all pipelines. See Table 5 for implementation details of the elements in this figure.

**Table 5. Sections and files from the execution of a generic Pipeline.**

| # | Input File Name | File Type | Description |
|---|---|---|---|
| **Pipeline *i*** : In this section the input files are specified for execution. | | | |
| 1 | Configuration input file | JSON | Specifies $h$-functions to execute within pipeline $i$, and their order of execution. |
| 2 | Input files | JSON | Input files to a pipeline, i.e., files required to execute $h$-functions in the pipeline (possibly outputs from upstream pipelines). |
| **Pipeline framework**: In this section the functions are invoked, specifying the order in which they are executed. | | | |
| 1 | Configuration file verification | JSON | Input files are validated against their corresponding JSON schema. |
| 2 | Pipeline framework code | Python | Reads and parses the configuration file and controls execution of the $h$-functions. |
| **Pipeline *i* Function Execution**: In this section the functions are executed. | | | |
| 1 | Function transformation | Python | Input files are transformed into a valid input file for function $h_i$. |
| 2 | Direct input file | JSON | Input files with the required formats that function $h_i$ receives as input for execution. |
| 3 | Schema files | JSON | Input files are validated against their corresponding JSON schema. |
| 4 | Function Execution | Multiple programming languages | Function $h_i$ code is executed. |
| 5 | Function Output Files | Multiple formats | Function $h_i$ output files. |

Fig 11 describes how these elements interact, here we define and describe them.

**Table 6. Configuration input file description.**

| # | Component | Description |
|---|-----------|-------------|
| 1 | experiment | Experiment Schema JSON file location. See Fig 24. |
| 2 | phasedesc | Phase Description Schema JSON file location. See Fig 25. |
| 3 | phase | Phase Schema JSON file location. See Fig 26. |
| 4 | action | Action Schema JSON file location. See Fig 27. |
| 5 | player | Player Schema JSON file location. See Fig 28. |
| 6 | functions | The parameters inside vary for every *h*-function. Fig 29 shows a definition for five functions. |

See Appendix B for details.

**7.1.3 Functions within pipelines.** Each pipeline has a list of available functions. The functions can be written in any programming language. Currently we have *h*-functions written in C++, Python, and R. A function may use as input any combination of outputs from preceding functions in the same pipeline, functions in preceding pipelines, files from previous iterations, and data from experiments.

Currently there are 29 functions across five pipelines. A summary of the *h*-functions in each of the five pipelines is provided in Table 7. Listings and details of all functions implemented per pipeline are provided in Appendix D (one table for each pipeline).

# 8 Case studies

The purpose of the three case studies is to demonstrate the utility (i.e., usefulness) of the pipeline system. The first case study (Study 1) uses all five pipelines. This study took two years to complete, in building software, running experiments, varying treatments, analyzing data, building multiple models, validating and exercising models, and hypothesis testing. We iterated over these operations, as suggested in Fig 3. The pipelines of this manuscript were used for all of the work in this case study. We consider this to be a very large case study. The purpose of case studies 2 and 3 are different. Our goal here is to demonstrate the versatility and wide applicability of the pipeline system. For each of these cases studies, we take experiments or computations from other researchers' works in the literature, and demonstrate through our data model that our system can analyze the data and computations of those works. In case study 3, we could also include their model in our pipelines. Other works in the literature [1, 3–6, 9, 55] can also be analyzed with our pipelines.

**Table 7. Summary table of *h*-functions.**

| Name | Acronym | Number of *h*-functions | Description of Some Functions |
|------|---------|-------------------------|-------------------------------|
| Experimental Data Transformation Pipeline | EDTP | 1 | $h_1$ transforms experimental raw data into our data common specification. |
| Data Analytic Pipeline | DAP | 1, 2, 3, 4, 5, 6, 7, 8, 9, 10, 11, 12, 13, 14 | $h_1$ detects common patterns between players and actions through a visualization. $h_3$ shows with data files and a plot how an action progresses in time during an experiment phase. $h_7$ through time series data files generates input for the Property Inference pipeline. |
| Property Inference Pipeline | PIP | 1, 2, 3, 4 | $h_1$ generates the properties for a Markovian transition matrix. $h_2$ outputs a file with the properties for an adapted conditional random fields (CRF) model. |
| Modeling and Simulation Pipeline | MASP | 1, 2, 3, 4, 5 | $h_1$ generates Agent Based Model Simulations outputs for self-consistency checks and predictions. $h_5$ executes agent based simulation component models to compare outputs with real actions from a real experiment. |
| Model Evaluation and Prediction Pipeline | MEAPP | 1, 2, 3, 4, 5 | $h_1$ compares experiment outputs with simulation outputs. $h_2$ generates statistical models to predict outcomes. |

## 8.1 Study 1: Entire system execution for collective identity experiments

Collective identity (CI), as defined by [43], is an individual's cognitive, moral, and emotional connection with an enclosing broader group such as a team or a community. CI is important in many applications and contexts, making it worthy of study. For example, CI is important in the formation and maintenance of teams, and team behavior [56, 57]. It is also important in the formation and enforcement of norms [56, 57].

Here, we use a complete cooperatively game to produce CI among team members that are playing. We want to measure the amount of CI created between team players in an experiment. The experiment includes 3 phases. In phase-1, the DIFI index [58] measures (for a baseline) the individual levels of CI. In phase-2, CI is created between team members using a *collaborative* anagram game; In phase-3, using the same index as in phase-1, the individual levels of CI in players are measured.

Here, we use the Dynamic Identity Fusion Index (DIFI) score [58] as a proxy for CI. The DIFI score is measured individually as part of our online experiments in the following way. A small (movable) circle represents an individual player and a second (stationary) larger circle represents the team. A player moves the small circle along a horizontal axis, where the distance between circle centroids represents that player's sense of identity with the team; it is their DIFI score. The range in DIFI distance value is, $-100 \leq DIFI \leq 125$; $DIFI = 0$ corresponds to the two circles just touching, $DIFI < 0$ means that the two circles are disjoint (an individual has no positive affinity for the team), and $DIFI > 0$ means that the two circles overlap (an individual identifies with the team).

As a priming activity to foster CI among team members, in phase-2, they play a *collaborative* word construction (anagram) game motivated by [6]. This Phase 2 is the focus of our case study.

**8.1.1 Web-based experiment software platform, game play and data collection.** We built a web application to conduct experiments. The primary components of our platform are the oTree framework [59], Django Channels and the online web interface. Each phase of the experiment has software, designed and developed, that interfaces with oTree. Interactions among players is supported by Django Channels technology; individual participants and the server communicate by websocket. Fig 12 shows the web interface for each player of the anagram game. The experiment interface enlists players from Amazon Mechanical Turk (MTurk) and registers actions from all the players in all phases. The clicks and their event times represent the actions for defined HTML objects like letters, and submit buttons.

In phase-2, at the beginning of a game, players receive three letters, and communication channels to $d$ number of other players; through these channels players can help each other to form new words by sharing letters. Based on the recruited number $n$ of players, the experimental platform creates a graph with a pre-defined regular degree $d$ on the $n$ players. Players of the game can perform the following actions, request letters from neighbors, reply to letter requests from neighbors, and form words; these actions are explained in detail in the caption of Fig 12.

The objective of the game is to form as many words as possible as a *team*. The total number of words formed by the team defines the earnings in a game. Earnings are divided uniformly between players. For a player to form a valid word, the word has to be unrepeated in the player's list of formed word; however more than one player can duplicate a word. Each player possess an infinite stock of each of the three initial letters received. This means a player can use these initial letters more than once to form words, and also openly share them with neighbors. These features are planned to promote cooperation.

**8.1.2 Data analysis, modeling and simulations, and modeling evaluations using the pipelines.** Some data model features from Table 3 are provided in Fig 13. For the DIFI

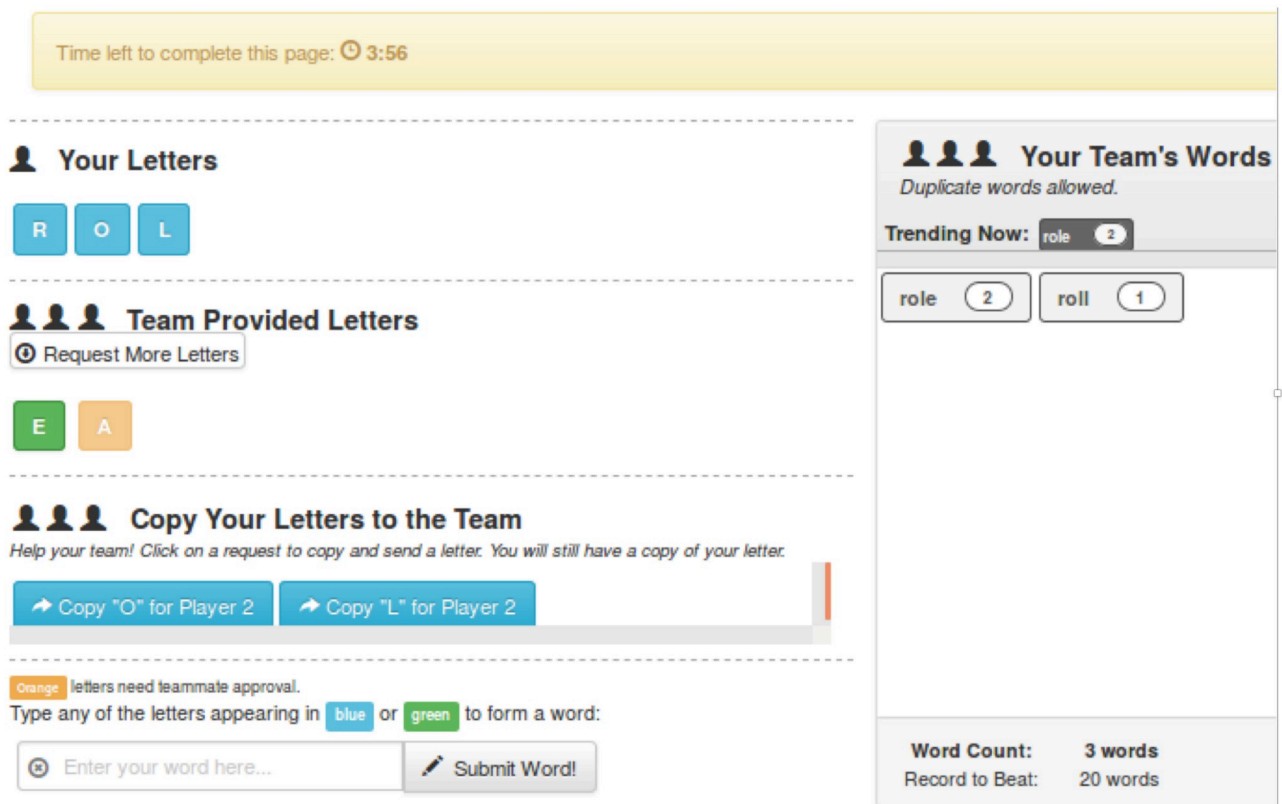

**Fig 12. The anagram game screen, phase-2, for one player.** This player has own letters "R," "O," and "L" and has requested an "E" and "A" from neighbors. The "E" is green, so this player's request has been fulfilled and so "E" can be used in forming words; but the request for "A" is still outstanding so cannot be used in words. Below these letters, it shows that Player 2 has requested "O" and "L" from this player. This player can reply to these requests, if she so chooses. Below that is a box where the player types and submits new words.

measures (phases 1 and 3), the action set $A$, with its one element (submit DIFI score), is shown, and the action sequence $T$ is the action tuple of submitting DIFI score for each agent. For phase 2, the word construction game, the edge set $E$ for the four players is provided, as is the action set $A$, containing four elements. The action "thinking" is a no-op in the model. Initial letter assignments to players, which are part of $B_j^v$ for each node (player) $v_j$, are shown. So, too, is an illustrative sequence of action tuples. For example, $T_3$ states that $v_i$ requests the letter "G" from $v_3$.

Several ABMs were built to model the phase 2 group anagram game. The ABM described here is build on a transition probability matrix where the transition probability from one action $a(t) = a_i$ at time $t$ to the next action $a(t + 1) = a_j$ for each agent $v$, $i, j \in [1..4]$ and $a(t) \in A$, is given by $\pi_{ij} = Pr(a(t+1) = j | a(t) = i)$ with $\sum_{j=1}^{4} \pi_{ij} = 1$. We use $i$ and $j$ to represent the actions $a_i$ and $a_j \in A$. Agent $v$ executes a stochastic process driven by transition probability matrix $\Pi = (\pi_{ij})_{m \times m}$, where $m \equiv |A|$ (here, = 4). A multinomial logistic regression model is used for $\pi_{ij}$. Details are in [7]. During the 5-minute game, the ABM predicts action tuples $T_i$ for players $v_i$ participating.

In this study, the system of Fig 3 is executed over many loops; some times completely and other times portions of it. In this case study we examine ***only the anagram game***. We perform one iteration of three experiments, with $n = 6$ for the number of players and $d = 5$ for the number of neighbors. Figs 14–16 display results for the Data Analytics Pipeline (DAP). Fig 17

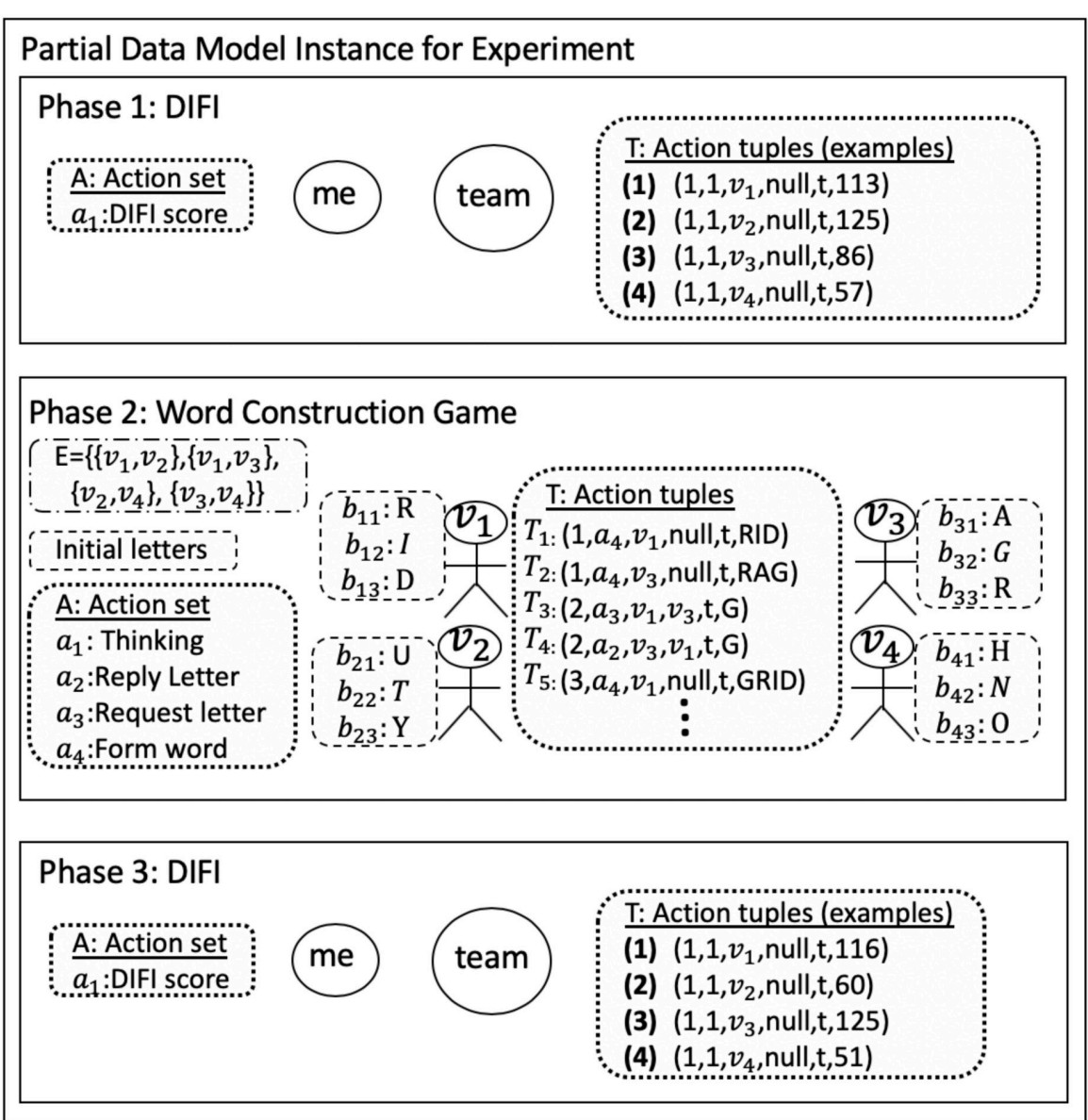

**Fig 13. Case study 1.** Partial representation of the data model for the online experiment composed of 3 phases with a set of *V* players ($n = |V|$). The phase 1 DIFI measure, a proxy for CI, uses a null (i.e., empty) network on *n* players; i.e., there are no edges in the graph because players play individually. In phase 2, a team-based CI-priming game, edges *E* are communication channels. Initial conditions $B^v$ include letter assignments to players. The individual DIFI measure is repeated in phase 3. The action set *A* and illustrative action tuples $T_i$ are given for each phase.

display results for the Property Inference Pipeline (PIP). Fig 18 display results for the Modeling and Simulation Pipeline (MASP) and Model Evaluation and Prediction Pipeline (MEAPP). The figure captions provide details. Here output data from a pipeline are inputs for another pipeline: (*i*) outputs from the DAP are inputs to the PIP; (*ii*) outputs from the PIP are inputs to the MASP; and (*iii*) outputs from the DAP and MASP are inputs to the MEAPP.

The following paragraph discusses special details of these results. Fig 14 presents a plot, generated by $h_3$, of the time series of words formed for each player of one game. When a new word is formed a step in a curve indicates the time. "Form word" is $a_4 \in A$ in Fig 13. $h_3$ can

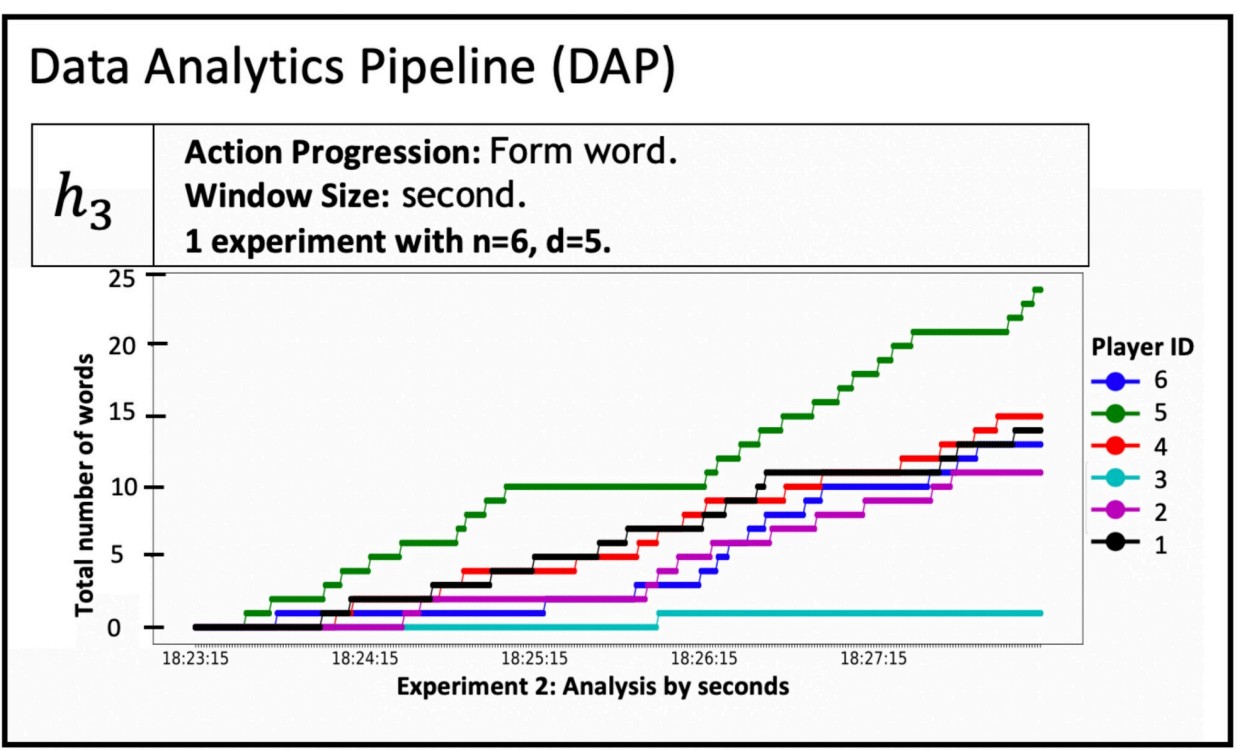

**Fig 14. The Data Analytics Pipeline (DAP) was executed to analyze phase 2 of three experiments with $n$ = 6 and $d$ = 5.** The time series of number of words formed by player for experiment #2 is generated by function $h_3$.

construct the time series for any action. These data, and the data generated by $h_5$ in Fig 15, are used to (*i*) understand player behaviors, (*ii*) help in idetifying the structure of ABMs, (*iii*) infer properties of ABMs, and (*iv*) assist in models validation with the comparison of model predictions. Function $h_7$ produces the data needed for property inference and showed in Fig 16.

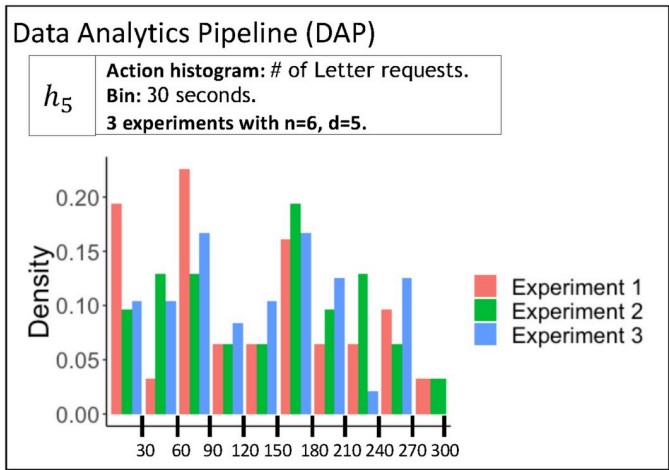

**Fig 15. The Data Analytics Pipeline (DAP) was executed to analyze phase 2 of three experiments with $n$ = 6 and $d$ = 5.** The histogram for the number of actions "letter request" for three experiments is generated by function $h_5$. The x-axis is time in the group anagram game, binned in 30 seconds intervals.

**Fig 16. The Data Analytics Pipeline (DAP) was executed to analyze phase 2 of three experiments with $n = 6$ and $d = 5$.** The discrete time actions for all three experiments is generates by function $h_7$. This latter output will inform the Property Inference pipeline for computing parameters for simulation models. Time (in seconds) is shown in the first row as 1, 2, 3, . . ., and counts of the $z$ vector components, per player and per experiment are given.

The $\beta$ coefficients in Fig 17 are parameters in the multinomial logistic regression model alluded to above. In the $\pi_{ij}$ terms above, each transition is from action $i$ to $j$. For example, the $\beta$ coefficients at the bottom are for the transition from forming word ($a_4$ in Fig 13) to the next actions being $a_2$ through $a_4$; the probability that the next action is $a_1$ (thinking) is 1 minus the sum of other three transition probabilities.

In Fig 18, the Modeling and Simulation Pipeline is employed to create all three plots (the first two for simulating experiments, the third for predictions beyond the experiments). The Model Evaluation and Prediction Pipeline is employed in the first two plots to compare experiments and model predictions.

Appendix F describes two more case studies. Study 2 in Appendix F.1 shows the data model for online experiment in [3]. Study 3 in Appendix F.2 shows the data model for a simulation study in [44].

## 9 Related work

We address several different topics below.

### 9.1 Online social science experiments

In order to understand human behavior, there has been significant interest in using online systems to carry out social science experiments. These experiments analyze a variety of phenomena, like collective identity [17, 60, 61], and cooperation and contagion [62], to name a few. The methodological and practical challenges of online interactive experimentation, and the value of an online labor market has been discussed in different studies [63, 64]. The benefits of online experiments, compared to in-person experiments, include reduced costs, an agile logistic process, and the collection of detailed data. Research teams use different options to design and deploy their online experiments. While some teams, create web-based programs especially designed for their research [17, 61, 62], others use web-based experimental platforms that provide this service [60, 63]. In [60] the online platform Volunteer Science [35] was used to

## Property Inference Pipeline (PIP)

| $h_2$ | Beta coefficients for Model ABM 1 generated from experiment data with n = 6, d = 5. The parameters are for the given features in the column names and the β coefficient for computing the next action i that are the row state/action labels, from state i to j, i, j ∈ {1, 2, 3, 4}. |
|---|---|

| Transition 1 to $j$ | (Intercept) | buffer $z_B$ | letter $z_L$ | Words $z_W$ | constant $z_C$ |
|---|---|---|---|---|---|
| 2 | -3.9240 | 0.2604 | -0.0312 | 0.0061 | -0.0172 |
| 3 | -2.9071 | -0.0895 | -0.0406 | -0.0111 | -0.0126 |
| 4 | -4.0571 | 0.0812 | 0.1796 | 0.0272 | -0.018 |

| Transition 2 to $j$ | (Intercept) | buffer $z_B$ | letter $z_L$ | Words $z_W$ | constant $z_C$ |
|---|---|---|---|---|---|
| 2 | -2.8873 | 1.2164 | 0.2115 | -0.1066 | 0 |
| 4 | -6.5411 | -6.3222 | 0.0799 | -1.6579 | -0.1185 |

| Transition 3 to $j$ | (Intercept) | buffer $z_B$ | letter $z_L$ | Words $z_W$ | constant $z_C$ |
|---|---|---|---|---|---|
| 2 | -5.5048 | 0.2570 | 0.2097 | -0.0523 | 0 |
| 3 | -4.1109 | -67.1075 | 0.0425 | -0.2558 | 0 |

| Transition 4 to $j$ | (Intercept) | buffer $z_B$ | letter $z_L$ | Words $z_W$ | constant $z_C$ |
|---|---|---|---|---|---|
| 2 | -5.2707 | 0.2285 | 0.1973 | -0.0681 | 0 |
| 3 | -1.3798 | 0.7187 | -3.3517 | 0.7732 | 0 |
| 4 | -3.4645 | -0.4355 | -0.0116 | 0.0769 | 0 |

**Fig 17. The Property Inference pipeline receives the input from $h_7$ of the Data Analysis Pipeline (DAP).** The parameters in this figure were generated to inform an ABM model for the Modeling and Simulation Pipeline (MASP). The transitions in the figure are from from $i$ to $j$, where $a_i \in A$ is the action at time $t$ and $a_j \in A$ is the action at $(t + 1)$. Rows not shown mean there are no such transitions in the data.

implement a web-based public goods experiment, and to recruit participants around the world. In [63], a repeated public goods experiment was implemented in the free web-based platform for interactive online experiments, LIONESS [36], and participants were recruited via Amazon Mechanical Turk (MTurk). In [37] a modular virtual lab named Empirica offers a development platform for virtual lab experiments, and they claim that is even accessible to novice programmers. There are tools that focus in Adaptive Experimentation, like Facebook Ax [38], an accessible, general-purpose platform for understanding, managing, deploying, and automating adaptive experiments. Usually these platforms only focus on the design and running of online lab experiments, but they don't offer a complete automated solution for experiments, analysis, modeling and simulation, and evaluation.

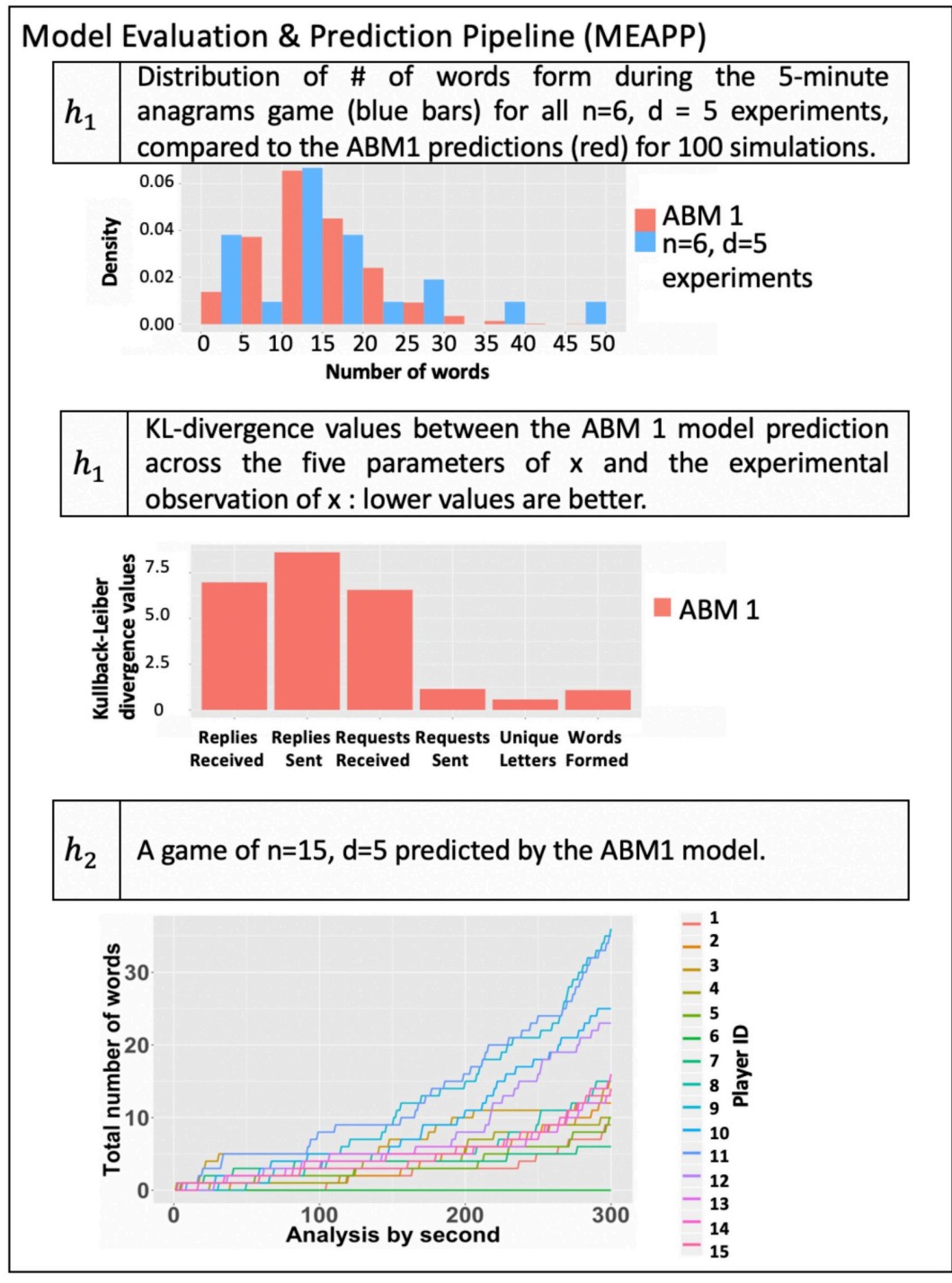

**Fig 18. The Modeling And Simulation Pipeline (MASP) and Model Evaluation And Prediction Pipeline (MEAPP) were run to obtain simulation results and model predictions, and to compare experimental data to model predictions.** All three plots contain model predictions and use results from $h_1$ of the MASP. Function $h_1$ of MEAPP plots corresponding experimental and model output data (top plot) and compares experiment and model output using KL-divergence (center plot) for six parameters. Function $h_2$ of MEAPP uses $h_3$ of the Data Analysis pipeline (DAP) to plot model predictions from $h_1$ of the MASP (bottom plot) where now $n = 15$ (in experiments, $n = 6$).

**(Networked) Experiments in the social sciences**. Experiments with interacting participants can be represented as networks, where edges represent interaction channels. There are several online and in-person experiments with individuals [60, 61, 65–69] and groups [1, 3–6, 9]. Some include modeling of the experiment [9]. Also, none of these works appears to do iterative evaluations involving modeling and experiments. There is no platform, that we know of, that allows the iterative process of data analysis, design of data-driven model to simulate experiments, model validation and verification in order to predict behavior. In this work our focus is to formalize a general methodology, through a generic data pipeline, for online controlled experiments of human subjects aim to explain diverse phenomena.

**Simulation frameworks**. There are many frameworks for developing simulations. In [19] four design patterns systematize and simplify the modeling and the implementation of multi-level agent-based simulations. In [20] a framework for developing agent-based simulators as mobile apps and online tools is presented. They present a case study in the field of health and welfare. In [21] a methodology for an artificial neural network based metamodeling of simulation models is presented. The model is for the case when online decision making routines are invoked repetitively by the simulation model throughout the simulation run. We believe, none of these frameworks provides composable and extensible pipelines for studying networked social science phenomena, in order to address social sciences for modeling/experiments.

## 9.2 Workflow systems

There are many workflow systems. Here, we cite several popular workflow systems and then describe how they relate to social sciences and pipelines for computation. Examples include Taverna [70] for bioinformatics, chemistry, and astronomy; Pegasus [71] and CyberShake, built on Pegasus [72], for large-scale workflows in astronomy, seismology, and physics; Kepler [73, 74] for ecology and environmental workflows. Other workflow engines include Toil [75], and Rabix [76] developed for computational biology.

We believe, none of these systems addresses social sciences for modeling/experiments as we do here. As an illustration, suicide data is analyzed with Taverna in [77] and Galaxy is used for genomic research [78]; neither has a component for modeling.

In the social sciences most workflows are for social network analyses [28]; we seek to go well beyond that. Also in [79], a taxonomy of features is defined from the way scientists make use of existing workflow systems; this provide end users with a mechanism by which they can assess the suitability of workflow to make an informed choice about which workflow system would be a good choice for a particular application. The importance of interoperability between these systems is detailed in [80] and identifies three dimensions; execution environment, model of computation (MoC), and language. MoCs provide the semantic foundation, but a data model is a prerequisite. [27, 28, 79, 81] are among the works that overview several workflow systems. An overview and discussion of future directions is provided in [82]. Challenges and future directions for life science workflows are provided in [83]. Ontologies for workflow objects are discussed in [84].

Workflow languages are usually represented in a textual manner, or through graphical interfaces. A textual representation is often employed for storing the workflows in files, even when a graphical representation is employed. For full interoperability, it is important to have the capacity to translate between workflow languages [80]. Wings [85] uses rich semantic representations to describe compactly complex scientific applications in a data-independent manner. Swift [86] and Swift/T [87, 88] are workflow languages built for executing parallel programs within workflows. NextFlow [89] is a domain specific language for computational workflow management systems. Workflow languages include Common Workflow Language

(CWL) [76, 90] and Workflow Description Language (WDL) [91]. Script of Scripts [92] is a workflow system with an emphasis on support for different scripting languages.

### 9.3 Microservices

Our pipelines take a microservices conceptual approach. First defined in 2012, Microservices [93] is an architectural style, addressing how to build, manage, and evolve architectures out of small, self-contained units [40–42, 94]. The *h*-functions of our pipelines have a narrow scope; this way, for new experiments and models new functions can be included in a specific way, promoting reuse by not presenting repeated capacities.

Microservices Architecture (MSA) and Service-Oriented Architecture (SOA) both rely on services as the main component. But they vary greatly in terms of service characteristics. SOA divides applications into sets of business applications offering services through different protocols. This aims to solve the problem of complexity. SOA applications are costly and complex and are designed to support high workloads, and a large number of users. In [93] is stated that microservices keep services independent so that a service can be individually replaced without impacting an entire application.

In 2012 [95] defined microservices as a way to more swiftly build software by dividing and conquering, using Conway's Law to structure teams. Issues, advantages and disadvantages of microservices are identified in [96]. For example an issue identified is the system decomposition. Advantages include the increase in scalability and the clear boundaries. Disadvantages include the difficulty to learn. The microservice architectural style is largely used by several companies such as Amazon [97], Netflix [98], and many others.

### 9.4 Data models

In [99], a data model is presented for supporting the modeling, execution and management of emergency plans before and during a disaster. In [100], aspects of a business data model are described. In [101], a data model is presented for capturing workflow audit trail data relevant to process performance evaluation. In [102], models for social networks that have mainly been published within the physics-oriented complex networks literature, are reviewed, classified and compared.

In [103], an object-relational graph data model is proposed for modeling a social network. It aims to illustrate the power of this generic model to represent the common structural and node-based properties of different social network applications. A multi-paradigm architecture is proposed to efficiently manage the system. In [104], a semantic model that can naturally represent various academic social networks is presented; it describes various complex semantic relationships among social actors.

**Formal models of pipelines**. The possibility of incorporating formal analytics into workflow design is investigated in [100]. It provides a model that includes data dependencies. The workflow design analytics they propose helps construct a workflow model based on information about the relevant activities and the associated data. Also, it helps determine whether the given information is sufficient for generating a workflow model and ensures the avoidance of certain workflow anomalies. A detailed treatment of data dependencies is found in [54].

In [105], to improve data curation process efficiency for biological and chemical oceanography data studies, pipelines are defined using a declarative language. The pipelines are serialized into formal provenance data structures using the Provenance Ontology (PROV-O) data model (defined in the paper).

### 9.5 "-Ilities;" reproducibility; interoperability; composability; extensibility; scalability; reusability; and traceability

Foreseeable and unforeseeable changes occur in a system, ilities are attributes that characterize a system's ability to respond to both. Ilities describe what a system should be, providing an enduring architecture that is potent and durable, yet flexible to evolve with the insertion of new systems.

The use of ilities for systems engineering of subsystems and components is investigated in [106]. They show how some ilities are passed and used as a non-functional property of electrical and structural subsystems in aircraft. They demonstrate that a useful practice for systems engineers, to ensure that customer needs are actually met by the system under design or service, is to flow ilities down to the subsystem level. The system ilities are passed down and translated from non-functional to functional requirements by subject matter experts.

Pipelines and workflows provide reproducibility [84], interoperability [107], reusability [84]. The microservices conceptual approach of our pipelines satisfy the reproducibility, interoperability and reusability properties. We show the pipeline composability feature, also it properties for extensibility, scalability, and traceability.

## 10 Conclusion, future work, and limitations

Online social science experiments are used to understand behavior at-scale. Considerable work is required to perform data analytics for custom experiments. Furthermore, modeling is often used to generalize experimental results, enabling a greater range of conditions to be studied than through experiments alone. In order to transition from experiments to modeling, model properties must also be inferred. Consequently, our work presents a software pipeline system for evaluating social phenomena that are generated through controlled experiments. Our work scope in this manuscript ranges from formal models through software design and implementation. Our models include a formal experimental data model (and data common specification), a network-based discrete dynamical systems model (graph dynamical system, GDS), and a formal model for pipeline composition. These models aid in reasoning—in a principled way—about the architecture, design, and implementation of five software pipelines, which currently contain 29 functions. The pipelines are composable and extensible, and they can be operationalized for different methodologies (e.g. deductive and abductive analyses). We provide three case studies, on collective identity, complex contagion, and explore-exploit behavior, respectively, to illustrate the successful use of the system. We are adding these pipelines to a larger job management system and are developing new *h*-functions for developing new models. Contact Vanessa Cedeno (vcedeno@vt.edu) or Chris Kuhlman (cjk8gx@virginia.edu) for the system code. A repository with a user manual is available at https://github.com/vcedeno/PLOS_ONE_Pipelines_Supporting_Information.

There are limitations to this work. There is a host of other types of experiments that might demand different types of data analytics, and there is a variety of modeling approaches, e.g., structural equation, statistical, differential equation models, that can be added to a pipeline system. Another limitation, and an opportunity for future work, is to provide a data specification for both experiments and analyses. Specifically, Section 2.1 identified experimental platforms that are customizable [35–38] in ways that are analogous to our approach for customizable software analysis pipelines. A single specification language for experiments and analyses could be used to coordinate experiments and analyses. Also, it may be possible to use artificial intelligence techniques to provide insight into external validation based on an experiment specification.

**Table 8. Data common specification.**

| # | Component Name | Parameter from Data Model (Table 3) | Table in Data Model UML (Fig 7) | Description |
|---|---|---|---|---|
| 1 | Experiment | Experiment Schema | Experiment Schema | Experiment description and definition of initial parameters (i.e., experiment id, number of phasers, number of players, begin time, duration and list of players). |
| 2 | Phase | Phase Schema: Phase schema id, Sequence, Phase Begin, Phase duration, Unit of time, Network definition, Meaning of an edge. | Phase Schema | An experiment can have many phases. This is the Phase description and definition of initial parameters (i.e., phase id, order in experiment, begin time, duration and list of players, connections between players, number of players). |
| 3 | Phase Description | Phase Schema: Node attributes for a phase, Edge attributes for a phase, Initial conditions for nodes, Initial conditions for edges. | Edge, Initial Conditions Edge, Edge Attributes, Initial Conditions Node. | A phase has a description (i.e., phase id, beginning parameters, end parameters, actions, relations between actions). |
| 4 | Player | Experiment Schema: Player id. | Player. | Player description (i.e., player id, experiment id, phase id). |
| 5 | Action | Phase Schema: Action set, Action sequence. | Action, Action Tuple | A experiment, a phase and players have actions associated with them (i.e., action id, phase id, action tuple id, player id, timestamp, payload). |

## A Appendix: Data common specification

This appendix provides a concrete view into the system. The definition of a data common specification in Fig 3 provides the bridge between the abstract data model and the implementation of the pipelines; see Fig 6. Table 8 shows a description of the elements of the Data Common Specification. JSON schemas provide a detailed specific view of the implementation aspect of our pipelines. Because we go into detail, this is an exemplar for other types of problems. These are the types of files we use in the case studies in Section 8.

Fig 19 shows the "Experiment" definition. Fig 20 shows the "Phase" definition. Fig 21 shows the "Phase Description" definition. Fig 22 shows the "Player" definition. Fig 23 shows the "Action" definition.

## B Appendix: Mapping of model onto the software system

In this appendix, we describe the characteristics of the implementation of an individual pipeline. Figs 24–28 each show a portion of the schema for a configuration file that specifies the JSON schema file location for the experiment, phase description, phase, action, and player respectively. Fig 29 shows an example of a Configuration Input file JSON schema describing how to execute up to five functions in a pipeline.

## C Appendix: Examples of the software system

This Appendix shows examples of input files for the Experimental Data Transformation Pipeline (Fig 30), and the Data Analytics Pipeline (Fig 31). Here we show how a function is executed in a generic pipeline. Input files are validated against their corresponding JSON schema. If necessary, file contents are transformed (possibly outputs from upstream functions) to obtain the direct input for a function in the correct format. After verification of formats by the corresponding JSON schemas, the function is executed and output files are generated (these digital object outputs may be, e.g., plot files, ASCII data files, and binary data files).

Fig 30 shows an example of the (1) Experimental Data Transformation Pipeline input files and the transformations they go through. Here, the function $h_1$ takes experimental raw data and transforms it to our Data Common Specification. CSV files are transformed into JSON files, then verified for input before executing function $h_1$. After execution, function $h_1$ outputs JSON schemas that become inputs for the Data Analytics Pipeline.

```
{
    "$schema": "http://json-schema.org/draft-04/schema#",
    "title": "Experiments",
    "description": "Experiment initial parameters",
    "type": "array",
    "items" : {
    "type": "object",
    "properties": {
        "experimentid": {
            "description": "The unique identifier for an experiment",
            "type": "integer"
        },
        "p": {
            "description": "Number of phases in experiment",
            "type": "integer"
        },
        "n": {
            "description": "Number of players in experiments",
            "type": "integer"
        },
        "startime": {
            "description": "Date and time of experiment",
            "type": "string"
        },
        "duration": {
            "description": "Experiment duration in minutes",
            "type": "number"
        },
        "activeplayers" : {
          "type" : "array",
            "items" : {
                "type" : "object",
                "properties" : {
                    "playerid": {
                    "description": "Active player in experiment",
                    "type": "string"
                    }
                }
            }
        }
    },
    "required": ["experimentid","p","n"]
    }
}
```

**Fig 19. JSON schema for the "Experiment" of the data common specification.**

```
{
    "$schema": "http://json-schema.org/draft-04/schema#",
    "title": "Phases",
    "description": "Phases in an experiment",
    "type": "array",
    "items" : {
    "type": "object",
    "properties": {
        "phaseid": {
            "description": "The unique identifier for a phase","type": "string"
        },
        "phasedescriptionid": {
            "description": "The phase description id","type": "string"
        },
        "experimentid": {
            "description": "Phase experiment id","type": "integer"
        },
        "phaseorder": {
            "description": "Order of phase in experiment","type": "integer"
        },
        "begin": {
            "description": "Start time of a phase in experiment","type": "number"
        },
        "duration": {
            "description": "Duration of a phase in seconds","type": "integer"
        },
        "d": {
            "description": "Connections between players","type": "integer"
        },
        "n": {
            "description": "Number of players","type": "integer"
        }
    },
    "required": ["phaseid","phasedescriptionid","experimentid",
            "phaseorder","begin","duration"]
    }
}
```

**Fig 20. JSON schema for the "Phase" of the data common specification.**

Fig 31 shows an example of the (2) Data Analytics Pipeline execution of function $h_7$ with configuration input files examples. Here, the input JSON files are verified, then transformed into function $h_7$ direct input. After verifying the input for the function, $h_7$ is executed and the output files returned. In this example, the output file is an input for the (3) Property Inference pipeline.

## D Appendix: Pipeline functions

In this Appendix, we describe the characteristics of the the atomic element of a pipeline: the function. If a new component is added to the pipeline, it is introduced by a new function. We provide a listing of types of functions as microservices within each of the five pipelines. We show five tables, one for each pipeline, with a list of available functions. Table 9 shows one function for the (1) Experimental Data Transformation Pipeline (EDTP). Table 10 shows fourteen functions for the (2) Data Analytics Pipeline (DAP) Table 11 shows four functions for the

```
{
    "$schema": "http://json-schema.org/draft-04/schema#",
    "title": "Phases",
    "description": "Phases in an experiment",
    "type": "array",
    "items" : {
    "type": "object",
    "properties": {
        "phaseid": {"description": "The unique identifier for a phase","type": "string"},
        "beginparameters" : {
          "type" : "array","items" : {"type" : "object",
                "properties" : {
                    "parameter": {
                    "description": "Beginning parameters in phase","type": "string"}}}},
        "endparameters" : {
          "type" : "array","items" : {"type" : "object",
                "properties" : {
                    "parameter": {
                    "description": "End parameters in phase","type": "string"}}}},
        "actions" : {
          "type" : "array","items" : {"type" : "object",
                "properties" : {
                    "action": {
                    "description": "Actions in phase","type": "string"}}}},
         "synchronousactions" : {"type" : "array","items" : {"type" : "object",
                "properties" : {
                    "actionrequest": {
                    "description": "Action request id","type": "string"},
                    "actionreply": {"description": "Action reply id",
                    "type": "string"}}}},
        "features" : {"type" : "array",
           "items" : {"type" : "object",
                "properties" : {"feature": {
                    "description": "feature vector for models","type": "string"}}}}},
    "required": ["phaseid"]
    }
}
```

**Fig 21. JSON schema for the "Phase Description" of the data common specification.**

(3) Property Inference Pipeline (PIP). Table 12 shows five functions for the (4) Modeling and Simulation Pipeline (MASP). Table 13 shows five functions for the (5) Model Evaluation and Prediction pipeline (MEAPP).

The functions provide a range of capabilities from simple plotting routines to cleaning and organizing, storing and accessing data sets, and inferring properties and running simulations. Users may add other functions and continue community-based development, as these functions are not exhaustive. Each function completes one well-defined task. Many of these functions can be used in multiple contexts; functions use the pipeline as a universal interface. For example, the action progression function $h_3$ of the Data Analytics Pipeline generates a plot of the number of actions $a_i$ per player in time $\forall a_i \in A$. Also, often a function represents a category of operation; e.g., there are six different agent-based models (ABMs) under $h_1$ of the Modeling and Simulation Pipeline. Currently, functions are written in the following Programming Languages (PLs) C++, Python, and R.

```
{
    "$schema": "http://json-schema.org/draft-04/schema#",
    "title": "Players",
    "description": "Player phases parameters",
    "type": "array",
    "items" : {
    "type": "object",
    "properties": {
        "phaseid": {
            "description": "The unique identifier for a phase",
            "type": "string"
        },
        "playerid": {
            "description": "Player id",
            "type": "string"
        },
        "neighbors" : {
          "type" : "array",
            "items" : {
                "type" : "object",
                "properties" : {
                    "neighbor": {
                    "description": "Neighbor of a player",
                    "type": "string"
                    }
                }
            }
        },
        "beginparameters" : {
         "type" : "array",
            "items" : {
                "type" : "object",
                "properties" : {
                    "parameter": {
                    "description": "Beginning parameter in phase",
                    "type": "string"
                    },
                    "value": {
                    "description": "Beginning parameter value",
                    "type": "string"
                    }
                }
            }
        },
        "endparameters" : {
          "type" : "array",
            "items" : {
                "type" : "object",
                "properties" : {
                    "parameter": {
                    "description": "End parameter in phase",
                    "type": "string"
                    },
                    "value": {
                    "description": "End parameter value",
                    "type": "string"
                    }
                }
            }
        }
    },
    "required": ["playerid","phaseid"]
    }
}
```

**Fig 22. JSON schema for the "Player" of the data common specification.**

```
{
    "$schema": "http://json-schema.org/draft-04/schema#",
    "title": "Actions",
    "description": "Actions during an experiment phase",
    "type": "array",
    "items" : {
    "type": "object",
    "properties": {
        "phaseid": {
            "description": "The unique identifier for a phase",
            "type": "string"
        },
        "n": {
            "description": "Number of players in phase",
            "type": "integer"
        },
        "d": {
            "description": "Number of connections between players",
            "type": "integer"
        },
        "actionlist" : {
          "type" : "array",
          "items" : {
              "type" : "object",
              "properties" : {
                  "player1": {
                  "description": "Player that initiates the action",
                  "type": "string"
                  },
                  "player2": {
                  "description": "Player that receives the action",
                  "type": "string"
                  },
                  "actionid": {
                  "description": "action id",
                  "type": "string"
                  },
                  "playerActionSeqid": {
                  "description": "Player unique action sequence",
                  "type": "integer"
                  },
                  "timestamp": {
                   "description": "Timestamp of action",
                   "type": "number"
                  },
                  "payload": {
                  "description": "payload",
                  "type": "string"
                  }
              },
              "required": ["player1","actionid","playerActionSeqid","timestamp","payload"]
          }
        }
    },
    "required": ["phaseid","n","d"]
    }
}
```

**Fig 23. JSON schema for the "Action" of the data common specification.**

```
"experiment":
    {
    "type": "string",
    "description":"Experiments description file"
    },
```

**Fig 24. To run a pipeline (called a job), a configuration input file specifies functions and their order of execution.** This figure shows a portion of the schema for a configuration file that specifies the experiment JSON schema file location.

```
"phasedesc":
    {
    "type": "string",
    "description":"Phases description file"
    },
```

**Fig 25. To run a pipeline (called a job), a configuration input file specifies functions and their order of execution.** This Figure shows a portion of the schema for a configuration file that specifies the phase description JSON schema file location.

```
"phase":
    {
    "type": "string",
    "description":"Phases registration file"
    },
```

**Fig 26. To run a pipeline (called a job), a configuration input file specifies functions and their order of execution.** This Figure shows a portion of the schema for a configuration file that specifies the phase JSON schema file location.

```
"action":
    {
    "type": "string",
    "description":"Actions registration file"
    },
```

**Fig 27. To run a pipeline (called a job), a configuration input file specifies functions and their order of execution.** This Figure shows a portion of the schema for a configuration file that specifies the action description JSON schema file location.

```
"player":
    {
    "type": "string",
    "description":"Players parameters registration file"
    },
```

**Fig 28. To run a pipeline (called a job), a configuration input file specifies functions and their order of execution.** This Figure shows a portion of the schema for a configuration file that specifies the player description JSON schema file location.

```
"functions": {
    "type": "array",
    "description": "Functions to be run for this pipeline",
    "minItems": 1,
    "items": {
        "type": "object",
        "properties": {
            "function": {
                "description": "Pipeline functions to execute",
                "type": "string",
                "enum": ["h1","h2","h3","h4","h5"]},
            "actionId": {
                "type": "string",
                "description": "Required for h2, h3, h4, h5"},
            "windowSize": {
                "type": "integer",
                "description": "Required for h3, h4"},
            "bin": {
                "type": "integer",
                "description": "Required for h5"}
        },
        "required": ["function"]
    }
}
```

**Fig 29. To run a pipeline (called a job), a configuration input file specifies functions and their order of execution.** In this configuration file there are five possible functions that can be executed in any order. This Figure shows a portion of the schema for a configuration file that specifies how to compose and execute one or more functions of a simple pipeline. For example, here it defines that a parameter called "actionId" is only necessary for functions $h_2$ through $h_5$.

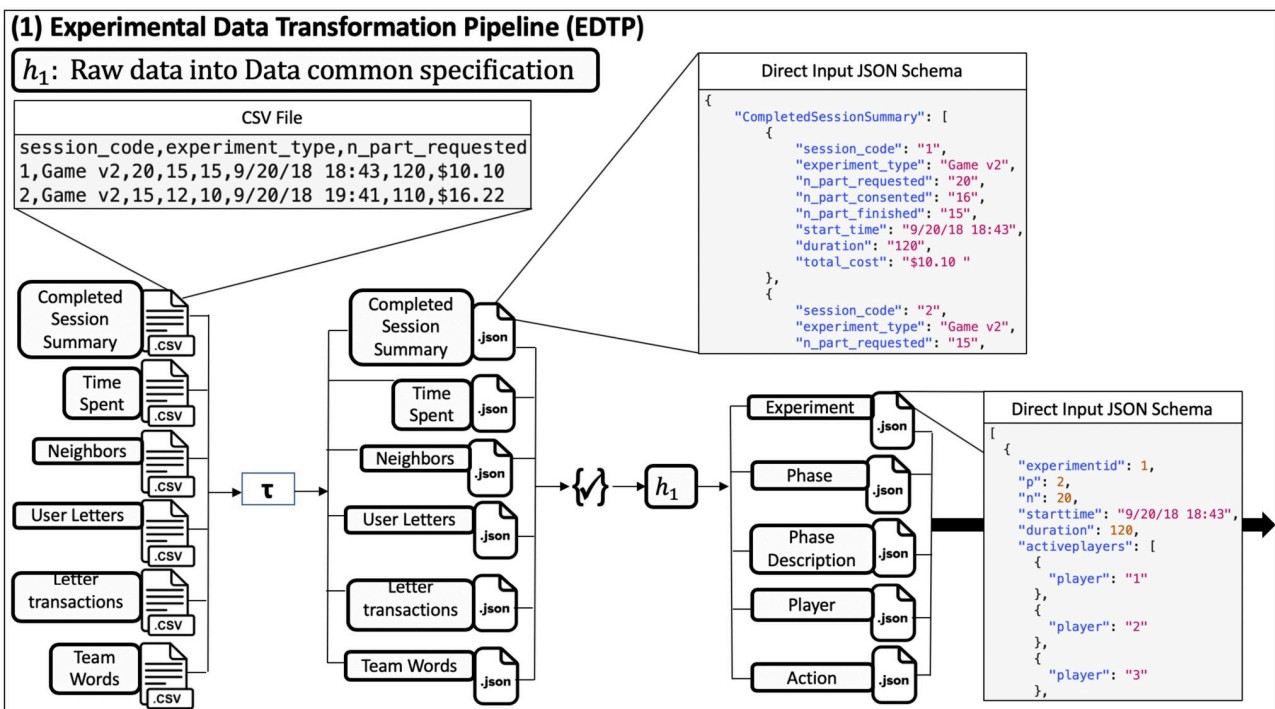

**Fig 30. This is an example of the (1) experimental data transformation pipeline execution to transform raw experimental data into the data common specification.** Here we show how function $h_1$ is executed. Here we show an input CSV file as an example for the "Completed Session Summary" input file. If necessary, file contents are transformed to obtain the direct input for a function in the correct format. Here we show how the "Completed Session Summary" CSV input file is transformed into a "Completed Session Summary" json file that becomes the input for the function. After verification of formats by the corresponding JSON schemas, the function is executed and output files are generated. Here we show the output json file for the "Experiment" data common specification.

# E Appendix: Microservices

## E.1 Characteristics

We provide a compact description of microservices [41, 42, 93, 95, 96, 108]. While there is no universally accepted of what a microservice is, we take the term to have the following features;

1. Autonomous (isolated, simple entity): a microservice is a separate entity. Although isolated services can add overhead, the resulting simplicity is worth it. This is analogous to the trade-offs between a distributed system and a shared memory system.

2. Smallness: the code for a microservice can be rewritten (constructed, tested, verified, documented) in two weeks. Often, they are less than 100 lines of code.

3. Smallness: people tend to have good intuition when a code base is too large; so sufficiently small is when this intuition does not hint at being too large.

4. Smallness: if the code base is too large to be managed by a small team, then it is not small enough.

5. Interdependence: there should be interdependence among a collection of services. As services get smaller, the benefits of interdependence increase. But smaller services create complexity (the "edges" between services). But teams should learn to handle this complexity.

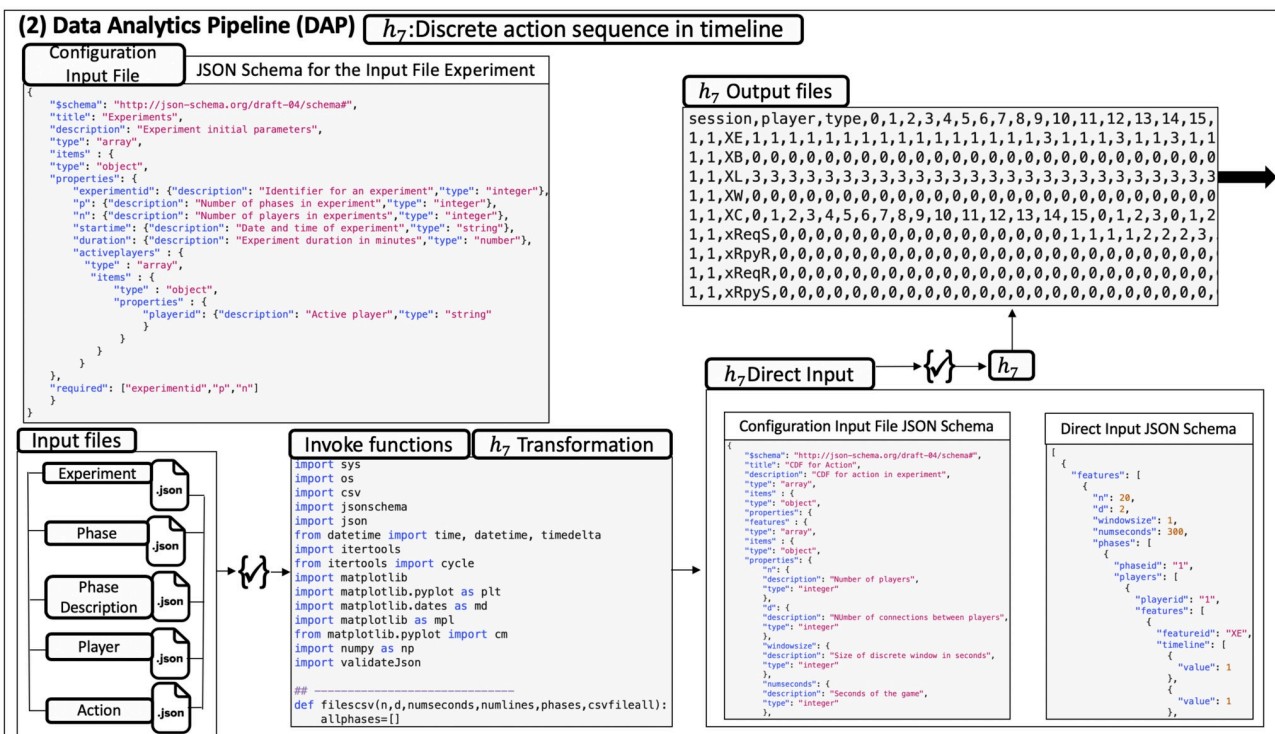

**Fig 31. This is an example of the (2) data analytics pipeline execution to analyze files of data in the common specification.** Here we show how function $h_7$ is executed. Input files are validated against their corresponding JSON schema. Here we show an example of a json schema file for the "Experiment" description input file. Fig 19 contains the whole file. After verification of formats by the corresponding JSON schemas, if necessary, file contents are transformed to obtain the direct input for a function in the correct format. After verification of formats by the corresponding JSON schemas, function $h_7$ is executed and output files are generated. In this example the output file is an input for the (3) Property Inference pipeline.

6. Communication among services: all cooperation among services is through network calls (versus direct invocation) to avoid tight coupling.

7. Change/upgrade: all microservices should be capable of changing independent of other microservices. In practice, this can be hard to do it, for example, a collection of services depend on lower level infrastructure.

8. Independent deployment: each microservice should be deployable, independent of all others.

9. Weary of Sharing Capability Between Services: the more multiple microservices share, the more services become coupled to internal representations and decreases autonomy.

**Table 9. Listing of types of functions as microservices for the (1) Experimental Data Transformation Pipeline (EDTP).**

| Pipeline: Experimental Data Transformation (EDTP) | | | | |
|---|---|---|---|---|
| # | Name | Description | Significance | Output type |
| $h_1$ | Raw data into Data common specification | Transform experimental raw data into our data common specification. | This is the only way an experiment data can go through our pipelines. | Data files |

Many functions may be considered as collections of functions because they can handle multiple types of data through the data model.

**Table 10. Listing of types of functions as microservices for the (2) Data Analytics Pipeline (DAP).**

**Pipeline: Data Analytics (DAP)**

| # | Name | Description | Significance | Output type |
|---|------|-------------|--------------|-------------|
| $h_1$ | Player interactions | Generate a timeline of individual and between-players actions. Each player represents a lane. Each action has a unique color. | Detect common patterns between players and actions. | Visualization |
| $h_2$ | Timestamp Delta between related actions | Construct a visualization of the timestamp delta between related actions. A request action has a correspondent receive action. Each request action represents a lane, a horizontal line represents the length of time it takes to receive a requested action. | Detect bursts in types of actions. Detect time patterns in types of actions. | Visualization |
| $h_3$ | Action progression | Generate a cumulative distribution plot for an action, by player. | Show how an action progresses in time during an experiment phase. | Data files and plot |
| $h_4$ | Average action | Generate plot of the average number of actions between players in a window size $s$. | Show how an average action progresses in time between experiments phases. | Data files and plot |
| $h_5$ | Action histogram | Generate a histogram of timestamps of an action. | Compare histograms among all experiment phases. | Data files and plot |
| $h_6$ | Histogram of related actions | Generate a histogram of timestamp delta between related actions. | Compare histograms between all experiments phases. | Data files and plot |
| $h_7$ | Discrete action sequence in timeline | Generate a discrete-time action sequence by phase. Each action, from the action set $A$ has a unique id definition. | Generate input for the Property Inference pipeline. | Time series data files |
| $h_8$ | Summary of actions. | Generate for each unique action the number of occurrences at the end of a phase, and the number of occurrences at the end of all experiments in the pipeline run. | Compare action occurrences among all experiments. | Data files |
| $h_9$ | Player categories. | Categorize players by performance in each action. | Analyze player performance by clustering them in categories. | Data files and plot |
| $h_{10}$ | Actions heat-map. | Generate heat-map by player for actions in a phase. | Analyze player performance by a heat-map visualization. | Data files and plot |
| $h_{11}$ | Summary of related actions. | Generate a summary at the end of a phase with the possible actions between neighbors and the occurred actions. | Compare related action occurrences among all experiments. | Data files |
| $h_{12}$ | Distance between actions. | Generate a file with distance between two actions. The distance has to be provided by the analyst (e.g, for the action of forming a word, the Levenshtein distance between two words formed). | Compare action characteristics in an experiment. | Data files |
| $h_{13}$ | Rank of actions. | Generate a file with rank of an action. The rank has to be provided by the analyst (e.g, for the action of requesting a letter, the letter rank comes from a specified list). | Compare action characteristics in an experiment. | Data files |
| $h_{14}$ | Score of actions. | Generate a file with a score of an action. The method to calculate the score has to be provided by the analyst (e.g, for the action of forming a word, the scrabble score for a word formed). | Compare action characteristics in an experiment. | Data files |

Many functions may be considered as collections of functions because they can handle multiple types of data through the data model.

**Table 11. Listing of types of functions as microservices for the (3) Property Inference Pipeline (PIP).**

**Pipeline: Property Inference (PIP)**

| # | Name | Description | Significance | Output type |
|---|------|-------------|--------------|-------------|
| $h_1$ | Properties for Markovian transition matrix. | Use of the sequences of discrete actions to generate the probability of transition from an action $a_i$ to an action $a_j$ as measured in the experiment data. | Generates the properties for a Markovian transition matrix. | Data files |
| $h_2$ | Properties for an adapted CRF model | Use of the sequences of discrete actions to generate a derived feature vector accounting for history effects, where the vector corresponds to the discrete-time sequences from the Data Analytics $h_7$ output. | Generates properties for an adapted conditional random fields (CRF) model. | Data files |
| $h_3$ | Coefficients in a hierarchical model | Generalize the model to take the number of neighbors into consideration, and also digest the additional experiment data where player degree increases or decreases. | Generate coefficients in a hierarchical model to augment the CRF model. | Data files |
| $h_4$ | Multilinear regression model | Construct multilinear regression model on action set $A$. | Generate structure of the model and parameter values | Data files |

Many functions may be considered as collections of functions because they can handle multiple types of data through the data model.

**Table 12. Listing of types of functions as microservices for the (4) Modeling and Simulation Pipeline (MASP).**

**Pipeline: Modeling and Simulation (MASP)**

| # | Name | Description | Significance | Output type |
|---|------|-------------|--------------|-------------|
| $h_1$ | Agent based model (ABM) | Execute agent based simulation models. Currently, six different models (stationary, dynamic conditional random fields (CRF). | Generate Agent Based Model Simulations outputs for self-consistency checks and predictions. | Data files |
| $h_2$ | Statistical regression | Compute a relation between selected and observed values. | Predict most probable value of the observed values for any selected values. | Data files |
| $h_3$ | Statistical regression | Regression equation that uses results from Phase 1 ABM to predict the Phase 2 DIFI. | Predict the Phase 2 DIFI (i.e., DIFI2) score per player. | Data files |
| $h_4$ | Statistical regression | Regression equation that uses results from ABM Phases to predict the Publics Good Game Contributions in the corresponding Phase. | Predict the Publics Good Game Contributions per player. | Data files |
| $h_5$ | Component model prediction | Execute agent based simulation component models to compare outputs with real actions from a real experiment. | Compare outputs between models. | Data files |

Many functions may be considered as collections of functions because they can handle multiple types of data through the data model.

**Table 13. Listing of types of functions as microservices for the (5) Model Evaluation and Prediction pipeline (MEAPP).**

**Pipeline: Model Evaluation and Prediction (MEAPP)**

| # | Name | Description | Significance | Output type |
|---|------|-------------|--------------|-------------|
| $h_1$ | Model Validation | Compares experiment outputs with simulation outputs. | Demonstrate that the model is a reasonable representation of the actual system. | Data files and plot |
| $h_2$ | Model Prediction | Generates statistical models to predict outcomes. | Forecast outcomes in an experiment. | Data files and plot |
| $h_3$ | Model Fusion | Generates model to predict outcomes by combining outputs from different models. | Predict the Phase 2 DIFI (i.e., DIFI2) score per player. | Data files |
| $h_4$ | Model Evaluation | Generates R-squared values by comparing experiment outputs with simulation outputs. | R-squared is a statistical measure of how close the data are to the fitted regression line. | Data files |
| $h_5$ | Cross-Validation | The original experiment sample is randomly partitioned into k equal size subsamples. Of the k subsamples, a single subsample is retained as the validation data for testing the model, and the remaining k-1 subsamples are used as training data. The cross-validation process is then repeated k times, with each of the k subsamples used exactly once as the validation data. | Demonstrate that the model is a reasonable representation of the actual system. All observations are used for both training and validation, and each observation is used for validation exactly once. | Data files |

Many functions may be considered as collections of functions because they can handle multiple types of data through the data model.

10. APIs (Application Programming Interfaces): specify/select/prefer technology-agnostic APIs so that the services are not constrained by technology. Achieve decoupling: the success of the "Change/upgrade" feature is an evaluation of decoupling success. Decoupling also requires good models.

## E.2 Benefits

Many of the benefits of microservices stem from their isolated, independent scope [41, 42, 93, 95, 96, 108].

1. Technology heterogeneity, including technology stacks, across microservices.

2. Technology changeout.

3. Technology evaluation in a controlled, limited way.

4. Easier to isolate problems and failures.

5. Scale-up can be focused to particular services. So, too, with on-demand provisioning.

6. Deployments/redeployments can be isolated to particular microservices. Smaller increments of (re)deployment means reducing the possibility of adverse ripple effects.

7. Improvements/new versions are eased in and old versions are eased out.

8. Smaller services translates to smaller teams.

9. Composability, reuse.

10. Choices to throw away code are made more easily (less ownership, less cost of construction).

11. Easier unit testing (generating, executing, and interpreting tests). For example, there are fewer paths through the code.

### E.3 Microservices as a type of service oriented architecture

Pipelines are intimately tied to microservices. While microservices may be used individually, typically, the small scope and limited features (or one feature) per service implies that they must be composed to accomplish many tasks. This composition can be accomplished with pipelines. This is not necessarily true with larger, more monolithic service oriented architectures (SOAs): these may provide broader-scope services within one module.

Microservices are one type of service oriented architecture (SOA). One example of the difference between the two is that microservices generally tend to avoid shared libraries that are used across microservices. This is because use of shared libraries means increased coupling of services. Based on the authors' experiences, this difference between microservices and SOAs in general is analogous to the difference between shared memory multi-process systems versus distributed systems, as described next.

By multi-process *shared memory* systems, we mean a software system that is composed of multiple processes that run asynchronously and use shared memory to exchange information (e.g., no message passing). In this environment, the processes are tightly coupled because if one process requires changes in shared storage structures, these will affect all other processes that use those storage structures. That is, the software for these other processes needs to be changed, too, leading to increased maintenance. Hence, there are a lot of interdependencies. However, in an asynchronous *distributed system*, each process has its own storage structures and memory, so that changes in storage structures for one process has no effect on other processes. While it is the case that additional infrastructure is required for distributed systems (e.g., for message passing), this additional requirement is offset by the autonomy realized for each process. The analogy here is that a multi-process shared memory system is a classic SOA, while microservices are the distributed system.

## F Appendix: Case studies

### F.1 Study 2: Data model for online experiment in [3]

**F.1.1 Overview.** In [3], the effects of network structure on complex contagion diffusion are studied by the spread of health behavior through networked online communities. We represent this experiment with the data model from Section 3. Each experiment, *exp_id*, consists

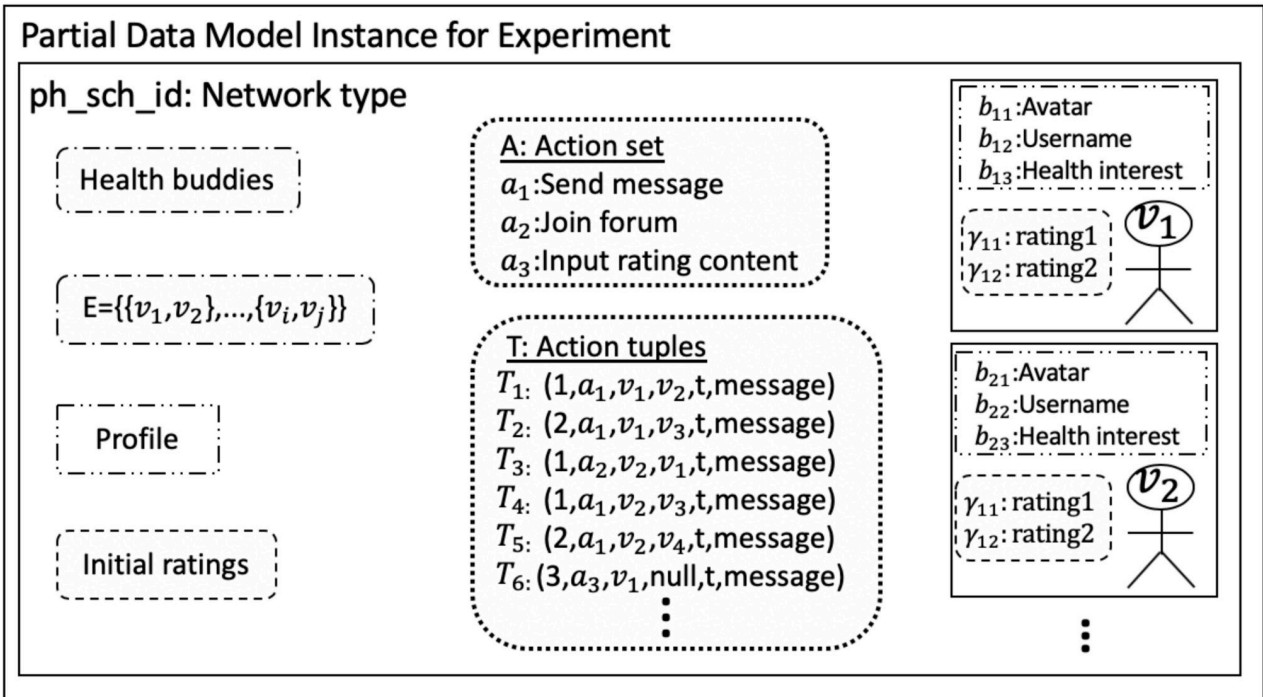

**Fig 32. Elements of the data model (Table 3), for the online social network experiment in [3].**

of two independent phases ($n_p$ = 2), one with $G(V', E')$ being a clustered-lattice network and another $H(V'', E'')$ being a random network. $V = V' \cup V''$ is the set of all players with player $v_i \in V$, and $1 \leq i \leq n$. There are $n/2$ players in each of the two networks, assigned randomly. $\Gamma_i$ contains variables for $v_i$'s profile (i.e., avatar, username, health interests), ratings of the forum content, and the state of $v_i$ in time, i.e., whether $v_i$ has joined the forum. The meaning of an edge is $\lambda$ = communication channel between pairs of subjects. $B_i^v$ contains initial conditions for the game, including values for the elements of $\Gamma_i$. The set of actions is $A = \{a_1, a_2, a_3\}$, where $a_1$ is "send a message" to encourage a neighbor to adopt a health related behavior; $a_2$ is "join forum" which notifies a participant every time a neighbor adopts the behavior; and $a_3$ is "input rating content" in the forum. Fig 32 shows many of these variables, and examples of action tuples. Here we also provide detail of the action sequence from Fig 32. In $T_1$, $v_1$ sends a message to $v_2$, then in $T_2$, $v_1$ sends a message to $v_3$. All these are signals from $v_1$ to encourage health buddies to join the forum. In $T_3$, $v_2$ decides to join because of $v_1$'s message. This is why the unique identifier $\sigma_i$ for the action sequence is the same as in $T_1$. After this, the news is propagated to $v_2$'s health buddy $v_3$ in $T_4$. $v_2$ sends a message to $v_4$ in $T_5$. In $T_6$, $v_1$'s inputs rating content to the forum. This data model instance, coupled with a GDS formulation (not shown), means that the experimental data can be analyzed (and modeled) with the pipeline system.

**F.1.2 Formal data model.** Table 14 details the online social network experiment in [3], defined with our data model. We define one experiment with two independent phases, one with a clustered-lattice network and another with a random network. Each has a population size n = 98 and number of health buddies per person d = 6.

Fig 33 shows the model of Table 14 translated into a entity-relationship diagram in unified modeling language (UML) form. This data model instance, that represents an experiment instance, means that the experimental data can be analyzed (and modeled) with the pipeline system. We can perform similar mappings for other social experiments [1, 9, 61].

**Table 14. Online social network experiment in [3], defined with our data model.**

| # | Parameter | Description |
|---|---|---|
| **Experiment Schema** | | |
| 1 | $exp\_id = 1$ | Experiment id for an experiment. |
| 2 | $n_p = 2$ | Number of phases in the experiment. |
| 3 | $n = 196$ | The number of unique players over all phases. |
| 4 | $t\_begin$ | Timestamp of experiment beginning. |
| 5 | $t\_end$ | Timestamp of experiment ending. |
| 6 | $V$ | $V = \{v_1, \ldots, v_{196}\}$, set of players over all phases. |
| **Phase Schema** | | |
| 1 | $ph\_sch\_id = 1$ | Id for phase schema. |
| 2 | $i_{n_p} = 1$ | Element of the sequence of phases of the experiment. |
| 3 | $t\_ph\_begin$ | Timestamp of phase beginning. |
| 4 | $t_p = 13$ | Number of time increments in the phase. |
| 5 | $u_p = days$ | Time unit of one time increment. |
| 6 | $G(V', E')$ | Clustered-lattice network, node set $V' = \{v_1, \ldots, v_{98}\}$ and edge set $E' = \{e_1, \ldots, e_{294}\}$, where the number of health buddies each person has is 6. |
| 7 | $\lambda$ | $\lambda$ = communication channel between health buddies. $\lambda \in \Lambda$ |
| 8 | $\Gamma$ | $\Gamma_j(t) = (\gamma_{j1}(t), \gamma_{j2}(t), \ldots, \gamma_{j.\eta_v}(t))$ is the sequence of $\eta_v$ attributes for $v_j \in V'$. $\eta_v$ = # of initial ratings in the forum to provide content for the early adopters. |
| 10 | $B^v$ | $B_j^v = (avatar_{j1}, username_{j2}, health\_interest_{j3}, \ldots)$. |
| 12 | $A$ | $A = \{a_1, a_2, a_3\}$ where $a_1$ is send message, $a_2$ is join forum, and $a_3$ is input rating content. |
| 13 | $T$ | $T_1 = (1, a_1, v_1, v_2, t, message)$. $v_1$ "sends message" to $v2$. <br> $T_2 = (2, a_1, v_1, v_3, t, message)$. $v_1$ "sends message" to $v3$. <br> $T_3 = (1, a_2, v_2, v_1, t, message)$. $v_1$ "joins forum" after $T_1$. <br> $T_4 = (1, a_1, v_2, v_3, t, message)$. $v_2$ "sends message" to $v3$. <br> $T_5 = (2, a_1, v_2, v_4, t, message)$. $v_2$ "sends message" to $v4$. <br> $T_6 = (3, a_3, v_1, null, t, message)$. $v_1$ "inputs rating content" to forum. <br> . . . |
| **Phase Schema** | | |
| 1 | $ph\_sch\_id = 2$ | Id for phase schema. |
| 2 | $i_{n_p} = 2$ | Element of the sequence of phases of the experiment. |
| 3 | $t\_ph\_begin$ | Timestamp of phase beginning. |
| 4 | $t_p = 13$ | Number of time increments in the phase. |
| 5 | $u_p = days$ | Time unit of one time increment. |
| 6 | $H(V'', E'')$ | Random network, node set $V' = \{v_{99}, \ldots, v_{196}\}$ and edge set $E' = \{e_1, \ldots, e_{294}\}$, where the number of health buddies each person has is 6. |
| 7 | $\lambda$ | $\lambda$ = communication channel between health buddies. $\lambda \in \Lambda$ |
| 8 | $\Gamma$ | $\Gamma_j(t) = (\gamma_{j1}(t), \gamma_{j2}(t), \ldots, \gamma_{j.\eta_v}(t))$ is the sequence of $\eta_v$ attributes for $v_j \in V'$. $\eta_v$ = # of initial ratings in the forum to provide content for the early adopters. |
| 10 | $B^v$ | $B_j^v = (avatar_j, username_j, health\_interest_j, \ldots)$. |
| 12 | $A$ | $A = \{send\_message, join\_forum, input\_rating\_content.\}$. |
| 13 | $T$ | $T_1 = (1, a_1, v_1, v_2, t, message)$. $v_1$ "sends message" to $v2$. <br> $T_2 = (1, a_2, v_2, v_1, t, message)$. $v_1$ "joins forum" after $T_1$. <br> . . . |

One experiment has two independent phases, one with a clustered-lattice network and another with a random network; each with population size n = 98 and number of health buddies per person d = 6.

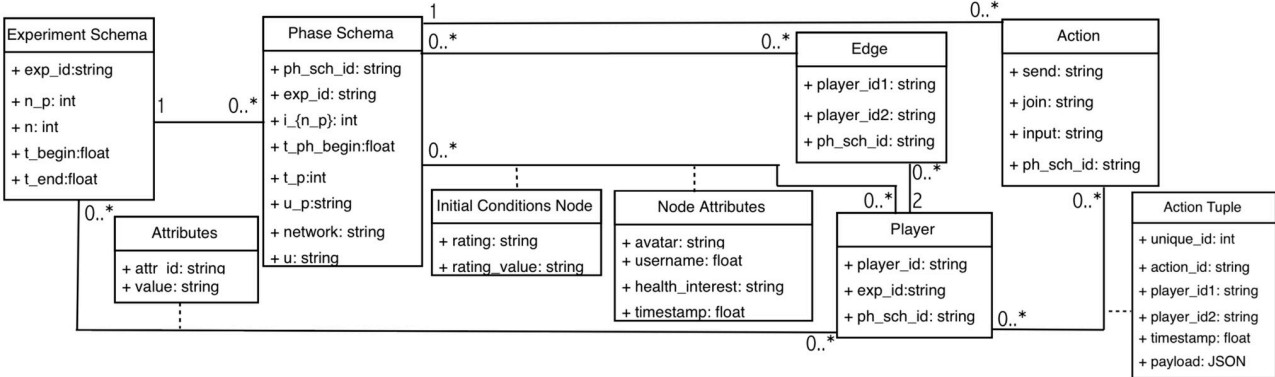

**Fig 33. Data model of Table 14 translated into a entity-relationship diagram in Unified Modeling Language (UML) form.**

**F.1.3 Formal GDS model.**   The GDS model for this system and experiments is that given in Section 4.2.

## F.2 Study 3: Data model for a simulation study in [44]

**F.2.1 Overview.**   In this case study, we evaluate research that is purely simulation-based. We cast their problem in terms of our data model. With this mapping, we then can reason that if we performed experiments according to this data model, we would have a correspondence between those experiments and the simulation system. Hence, in a sense, this case study demonstrates a process of going from modeling to experiments. Another note is that even with simulation models and no experiments, we can still use our pipeline system.

The model in [44] investigates how the structure of communication networks among actors can affect system-level performance. This is an agent-based computer simulation model of explore-exploit tradeoffs, with information sharing. [44] produces an arbitrarily large number of statistically identical "problem" for the simulated agents to solve (explore). Also, the less successful emulate the more successful (exploit). They state that solutions involve the conjunction of multiple activities, in which the impact of one dimension on performance is contingent on the value of other dimensions. For example, activities A, B, and C each actually hurt performance unless all are performed simultaneously, in which case performance improves dramatically. These are defined as synergies, and the presence of such synergies produces local optima.

**F.2.2 Formal data model.**   Table 15 details the model in [44], defined with our data model. We define one experiment with one phase, with a population of 100, 20 human activities, and 5 synergies (i.e., activities that performed simultaneously improves dramatically the activity performance). Here we also provide an example of an action sequence. In $T_1$, $v_1$ posts a solution, then in $T_2$, $v_2$ posts a solution. All these are signals from $v_1$ to encourage health buddies to join the forum. In $T_3$, $v_3$ evaluates $v_1$ solution. In $T_4$ $v_3$ copies solution from $v_1$. The payload will have the information of how accurate agents copy the solution from other, (i.e.) if it was "mimic" or "adapt".

Fig 34 shows the model of Table 15 translated into a entity-relationship diagram in unified modeling language (UML) form.

This data model instance, that represents a modeling instance, means that the computational modeling results can be analyzed with the pipeline system.

**Table 15. How the structure of communication networks among actors can affect system-level performance is studied in [44].**

| # | Parameter | Description |
|---|---|---|
| **Experiment Schema** | | |
| 1 | $exp\_id = 1$ | Experiment id for an experiment. |
| 2 | $n_p = 1$ | Number of phases in the experiment. |
| 3 | $n = 100$ | The number of unique players over all phases. |
| 4 | $t\_begin$ | Timestamp of experiment beginning. |
| 5 | $t\_end$ | Timestamp of experiment ending. |
| 6 | $V$ | $V = \{v_1, \ldots, v_{100}\}$, set of players over all phases. |
| **Phase Schema** | | |
| 1 | $ph\_sch\_id = 1$ | Id for phase schema. |
| 2 | $i_{n_p} = 1$ | Element of the sequence of phases of the experiment. |
| 3 | $t\_ph\_begin$ | Timestamp of phase beginning. |
| 4 | $t_p = converge$ | The phase runs until it converges on a single solution. |
| 5 | $u_p = seconds$ | Time unit of one time increment. |
| 6 | $G(V', E')$ | Linear network, node set $V' = \{v_1, \ldots, v_{100}\}$ and edge set $E' = \{e_1, \ldots, e_{98}\}$. |
| 7 | $\lambda$ | $\lambda$ = influence channel between neighbors. $\lambda \in \Lambda$ |
| 8 | $\Gamma$ | $\Gamma_j(t) = (\text{density}_j(t), \text{average\_path\_length}_j(t), \text{score}_j(t))$ |
| 10 | $B^v$ | $B^v_j = (\text{human\_activities}_j, \text{synergies}_j, \ldots)$. |
| 12 | $A$ | $A = \{\text{post\_solution, evaluate, copy\_solution.}\}$. |
| 13 | $T$ | $T_1 = (1, a_1, v_1, null, t, solution)$. $v_1$ "posts solution". $T_2 = (1, a_1, v_2, null, t, solution)$. $v_2$ "posts solution". $T_3 = (1, a_2, v_3, v_1, t, solution)$. $v_3$ "evaluates" $v_1$ solution. $T_4 = (1, a_3, v_3, v_1, t, solution)$. $v_3$ "copies solution" from $v1$. $\ldots$ |

Here we define this model with our data model.

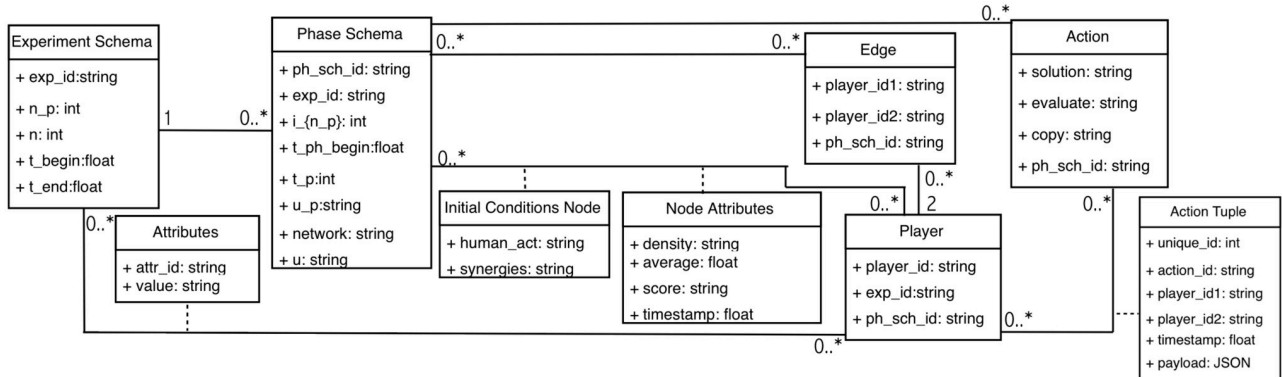

**Fig 34. Data model of Table 15 translated into a entity-relationship diagram in Unified Modeling Language (UML) form.**

## Supporting information

**S1 File.**
(ZIP)

## Author Contributions

**Conceptualization:** Xinwei Deng, Chris J. Kuhlman.

**Data curation:** Vanessa Cedeno-Mieles, Zhihao Hu, Xinwei Deng, Brian J. Goode, Parang Saraf, Nathan Self.

**Formal analysis:** Vanessa Cedeno-Mieles, Zhihao Hu, Yihui Ren, Xinwei Deng, Chris J. Kuhlman.

**Funding acquisition:** Noshir Contractor, Joshua M. Epstein, Brian J. Goode, Chris J. Kuhlman, Michael Macy, Madhav V. Marathe, Naren Ramakrishnan.

**Investigation:** Vanessa Cedeno-Mieles, Zhihao Hu, Yihui Ren, Xinwei Deng, Chris J. Kuhlman.

**Methodology:** Vanessa Cedeno-Mieles, Zhihao Hu, Xinwei Deng, Chris J. Kuhlman.

**Resources:** Madhav V. Marathe, Naren Ramakrishnan.

**Software:** Vanessa Cedeno-Mieles, Zhihao Hu, Xinwei Deng, Saliya Ekanayake, Chris J. Kuhlman, Dustin Machi.

**Supervision:** Yihui Ren, Xinwei Deng, Brian J. Goode, Chris J. Kuhlman.

**Validation:** Noshir Contractor, Saliya Ekanayake, Joshua M. Epstein, Brian J. Goode, Gizem Korkmaz, Dustin Machi, Michael Macy, Madhav V. Marathe, Naren Ramakrishnan, Parang Saraf, Nathan Self.

**Visualization:** Vanessa Cedeno-Mieles, Zhihao Hu.

**Writing – original draft:** Vanessa Cedeno-Mieles, Zhihao Hu, Xinwei Deng, Chris J. Kuhlman.

**Writing – review & editing:** Vanessa Cedeno-Mieles, Zhihao Hu, Xinwei Deng, Chris J. Kuhlman.

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
