## [Decision Letter · Decision Letter 0]

1 Sep 2020

PONE-D-20-13649

Data Analysis and Modeling Pipelines for Controlled Networked Social Science Experiments

PLOS ONE

Dear Dr. Cedeno-Mieles,

Thank you for submitting your manuscript to PLOS ONE. After careful consideration, we feel that it has merit but does not fully meet PLOS ONE’s publication criteria as it currently stands. Therefore, we invite you to submit a revised version of the manuscript that addresses the points raised during the review process.

We look forward to receiving your revised manuscript.

Kind regards,

Ning Cai, Ph.D.

Academic Editor

PLOS ONE

Additional Editor Comments:

The four reviewers have approached a consensus as their impressions to the submission: 1) the topic is interesting; 2) however, the readability is flawed and requires significant improvement.

Journal Requirements:

"This work has been partially supported by DARPA Cooperative Agreement

D17AC00003 (NGS2), DTRA CNIMS (Contract HDTRA1-11-D-0016- 0001), NSF

DIBBS Grant ACI-1443054, NSF BIG DATA Grant IIS-1633028, NSFCRISP 2.0 Grant

1832587, NSF Grants DGE-1545362 and IIS-1633363, and ARL Grant

W911NF-17-1-0021. The U.S. Government is authorized to reproduce and distribute

reprints for Governmental purposes notwithstanding any copyright annotation thereon.

Disclaimer: The views and conclusions contained herein are those of the authors and

should not be interpreted as necessarily representing the official policies or endorsements,

either expressed or implied, of DARPA, DTRA, NSF, ARL, or the U.S. Government. "

Reviewers' comments:

Reviewer's Responses to Questions

**Comments to the Author**

1. Is the manuscript technically sound, and do the data support the conclusions?

Reviewer #1: Yes

Reviewer #2: Partly

Reviewer #3: Yes

Reviewer #4: Partly

2. Has the statistical analysis been performed appropriately and rigorously? 

Reviewer #1: Yes

Reviewer #2: N/A

Reviewer #3: Yes

Reviewer #4: I Don't Know

3. Have the authors made all data underlying the findings in their manuscript fully available?

Reviewer #1: Yes

Reviewer #2: Yes

Reviewer #3: Yes

Reviewer #4: Yes

4. Is the manuscript presented in an intelligible fashion and written in standard English?

Reviewer #1: Yes

Reviewer #2: Yes

Reviewer #3: Yes

Reviewer #4: No

5. Review Comments to the Author

Reviewer #1: (1) Everything is ok except that the abstract section is tediously long.

(2) Check all the equations and mathematical symbols for ensuring the correctness.

(3) Some newly published references should be included.

Reviewer #2: This paper, “Data Analysis and Modeling Pipelines for Controlled Networked Social Science Experiments,” shows a systematic research framework to systematically integrate human-subject experiments, agent-based modeling, and simulations in social sciences. I appreciate the authors’ approach to use the computer-science idea of “pipelines” (more generally, the concept of encapsulation) to make social-science methods more reusable and thus more efficient and verifiable. I think, however, the authors could do better to clarify the original contribution of this paper as a report of scientific research.

As the authors mentioned in the paper, they have shown the original idea in the paper [35] (and used the developed software in [7, 23, 45]). Actually, I found several figures (Figures 2 and 3) and descriptions of this paper are also shown in [35], but there is no citation in the figure captions and the manuscript. Although this paper has Section 1.7 that refers to the extensions from the original paper [35], I would suggest that the authors should focus on meaningful advancements for scientific contributions from [35]. The authors stressed this paper’s originality at p. 9: “The purpose of this paper is to describe the software system used in those studies ([7, 23, 45]), and that can be used for other networked experiments.” However, I do not see just detail technical specification as a report of scientific research. As a scientific report, the authors should show that the technical specification and the extensions of [35] address methodological challenges in social sciences.

In this regard, I suggest that the authors verify the effect of the developed framework more carefully. The authors pointed out that the experiments, modeling, and simulations in social sciences “are often conducted a heuristic manner by individual scientists, leading to inefficiencies, duplication and an overall decrease in human productivity. Automating these steps can lead not only to improved productivity, but also to improved scalability and reproductivity.” (p.3) Thus, the authors have developed the software to automate these method steps using the pipeline framework. However, I could not find the evaluation whether their software addresses the challenge of inefficiencies, duplication, productivity, and reproductivity in social-science researches. For evaluation, the authors refer to three case studies (pp. 8-9). However, 1) only the case study 1 has the full process of the proposed system; 2) the single study did neither show any tests in terms of the above challenges nor evaluate the general-purpose versatility of the framework; 3) I do not think just 15-25 subjects are enough to evaluate social-science research methods at scale (p. 9); and 4) the authors used “millions of artificial agents” to complement the scarcity of human subjects (p.9), and that means they used agent-based simulations for evaluation. A research method cannot be evaluated using a component of the research method.

I am also concerned about the research framework itself. Social-science researchers often use lab experiments to understand some causality in social phenomena with randomized controlled settings. Agent-based models and simulations help them to confirm the mechanisms that they discover with experiments or to specify the key factors that they should explore with experiments. From this perspective, the systematic integration of experiments, models, and simulations is reasonable. However, the robust causality of lab experiments come from the careful design to compare the experimental treatments with control. I suggest that the authors should clarify how the technical specifications can increase or at least maintain the strength of lab experiments.

Another concern relates to the closed loop system shown in Figure 2. Such a feedback system, which is often used in engineering, works well to optimize something under the given boundary condition. However, boundary conditions are not always obvious in social-science researches, especially in the exploration phase. With wrong boundary conditions or no external assessment components, the system can be led to an improper sub-optimal status. Thus, lab experiments often need external validity. However, I was not able to find such a function or component in the pipeline specifications. On the other hand, in the testing phase after exploration, I’m not sure why we need to use the generalized pipeline system.

Furthermore, social-science researches including NESS experiments have specific difficulties to deal with real humans in general. For example, research results might differ between different group of people. Even within the same person, her behavior and decision making might change over time. Thus, for example, the second experiment several months after the first one might give different results because of the drifting. I am not sure how the proposed framework can address such a challenge. I suggest that the authors should specify how the research framework can deal with the difficulties to deal with the variation of social behaviors over population and over time with the real-world content because this study aims to give a general framework for social sciences and not for computer science.

Reviewer #3: I appreciate the sophisticated design reflected in this work. It's a thoughtful system. However, it feels a lot more like a white paper or technical report than a scientific paper proving this pipeline is the correct solution, as a consequence, I believe there's an opportunity to reframe the paper as a methods only contribution and to identify upfront what that implies about external adoption of these tools. In particular, I believe this requires isolating the design and goals of the system from the implementation, e.g., much of the rhetoric on micro services is only relevant for certain kinds of empirical and modeling problems but the broader implication of the architecture is quite generally useful.

I also find that this work is in discussion with other innovations in online lab experiment design, which I suggest considering (as a potential context for the design of related systems — e.g., arxiv.org/abs/2006.11398).

As a minor extension to this point, I think the empirical context where these tools are important can be outlined in more detail and with more consistent examples, giving a reader a regular sense of the decisions made in the modeling system design.

An exciting extension to the current work would be formalizing the use of this method with adaptive experimentation techniques (e.g., ax.dev).

As a consequence of the points above, I'm suggesting a minor revision, as I don't believe these kinds of changes will take substantially more work. That said, I don't think these issues are trivial. Without some reframing, I don't think this will be suitable for publication as a scientific report.

Minor fixes:

THere're several typos, one that I recall is "For each of theses topics" in the caption for Table 2 should be "these"

Reviewer #4: Thank you for giving me the opportunity to review your paper! I have a particular interest in how simulation and experiment can be used together to develop and test social theory, and have grappled with some of the same issues that you raise as I bring together experiment and simulation in my own work. There is a lot of room for methodological improvements to be made in this area, and I am encouraged to see such a competent list of authors thinking seriously about how to do so. =)

I am afraid that I found the paper somewhat difficult to understand, and so my comments on the technical work you describe may miss the mark. I beg your pardon should that be the case!

This paper introduces a set of software tools to help social scientists integrate simulation of social theories with networked experiments designed to evaluate those theories. Bringing these two components together into a single computational framework speeds the researcher’s ability to cycle through stages of theory development and testing, and improves our ability to learn about the world. The paper proposes a standard language for describing simulations and experiments, such that any experiment/simulation combination that takes advantage of this “abstract data model” should be able to take advantage of the tools proposed here to reduce the overhead work associated with organizing a simulation/experiment workflow. By formalizing the theory building and testing process, the tools should also improve the replicability of research by helping to document the steps that the researcher followed in their research process.

There is a phrase amongst designers that a tool (product, etc.) should be “useful, usable, and used”. These tools jump right out at me as being very useful, as one of the major challenges to social scientific theorizing is the difficulty of empirical verification, and anything that improves that workflow ought to be widely adopted. However, I find that I am unable to assess whether the tools will be “usable and used”.

Writing a “Tools” paper is really hard. The author of any tools paper knows so much about the technical implementation of the tool, and the difficulty of constructing it, they naturally want to share how those challenges were overcome. If the paper is written for an audience of potential *users*, however, it doesn’t benefit the reader to know what you have done as authors and tool developers. The reader wants to know what *she* can do because of your contributions. (Or at least, I do!) In my reading of the paper, I am unsure how to apply the tools you have developed in my own research - and this is the type of research I do every day.

The reason for my confusion may be that much of the paper deals with the technical implementation of pipelines, h-functions, microservices, and other pieces of machinery; and yet I don’t understand how these elements relate to me as a social scientist. I’m not sure if the solution to this is to provide more exposition (so that I know why I need the technical details), or to switch the focus away from implementation towards use. For example, when I reach line #615 “Let P be a collection of pipelines, with pipeline P∈P represented as P(Q,ˆQ,SID,S,TID,T,H).”, I am not sure why I need to be considering sets of pipelines at this point in the paper, or how doing so will improve my ability to conduct social science. That is not to say that a formal model of the pipeline is not important to developing the tool, or that presenting it in the paper is unhelpful, merely that it is not relevant to a social scientist trying to decide whether and how to use the authors’ tools. I hate to say it, but I had trouble following the thread of the paper and understanding why each section was included and what it had to teach me.

It could be that I am assuming the wrong audience for the paper. Rather than social scientists who might use the tool, this could be for other tool designers working to construct similar types of tools, in which case the level of technical detail may be appropriate.

However, at this point I am unable to confidently endorse the technical contribution of the paper for a social science audience. This is not because I have found errors or problems in the underlying tools you have developed, but because I can’t evaluate the work with enough understanding to know if errors are present or not. This is difficult for me to admit as a reviewer - if I was more knowledgeable or clever, would I be able to give a better review? Regardless, I do feel that if this paper is written for an audience of tool users, then each piece of the paper should be comprehensible to that user.

It is difficult for me to make this recommendation, as I can see how much work you have done to develop these tools, and the significant ‘upside potential’ if such tools were in wide use. However, it would truly be unfair to the work that you have done in creating these tools if they were not used, and were not able to attract a community that could push their development forward.

If you are committed to supporting these tools into the future, you will no doubt put significant effort into describing the use cases for these tools, and creating user-friendly documentation with concrete examples that users can work through themselves. This material would be inordinately helpful in a paper designed to introduce the tool, such as this one. If I were to make a suggestion as to how this paper could be more tightly integrated, it would be to start with a very concrete case study, in which a named researcher asked a real question about social reality, and then walk through her workflow. At each stage in the workflow, highlight the challenges she faces and why the existing solutions fail. Then, introduce your tool as a way to solve those concrete issues. Use language that acknowledges the user (for example, from the caption for fig 26 “To run a pipeline (called a job), a configuration input file specifies functions and their order of execution”, instead say “In the configuration file, specify which functions should be run, and in what order.”) The passive voice is just impeding clarity. Push the proofs and the technical details to the appendix, and tell me what I can do with the tool, why it’s important, and how to do it. If I don’t have a really good grasp of what is happening, the details of the machinery are meaningless.

Don’t be too discouraged that I’m recommending against publishing the paper as currently written. I am suggesting that the editor be open to receiving another paper from you on this topic that reads more clearly. No rejections are fun, and I know that I am suggesting a substantial rewrite of the paper. But (having done this myself) I don’t think it will be that much more work than you will need to do anyway in creating user documentation, in whatever form that takes. If this work is important to you going forwards, it’s worth creating a community of users around, and to do that, you need the “story” surrounding the tool to be clear and compelling. The need for this sort of tool is clear, so I have no doubt you can make that happen.

---

## [Author Response · Author response to Decision Letter 0]

16 Oct 2020

Response to Reviewer 1

We thank Reviewer 1 for his/her comments and we address these comments below.

Reviewer 1, comment #1

Everything is ok except that the abstract section is tediously long.

Response to reviewer 1, comment #1

We have reduced the abstract from 435 words to 215 words.

Reviewer 1, comment #2

Check all the equations and mathematical symbols for ensuring the correctness.

Response to reviewer 1, comment #2

We have checked all the equations and mathematical symbols to ensure the correctness.

Reviewer 1, comment #3

Some newly published references should be included.

Response to reviewer 1, comment #3

We have included some newly published references like [66,67,114,115] in the Introduction: Section 2.1 Software Pipelines, and in the Related Work section 9.1.

Response to Reviewer 2

We thank Reviewer 2 for his/her comments and we address these comments below.

Reviewer 2, comment #1

This paper, “Data Analysis and Modeling Pipelines for Controlled Networked Social Science Experiments,” shows a systematic research framework to systematically integrate human-subject experiments, agent-based modeling, and simulations in social sciences. I appreciate the authors’ approach to use the computer-science idea of “pipelines” (more generally, the concept of encapsulation) to make social-science methods more reusable and thus more efficient and verifiable. I think, however, the authors could do better to clarify the original contribution of this paper as a report of scientific research.

Response to reviewer 2, comment #1

We have attempted to clarify the purpose of the paper and the contributions. We have modified the Abstract and Section 2 significantly to emphasize the original contributions of our work. In particular, we dedicate new Section 2 to the novelty and contributions.

Specifically, in the Abstract, we now state:

“Our contribution is to describe the design and implementation of a software system to automate many of the steps involved in analyzing social science experimental data, building models to capture the behavior of human subjects, and providing data to test hypotheses. The proposed pipeline framework consists of formal models, formal algorithms, and theoretical models as the basis for the design and implementation.”

At the start of Section 2, we state:

“Our work is to provide an automated and extensible software system for evaluating social phenomena via iterative experiments and modeling.”

Then, in Section 2.1, after we introduce the pipelines, we state: “The focus of this paper is on formal theoretical models, and the architecture, design, implementation, and use of the pipelines that instantiate these models in software. The goal of the software system is to automate many of the steps in analyzing social science experimental data, and building and exercising models.”

Reviewer 2, comment #2

As the authors mentioned in the paper, they have shown the original idea in the paper [35] (and used the developed software in [7, 23, 45]). Actually, I found several figures (Figures 2 and 3) and descriptions of this paper are also shown in [35], but there is no citation in the figure captions and the manuscript.

Response to reviewer 2, comment #2

We now have a citation in the figure captions. Figures 3 and 4 in the manuscript are similar to the Figures 1 and 2 in [35], but they are updated versions. Figure 3 includes a newly implemented pipeline for this work, the Experimental Data Transformation Pipeline (EDTP). We redrew Figure 4.

In Figures 3, 4 and 9 in the manuscript, we now cite [35]. All other figures are different.

Reviewer 2, comment #3

Although this paper has Section 1.7 that refers to the extensions from the original paper [35], I would suggest that the authors should focus on meaningful advancements for scientific contributions from [35].

Response to reviewer 2, comment #3

This has been done. It now appears as Section 2.4.

Reviewer 2, comment #4

The authors stressed this paper’s originality at p. 9: “The purpose of this paper is to describe the software system used in those studies ([7, 23, 45]), and that can be used for other networked experiments.” However, I do not see just detail technical specification as a report of scientific research. As a scientific report, the authors should show that the technical specification and the extensions of [35] address methodological challenges in social sciences.

Response to reviewer 2, comment #4

Thanks for pointing it out. The sentence that we wrote and that you quoted is not very precise in the original manuscript. In this revision, we have clarified this point. Our work in this paper is the first presentation of these pipelines. So, the purpose of this paper is to present the design and implementation of a software system to analyze social science experiments. Furthermore, the design is based on formal theoretical models that we develop and that provide a principled way of designing the system.

The challenges are summarized in Section 1.2 and the extensions from [35] are in Section 2.4.

Reviewer 2, comment #5

In this regard, I suggest that the authors verify the effect of the developed framework more carefully. The authors pointed out that the experiments, modeling, and simulations in social sciences “are often conducted a heuristic manner by individual scientists, leading to inefficiencies, duplication and an overall decrease in human productivity. Automating these steps can lead not only to improved productivity, but also to improved scalability and reproductivity.” (p.3) Thus, the authors have developed the software to automate these method steps using the pipeline framework. However, I could not find the evaluation whether their software addresses the challenge of inefficiencies, duplication, productivity, and reproductivity in social-science researches. For evaluation, the authors refer to three case studies (pp. 8-9). However, 1) only the case study 1 has the full process of the proposed system; 2) the single study did neither show any tests in terms of the above challenges nor evaluate the general-purpose versatility of the framework; 3) I do not think just 15-25 subjects are enough to evaluate social-science research methods at scale (p. 9); and 4) the authors used “millions of artificial agents” to complement the scarcity of human subjects (p.9), and that means they used agent-based simulations for evaluation. A research method cannot be evaluated using a component of the research method.

Response to reviewer 2, comment #5

We understand your comments and agree with them. There is much to unpack here. There are two broad themes here, which we take one at a time.

First, our claims about the benefits of our approach come only from our own work. We have modified the text in Section 1.2: “However, current practice often entails producing custom programs and analytical scripts that pertain to the experiments and modeling. Our lab has found that this often leads to inefficiencies and duplication of effort.” In the past, we would build isolated codes, keep track of them, and reason about them in different ways depending on how they were composed, and even based on who wrote the code. We have found by using the pipeline software approach, we realize a much more uniform code base that is more extensible and more maintainable. This all goes toward increased productivity. Our basis of comparison is our past work.

Second, we now address the case studies. Text has been added to the beginning of Section 8.

Yes, only one case study (study 1) addressed the full system. Case study 1 shows the complexity of social science experiments and how the proposed pipeline system enabled automation of many steps. Specifically, that study took two years to complete, in building software, running experiments, analyzing data, building multiple models, validating and exercising models, and hypothesis testing. We executed several loops embodied in Figure 1, for this single case study. The pipelines in this work were used for all of the work in case study 1. That work also involved the entire author list, in various tasks of the work. Thus, case study 1 is a massive one. The project did not have the resources for more efforts like this one. We have made this more clear at the beginning of Section 8.

The general-purpose versatility of the pipelines is demonstrated by the second and third case studies, in Appendix F. We felt it a more powerful argument to demonstrate the pipelines’ applicability to other researchers’ works to demonstrate the system’s versatility.

We agree with your statement on scale. As you state, 15-25 subjects are enough to evaluate social-science research methods at scale. The problem here is our use of the word scale. There are many experiments that have been conducted through web apps that have up to, say, 20 subjects per experiment. We cite several of these. We are focused on this regime of experiment. Hence, we used for “scale” the scale of experiments. We did not intend to mean a population level (e.g., a US city level) scale. This is our error. We deleted this text so that it is not confusing.

You state: “the authors used “millions of artificial agents” to complement the scarcity of human subjects (p.9),”. It was not our intention to imply that we are substituting agents to complement the scarcity of human subjects. We are not doing this. Our only purpose in making that statement was to address the question: If we have some magic capability to generate experimental data on some n number of people in one experiment, what number of human subjects could our pipelines handle? So, we duplicated real data and assigned them to fictitious subjects and ran them through the codes. We were only curious about how many agents the software could handle, on our hardware. We did not mean to imply anything about running massive numbers of experimental subjects. There is a host of well-known issues that arise with massive experiments, as you are probably alluding to. We have deleted these statements to prevent confusion.

We moved much of this text to Section 8, simply to shorten the Introduction, which is why our comments are now there.

Reviewer 2, comment #6

I am also concerned about the research framework itself. Social-science researchers often use lab experiments to understand some causality in social phenomena with randomized controlled settings. Agent-based models and simulations help them to confirm the mechanisms that they discover with experiments or to specify the key factors that they should explore with experiments. From this perspective, the systematic integration of experiments, models, and simulations is reasonable. However, the robust causality of lab experiments come from the careful design to compare the experimental treatments with control. I suggest that the authors should clarify how the technical specifications can increase or at least maintain the strength of lab experiments.

Response to reviewer 2, comment #6

We agree with your comments. We did not intend to subordinate lab experiments to modeling. In Section 2.1, we state the importance of experiments as follows. 

“The importance of experiments, even with modeling, is observed in Figure 3 because experimental data play a major role in pipelines 1, 2, 3, and 5 (four of the five pipelines). Experiments are critical, for example, in establishing causality, by comparing results from control experiments with those using treatments.”

In Section 1.1 we also now state:

“This composability [of pipelines] also enables abduction using an experiment-only approach by removing the modeling activities in Fig. 1.”

Finally, this is one the three broad themes we sought to rectify in the text that is a response to the editor.

Reviewer 2, comment #7

Another concern relates to the closed loop system shown in Figure 2. Such a feedback system, which is often used in engineering, works well to optimize something under the given boundary condition. However, boundary conditions are not always obvious in social-science researches, especially in the exploration phase. With wrong boundary conditions or no external assessment components, the system can be led to an improper sub-optimal status. Thus, lab experiments often need external validity. However, I was not able to find such a function or component in the pipeline specifications. On the other hand, in the testing phase after exploration, I’m not sure why we need to use the generalized pipeline system.

Response to reviewer 2, comment #7

 We would like to address this comment in two parts.

First, our system is not a closed loop system for boundary conditions. These conditions can change from iteration to iteration, as the analyst or researcher desires. In our work, we found the following to be the case. During a loop, we analyzed experimental data and built and exercised models, and those results gave us insights into the experiments that we wanted to perform next (i.e., the experiments changed across the loops). But we had no automated way of making these decisions. Instead, we relied on the expert knowledge of our social scientists to guide these changes.

Second, regarding validation, our system does not analyze an experimental specification to validate that the proposed experiment addresses the phenomenon under consideration. This is an important step, but it is outside the scope of our pipelines. Our pipelines implement and execute software that the researcher specifies, and those specifications will be based on the issues of validation that you raise.

Reviewer 2, comment #8

Furthermore, social-science researches including NESS experiments have specific difficulties to deal with real humans in general. For example, research results might differ between different group of people. Even within the same person, her behavior and decision making might change over time. Thus, for example, the second experiment several months after the first one might give different results because of the drifting. I am not sure how the proposed framework can address such a challenge. I suggest that the authors should specify how the research framework can deal with the difficulties to deal with the variation of social behaviors over population and over time with the real-world content because this study aims to give a general framework for social sciences and not for computer science.

Response to reviewer 2, comment #8

We agree that this is an important issue. However, this question you raise is a question for the social science community. Our scope is the software pipelines and the work they perform, to support social science researchers. Once the experimentalist decides how to study this problem (i.e., what experiments to perform), software may be written to analyze the experiments that the researcher specifies.

Our software architecture can accommodate such needs. However, this would require more functions within the system to analyze such data over time and to model these phenomena. These could be done, but this would first require separate experiments focusing on these issues, which we do not have.

We discuss limitations of our system in Section 10.

Response to Reviewer 3

We thank Reviewer 3 for his/her comments and we address these comments below.

Reviewer 3, comment #1

I appreciate the sophisticated design reflected in this work. It's a thoughtful system. However, it feels a lot more like a white paper or technical report than a scientific paper proving this pipeline is the correct solution, as a consequence, I believe there's an opportunity to reframe the paper as a methods only contribution and to identify upfront what that implies about external adoption of these tools.

Response to reviewer 3, comment #1

We appreciate your constructive comment. In this manuscript, we have reorganized the manuscript significantly to emphasize the methodology contribution of this work. In particular, we split up the original Introduction into two sections with Section 1 focused on the problem and Section 2 focused on the contribution and novelty of this work.

Our view of external adoption of this tool is that it is one part of an integrated, holistic approach to conducting social science research. In Section 2.1 we have clarified this point with more explanations as follows:

“The focus of this paper is on formal theoretical models, and the architecture, design, implementation, and use of the pipelines that instantiate these models in software. The goal of the software system is to automate many of the steps in analyzing social science experimental data, building and exercising models. We presume that in the great majority of cases, no one person is going to identify a social science problem or question; specify experiment requirements and design; build experimental platforms and execute experiments; specify analyses; build software to analyze experiments and perform data analyses; specify, design, build, and validate models of experiments; and evaluate hypotheses. Rather, we view these social science researches as “team science,” and as such, this system is not focused on all members of such a team. So while all team members can have a general appreciation of the need for and value of such a system, the paper is focused on the team members who design and build software to automate many analysis steps. “

Reviewer 3, comment #2

In particular, I believe this requires isolating the design and goals of the system from the implementation, e.g., much of the rhetoric on micro services is only relevant for certain kinds of empirical and modeling problems but the broader implication of the architecture is quite generally useful.

Response to reviewer 3, comment #2

We agree that one of the key ideas in properly building software is to separate goals of the system (requirements) from design, and separating both of these from implementation. This is what we have done. Section 1 provides motivation and requirements, Section 5 provides a conceptual view of the pipeline framework and pipelines (high level design), Section 6 presents a formal theoretical model of the pipeline framework and the h-functions (tasks), and Section 7 (along with appendices) addresses design and implementation. We have added Figure 2 in Section 1 on the roadmap of our paper, emphasizing these relationships that you state, to address your point.

Reviewer 3, comment #3

I also find that this work is in discussion with other innovations in online lab experiment design, which I suggest considering (as a potential context for the design of related systems — e.g., arxiv.org/abs/2006.11398).

Response to reviewer 3, comment #3

We thank the reviewer for this reference. We were not aware of it. It appears to be very complementary to our work. Note that Fig. 4 has a box for experiments, which produce raw data, and tools like Emperica seem to be a natural fit with our analysis pipelines. We have added this to the Introduction, and a couple more references related to the reference you supplied. We have emphasized the complementary nature of online lab innovations and software pipelines.

We have included in the related work your references for Empirica [114] and Facebook Ax [115]. We have included a couple more references [66,67] about web-based platforms for interactive online experiments. They are also mentioned in Section 2.1.

Reviewer 3, comment #4

As a minor extension to this point, I think the empirical context where these tools are important can be outlined in more detail and with more consistent examples, giving a reader a regular sense of the decisions made in the modeling system design.

Response to reviewer 3, comment #4

It is a very good point and we agree with it. We feel it is a bit beyond the scope of this manuscript, which is focused more on the software side. To your point, we now discuss empirical context at the end of Section 2.3, and in the context of our work on collective identity, which uses these pipelines, please see reference [X], which is in Cedeno-Mieles et al. (2020). 

Cedeno-Mieles V, Hu Z, Ren Y, Deng X, Adiga A, Barrett C, et al. Networked experiments and modeling for producing collective identity in a group of human subjects using an iterative abduction framework. Social Network Analysis and Mining. 2020;10(11).

Reviewer 3, comment #5

An exciting extension to the current work would be formalizing the use of this method with adaptive experimentation techniques (e.g., ax.dev).

Response to reviewer 3, comment #5

We agree with your comment, and have included this reference and this idea. Thank you; we were not aware of this work. See Section 9 (Related Work). We have added a few sentences to Section 10, in the context of limitations and future work, based on your comment. We have added limitations based on your comments.

Reviewer 3, comment #6

As a consequence of the points above, I'm suggesting a minor revision, as I don't believe these kinds of changes will take substantially more work. That said, I don't think these issues are trivial. Without some reframing, I don't think this will be suitable for publication as a scientific report.

Response to reviewer 3, comment #6

We have made the changes above, per your comments.

Reviewer 3, comment #7

Minor fixes:

There are several typos, one that I recall is "For each of theses topics" in the caption for Table 2 should be "these"

Response to reviewer 3, comment #7

We have read the manuscript for typos and fixed them.

Response to Reviewer 4

We thank Reviewer 4 for his/her comments and we address these comments below.

Reviewer 4, comment #1

Thank you for giving me the opportunity to review your paper! I have a particular interest in how simulation and experiment can be used together to develop and test social theory, and have grappled with some of the same issues that you raise as I bring together experiment and simulation in my own work. There is a lot of room for methodological improvements to be made in this area, and I am encouraged to see such a competent list of authors thinking seriously about how to do so. =)

I am afraid that I found the paper somewhat difficult to understand, and so my comments on the technical work you describe may miss the mark. I beg your pardon should that be the case!

Response to reviewer 4, comment #1

No response requested.

Reviewer 4, comment #2

This paper introduces a set of software tools to help social scientists integrate simulation of social theories with networked experiments designed to evaluate those theories. Bringing these two components together into a single computational framework speeds the researcher’s ability to cycle through stages of theory development and testing, and improves our ability to learn about the world. The paper proposes a standard language for describing simulations and experiments, such that any experiment/simulation combination that takes advantage of this “abstract data model” should be able to take advantage of the tools proposed here to reduce the overhead work associated with organizing a simulation/experiment workflow. By formalizing the theory building and testing process, the tools should also improve the replicability of research by helping to document the steps that the researcher followed in their research process.

Response to reviewer 4, comment #2

No response is required.

Reviewer 4, comment #3

There is a phrase amongst designers that a tool (product, etc.) should be “useful, usable, and used”. These tools jump right out at me as being very useful, as one of the major challenges to social scientific theorizing is the difficulty of empirical verification, and anything that improves that workflow ought to be widely adopted. However, I find that I am unable to assess whether the tools will be “usable and used”.

Response to reviewer 4, comment #3

On usefulness. We agree that our pipelines are useful. We have found them to be so.

On usable. Our past work has used this system to analyze fairly sophisticated collective identity experiments. See references 7, 45, and 23 in Section 2.3. Those papers do not mention the pipelines; they focus only on social science problems and solutions. Our case studies in this paper are also designed to demonstrate usability: two of our examples focus on how our pipelines can be used to analyze other researchers’ data.

On used. As for “used,” we cannot tell right now. The purpose of this paper is to describe the system, so that people understand it and its principles, and decide whether they want to use it. Hence, this paper is a needed step for the system to be used. 

Empirical verification is challenging. In the current system as described, this has to be reasoned about by social scientists and the team that is specifying the experiments. This is what we did in our work. Currently, we have no pipeline that automates a process of empirical validation. This appears to us to require a separate thread of inquiry.

Reviewer 4, comment #4

Writing a “Tools” paper is really hard. The author of any tools paper knows so much about the technical implementation of the tool, and the difficulty of constructing it, they naturally want to share how those challenges were overcome. If the paper is written for an audience of potential *users*, however, it doesn’t benefit the reader to know what you have done as authors and tool developers. The reader wants to know what *she* can do because of your contributions. (Or at least, I do!) In my reading of the paper, I am unsure how to apply the tools you have developed in my own research - and this is the type of research I do every day.

Response to reviewer 4, comment #4

Thanks for the comment. The case studies, though not exhaustive, are designed to demonstrate the use of the system (Section 8).

Based on our experiences with researchers from social science, it often requires a team to study social science. Indeed, our system was designed, built, and used by social scientists, statisticians, mechanical engineers, and computer scientists. Our description in the manuscript is addressed to two types of individuals: software builders and others. The “others” include social scientists, statisticians, and data analysts. Sections 1 through 4, and 8 are intended for this group. For software builders, these sections, and Sections 5 through 7, are intended. There are exceptions to this categorization, of course. But, for example, none of our social scientists could have designed and built this system, just as none of our software team knows social science theory and principles as our social scientists do. 

If you have software developers on your team, we are confident that Sections 5 through 7, and the accompanying appendices, will resonate with them and will be useful to them. These details are required so that software developers can act on the ideas in this paper and operate and extend the system. 

We have added a paragraph to Section 2.1, on the team science aspect of this work, to make this clear.

Reviewer 4, comment #5

The reason for my confusion may be that much of the paper deals with the technical implementation of pipelines, h-functions, microservices, and other pieces of machinery; and yet I don’t understand how these elements relate to me as a social scientist. I’m not sure if the solution to this is to provide more exposition (so that I know why I need the technical details), or to switch the focus away from implementation towards use. For example, when I reach line #615 “Let P be a collection of pipelines, with pipeline P∈P represented as P(Q,ˆQ,SID,S,TID,T,H).”, I am not sure why I need to be considering sets of pipelines at this point in the paper, or how doing so will improve my ability to conduct social science. That is not to say that a formal model of the pipeline is not important to developing the tool, or that presenting it in the paper is unhelpful, merely that it is not relevant to a social scientist trying to decide whether and how to use the authors’ tools. I hate to say it, but I had trouble following the thread of the paper and understanding why each section was included and what it had to teach me.

Response to reviewer 4, comment #5

We agree with the reviewer that the technical implementation of pipelines may get the reader a bit distracted from our key points. In this revision, we have put a paragraph in Section 1.2 for what purpose, and have put a small section at the start of Sections 3-8 to state a roadmap of each section.

The reason for the inclusion of technical details is addressed in our response to your previous comment. Also, the text you cite is in Section 5, and this is addressed in the previous response, too. One additional point: we use theoretical models, such as that in Section 5, to formalize the design and requirements---our software design must implement the features of the model. Here, we would like to remark that these details are important. Researchers often do not include enough information to enable reproducibility. We take this as a complement. Without Sections 5 through 7, and accompanying appendices, one would not be able to understand the system. We have added a statement at the beginning of each of Sections 5 through 7.

As our proposed pipeline system is non-trivial, one has to invest time in playing with them to understand features. In particular, software people will understand these architectural and design decisions and appreciate their importance. Our perspective is that the decision to use our system is a team decision, requiring people with social science and software experience.

Reviewer 4, comment #6

It could be that I am assuming the wrong audience for the paper. Rather than social scientists who might use the tool, this could be for other tool designers working to construct similar types of tools, in which case the level of technical detail may be appropriate.

Response to reviewer 4, comment #6

Our audience can be the entire social science team. We hope that the revised manuscript can provide proper contents for different team members at their desired levels of detail. The manuscript includes the technical details because software team members will want and need it.

Reviewer 4, comment #7

However, at this point I am unable to confidently endorse the technical contribution of the paper for a social science audience. This is not because I have found errors or problems in the underlying tools you have developed, but because I can’t evaluate the work with enough understanding to know if errors are present or not. This is difficult for me to admit as a reviewer - if I was more knowledgeable or clever, would I be able to give a better review? Regardless, I do feel that if this paper is written for an audience of tool users, then each piece of the paper should be comprehensible to that user.

Response to reviewer 4, comment #7

The potential audience of our work could be the full social science team, where different team members will desire different levels of detail. Thus the manuscript includes sufficient detail because the software team members will want and need it.

We hope that our clarifications and changes, itemized above, have helped in understanding this system. We have made several changes to the manuscript based on your and other reviewers’ comments to emphasize the challenges, novelty, and contributions of the work.

Reviewer 4, comment #8

It is difficult for me to make this recommendation, as I can see how much work you have done to develop these tools, and the significant ‘upside potential’ if such tools were in wide use. However, it would truly be unfair to the work that you have done in creating these tools if they were not used, and were not able to attract a community that could push their development forward.

Response to reviewer 4, comment #8

We appreciate your valuable comments. In this revision, we have made significant efforts to clarify the work and why it is useful. Please see the summary table at the beginning of this response.

Reviewer 4, comment #9

If you are committed to supporting these tools into the future, you will no doubt put significant effort into describing the use cases for these tools, and creating user-friendly documentation with concrete examples that users can work through themselves. This material would be inordinately helpful in a paper designed to introduce the tool, such as this one. If I were to make a suggestion as to how this paper could be more tightly integrated, it would be to start with a very concrete case study, in which a named researcher asked a real question about social reality, and then walk through her workflow. At each stage in the workflow, highlight the challenges she faces and why the existing solutions fail. Then, introduce your tool as a way to solve those concrete issues. Use language that acknowledges the user (for example, from the caption for fig 26 “To run a pipeline (called a job), a configuration input file specifies functions and their order of execution”, instead say “In the configuration file, specify which functions should be run, and in what order.”) The passive voice is just impeding clarity. Push the proofs and the technical details to the appendix, and tell me what I can do with the tool, why it’s important, and how to do it. If I don’t have a really good grasp of what is happening, the details of the machinery are meaningless.

Response to reviewer 4, comment #9

We agree with you on making the tools user-friendly. We will follow the standard procedure to put the pipelines into an open-source github repository (https://github.com/).

Thanks for the suggestion on using a more positive voice. We also make proper modifications on the writing of the case study. Also, for Figure 26 and others, we have included in Appendices A through C detailed input file formats and contents: so that analysts have examples of how to generate inputs for analyses.

Reviewer 4, comment #10

Don’t be too discouraged that I’m recommending against publishing the paper as currently written. I am suggesting that the editor be open to receiving another paper from you on this topic that reads more clearly. No rejections are fun, and I know that I am suggesting a substantial rewrite of the paper. But (having done this myself) I don’t think it will be that much more work than you will need to do anyway in creating user documentation, in whatever form that takes. If this work is important to you going forwards, it’s worth creating a community of users around, and to do that, you need the “story” surrounding the tool to be clear and compelling. The need for this sort of tool is clear, so I have no doubt you can make that happen.

Response to reviewer 4, comment #10

Thanks much for your valuable comments. In this revision, we have made significant modifications on the paper and have tried our best to clarify the challenges, contribution, and usefulness of the proposed pipeline framework.

<< end of comments >>

---

## [Decision Letter · Decision Letter 1]

3 Nov 2020

Data analysis and modeling pipelines for controlled networked social science experiments

PONE-D-20-13649R1

Dear Dr. Cedeno-Mieles,

We’re pleased to inform you that your manuscript has been judged scientifically suitable for publication and will be formally accepted for publication once it meets all outstanding technical requirements.

Kind regards,

Ning Cai, Ph.D.

Academic Editor

PLOS ONE

Additional Editor Comments (optional):

One reviewer who'd been holding negative attitude is basically satisfied with this round of revision. However, he requests that the tool referenced be available in some form (for instance, put the manual online). I agree with him in this point. Please link this article to some entry about the tool before publication.

Reviewers' comments:

Reviewer's Responses to Questions

**Comments to the Author**

1. If the authors have adequately addressed your comments raised in a previous round of review and you feel that this manuscript is now acceptable for publication, you may indicate that here to bypass the “Comments to the Author” section, enter your conflict of interest statement in the “Confidential to Editor” section, and submit your "Accept" recommendation.

Reviewer #1: All comments have been addressed

Reviewer #3: All comments have been addressed

Reviewer #4: (No Response)

2. Is the manuscript technically sound, and do the data support the conclusions?

Reviewer #1: Yes

Reviewer #3: Yes

Reviewer #4: Yes

3. Has the statistical analysis been performed appropriately and rigorously? 

Reviewer #1: Yes

Reviewer #3: Yes

Reviewer #4: Yes

4. Have the authors made all data underlying the findings in their manuscript fully available?

Reviewer #1: Yes

Reviewer #3: Yes

Reviewer #4: Yes

5. Is the manuscript presented in an intelligible fashion and written in standard English?

Reviewer #1: Yes

Reviewer #3: Yes

Reviewer #4: Yes

6. Review Comments to the Author

Reviewer #1: It can be accepted in current form. It has been well modified. This version is good enough for being published.

Reviewer #3: I thank the authors for their substantial efforts in improving the paper and addressing my comments. I believe the paper is now ready for publication.

Reviewer #4: Thank you for outlining the audiences you see for this tool. I find the paper significantly easier to work through given the context you described in your response, and in the additions to the paper.

In your new abstract, you list your intended contribution as: “to describe the design and implementation of a software system …”. However, I expect that you also hope for something larger: to provide the tool you have developed in-house to a community of researchers, who could then go on to use the tool themselves. If this is true, some form of user documentation is still needed.

This paper does not need to include user documentation itself. I would feel comfortable with the paper in its current format, with the addition of a sentence to the effect that “user documentation is available at <your here="" url=""> including instructions for installing and getting started with the software”. The paper suggested by Reviewer 3 Comment 3 (arxiv.org/abs/2006.11398) may be a good model - it doesn’t directly show users how to get started, but it directs the user to where documentation can be found.

Releasing a piece of open source software involves more than “putting the pipelines into an open-source repository”, as you mentioned in your response. If you haven’t provided enough supporting user documentation, you haven’t really contributed a tool to the community. Instead you have reported on a tool that is used in-house. I’m not sure either of us would be satisfied if that were the only outcome.

A good model may be the documentation for ax.dev suggested by reviewer 3, (https://ax.dev/docs/installation.html) which includes instructions for installation and example scripts.

As a reviewer of a tools paper, I would love to have the opportunity to try out the tool being presented, as the proof of the pudding is (as they say) in the eating.</your>

---

## [Editor Report · Acceptance letter]

10 Nov 2020

PONE-D-20-13649R1 

Data analysis and modeling pipelines for controlled networked social science experiments 

Dear Dr. Cedeno-Mieles:

I'm pleased to inform you that your manuscript has been deemed suitable for publication in PLOS ONE. Congratulations! Your manuscript is now with our production department. 

Kind regards, 

on behalf of

Dr. Ning Cai 

Academic Editor

PLOS ONE